# QUALITY-DIVERSITY THROUGH AI FEEDBACK

**Herbie Bradley**[1,2,3*]   **Andrew Dai**[4*]   **Hannah Teufel**[4]   **Jenny Zhang**[5,6]
**Koen Oostermeijer**[4]   **Marco Bellagente**[7†]   **Jeff Clune**[5,6,8]   **Kenneth Stanley**[9]
**Grégory Schott**[4]   **Joel Lehman**[10]
[1]CarperAI   [2]CAML Lab, University of Cambridge   [3]EleutherAI   [4]Aleph Alpha
[5]Department of Computer Science, University of British Columbia
[6]Vector Institute   [7]Stability AI   [8]Canada CIFAR AI Chair   [9]Maven   [10]Stochastic Labs

## ABSTRACT

In many text-generation problems, users may prefer not only a single response, but a diverse range of high-quality outputs from which to choose. Quality-diversity (QD) search algorithms aim at such outcomes, by continually improving and diversifying a population of candidates. However, the applicability of QD to qualitative domains, like creative writing, has been limited by the difficulty of algorithmically specifying measures of quality and diversity. Interestingly, recent developments in language models (LMs) have enabled guiding search through *AI feedback*, wherein LMs are prompted in natural language to evaluate qualitative aspects of text. Leveraging this development, we introduce Quality-Diversity through AI Feedback (QDAIF), wherein an evolutionary algorithm applies LMs to both generate variation and evaluate the quality and diversity of candidate text. When assessed on creative writing domains, QDAIF covers more of a specified search space with high-quality samples than do non-QD controls. Further, human evaluation of QDAIF-generated creative texts validates reasonable agreement between AI and human evaluation. Our results thus highlight the potential of AI feedback to guide open-ended search for creative and original solutions, providing a recipe that seemingly generalizes to many domains and modalities. In this way, QDAIF is a step towards AI systems that can independently search, diversify, evaluate, and improve, which are among the core skills underlying human society's capacity for innovation.[1]

## 1   INTRODUCTION

Human innovation is not only a generative capacity for creativity, but also includes the ability to evaluate the subjective quality of new ideas and artifacts. Great ideas are rarely generated all at once out of whole cloth, but rather gradually emerge through divergent chains of elaboration and revision (Stanley & Lehman, 2015). To successfully navigate such a tree of ideas, the creator must evaluate which steps in a chain are worth pursuing further, a question that can be highly subjective, especially in domains with artistic or literary dimensions.

Until now, even if AI could provide candidates, the hope for such subjectively tinged evaluation lay firmly with humans. However, the emerging foundation model technology of recent years (Bommasani et al., 2021) now means that the model can also play the role of evaluator, even when the evaluation is in part subjective (Madaan et al., 2023). In this way, for the first time, an entire ideation process that returns a diverse set of interesting artifacts can in principle be automated. This process cannot be run by LMs entirely on their own, but requires chaining together a search algorithm with model calls in a nuanced way. This paper highlights one way to achieve this potential: to combine LMs with the field of quality-diversity (QD) (Mouret & Clune, 2015), which centers on how to design search processes that produce high-quality solutions that span a design space.

---

*Equal contribution. Order chosen in support of student authorship. Author contributions listed [here].
Correspondence: andrew.dai@aleph-alpha.com, mail@herbiebradley.com, lehman.154@gmail.com
†Work done while at Aleph Alpha

[1]Project Page: https://qdaif.github.io/

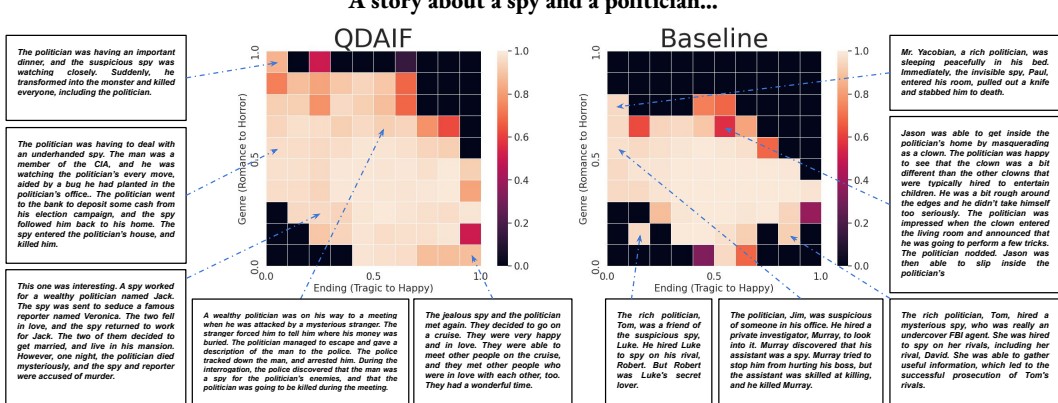

Figure 1: **QDAIF (left) covers more the search space with diverse, high-quality stories compared to the Baseline (right).** The baseline is **LMX, Quality-Only** (Meyerson et al., 2023), which optimizes only for the quality of solutions. QDAIF discovered more interesting stories about a spy and a politician, covering examples such as romance stories with a happy-ending, to horror stories with a tragic-ending. The baseline produced a story (right-middle position, starting with "Jason") with a lower quality score due to the lack of a desired spy character (denoted by the red-colored bin, for a story with a neutral ending, and leaning to horror). QDAIF discovered a better, more-relevant story (bottom-middle position, starting with "a wealthy politician") for this same neutral bin.

The main insight in QD algorithms is to explicitly maintain and seek high-quality diverse responses. Typically such search algorithms require hand-designed measures of diversity and quality, as well as a way to generate meaningful variation. Yet the most interesting and complex domains nearly always involve notions of performance, diversity, and variation that are subjective or difficult to specify algorithmically. Extending work that generates variation through LMs (Lehman et al., 2022; Meyerson et al., 2023) and evaluates the quality of potential solutions through LMs (Ahn et al., 2022), we show that LMs can also be used to evaluate qualitative aspects of diversity. In this way, LMs can instantiate the three main ingredients of QD search, thereby enabling powerful new QD algorithms that can ride the coattails of continual LM advances, which we name Quality-Diversity through AI Feedback (QDAIF). Such QDAIF can explore and return diverse, high-quality responses to an LM prompt through more-intuitive diversity measures, without the need for model fine-tuning (although, it could also be used for LMs to self-improve by generating fine-tuning data (Lehman et al., 2022; Chen et al., 2023)), an interesting direction for self-curated effective learning environments via generated data, towards AI-generating algorithms (Clune, 2019)).

We evaluate QDAIF across three creative writing domains: opinion writing, short stories, and poetry. In such domains, users often enjoy seeing a wide range of possible stories or poems from which to choose or be inspired. Quantitative results indicate that QDAIF significantly outperforms existing baselines. Additionally, through human evaluation, we observe a strong alignment between human and AI-generated feedback, providing empirical evidence that AI feedback is grounded and that the method can work in practice (i.e. it yields improved quality and diversity as measured by humans). Overall, QDAIF brings us a step closer to AI models that can independently search and innovate, keystone abilities of humans that allow them to create culture and science (Stanley et al., 2017).

## 2 BACKGROUND & RELATED WORK

### 2.1 EVOLUTION THROUGH LARGE MODELS

Advancements in language models have enabled new kinds of powerful search algorithms that apply LMs as search operators, e.g. to create text variation or evaluate solutions. While other search algorithms could also be used, this paper creates a QDAIF algorithm by extending upon Evolution through Large Models (ELM) (Lehman et al., 2022), a framework for evolutionary search for code or text that uses LMs to generate intelligent variation (for example through specialized language models trained on code diffs (Bradley et al., 2023b), or through simple few-shot prompting

(Meyerson et al., 2023; Chen et al., 2023)). Most QDAIF results in this paper generate new search candidates through Language Model Crossover (LMX) (Meyerson et al., 2023), a recent and general few-shot prompting approach that can evolve e.g. mathematical expressions, sentences, Python programs, and prompts for text-to-image models, by leveraging in-context learning capabilities of LMs (Brown et al., 2020). The approach is simple: A few existing search candidates are concatenated into a prompt, predisposing the LM to generate new, similar candidates. In this way, LMX enables creating intelligent variation without requiring any specially-trained models. Our experimental implementation builds on OpenELM (Bradley et al., 2023a), a versatile open-source Python library designed for research into LM-based evolutionary algorithms.

## 2.2 QUALITY DIVERSITY ALGORITHMS

Traditional optimization algorithms aim to discover a single high-quality solution, which while appropriate for many situations, can fail to *illuminate* the full range of possible high-quality solutions. For creative and design problems in particular, a user may want to choose what they think is most appropriate from a diversity of such candidates. In contrast, Quality Diversity (QD) algorithms aim to optimize not just for a single optimal solution, but for a diverse set of high-quality solutions (Lehman & Stanley, 2011b; Mouret & Clune, 2015; Pugh et al., 2016; Fontaine & Nikolaidis, 2021). QD algorithms can thus provide a richer landscape of solutions, enabling adaptability and flexibility in addressing multifaceted challenges (Cully et al., 2015). In addition to a quality measure (i.e. an objective function), QD requires a metric such that it can encourage desired axes of diversity. For instance, Lehman et al. (2022) evolved Python programs to design varied locomoting robots, where the diversity dimensions are the robot's height, width, and mass.

A significant limitation of existing QD algorithms lies in their reliance on low-level quality and diversity measures (Mouret & Clune, 2015). This requirement confounds applying QD algorithms to complex and creative domains, such as the creative writing ones explored in this paper. Intuitively, such measures (e.g. sensor readings (Cully et al., 2015), feature engineering (Manning, 2009)) lack the subtlety and depth needed to capture the complexities of human creativity and intuition, e.g. nuances, moods, or cultural references that resonate in human experience. Interestingly, from having trained on vast amounts of human-generated data, LMs can begin to emulate such human-nuanced judgments (cf. Section 2.3). Thus, by employing an LM to evaluate both quality and diversity, QDAIF significantly simplifies and enlarges the range of domains QD can be applied to.

Feedback from learned ML models has been used in prior work to reduce the need for hand-crafted heuristics or expensive ground-truth evaluations. In model-based QD, surrogate models supply learned feedback. Gaier et al. (2017) introduced the use of surrogates (via a Gaussian process) to predict fitness (quality). Subsequently, Keller et al. (2020) introduced a learned model to predict both fitness and behavior characteristics (diversity), becoming a standard approach (Lim et al., 2021; 2022; Zhang et al., 2022; Bhatt et al., 2022). Surrogate models require domain-specific training data to update their predictions, whereas AI feedback leverages off-the-shelf instruction-tuned LMs (Chung et al., 2022; Ouyang et al., 2022) to automate expensive human feedback for a variety of evaluation tasks. More recently, Fontaine & Nikolaidis (2021) utilized CLIP embeddings (Radford et al., 2021) as both quality and diversity measures to navigate the search space of StyleGAN (Karras et al., 2019), producing a range of faces with the desired characteristic (e.g. "A person with red hair"). We show that using pre-trained surrogate models is more prone to reward hacking in the natural language case (Skalse et al., 2022; Lehman et al., 2019) (cf. Appendix A.2). Hence, QDAIF capitalizes on the strengths of general-purpose LMs for evaluating generated solutions.

## 2.3 AI FEEDBACK

Recently, LMs have been leveraged to provide feedback on the training, evaluation, or problem-solving capabilities of other LMs (Bai et al., 2022; Perez et al., 2022; Shinn et al., 2023; Wang et al., 2023a; Colas et al., 2023; Zhang et al., 2023; Lee et al., 2023). Bai et al. (2022) show that using LM-generated critiques and refinements has been instrumental in enhancing performance on metrics like helpfulness and harmlessness. One promising direction for AI feedback is self-refinement, where LMs evaluate and score their generations, then iteratively improve their output (Bai et al., 2022; Madaan et al., 2023). Self-refinement demonstrated significant improvement in output quality as gauged by human evaluators (Madaan et al., 2023), underscoring generation-discrimination

Figure 2: **Overview of Quality-Diversity through AI Feedback (QDAIF).** Dark components are where Language Models (LM) are employed. QDAIF randomly selects a solution from the QD archive. This chosen solution (parent) forms part of the prompt that is fed into an LM, undergoing LMX mutation to produce a new solution. An LM then evaluates the quality and diversity attributes of the new solution. We compare the newly evaluated solution with its existing solutions in the QD archive, and update it.

discrepancy (Saunders et al., 2022, p.12) (i.e. it is often easier for a model to evaluate the quality of a generation than to generate the same quality in text). Complementary to single-objective optimization with self-refine, QDAIF utilizes AI feedback to assess diversity in addition to quality, facilitating more varied and improved text generation over multiple iterations of refinement.

## 3 APPROACH

Figure 2 provides an overview of the approach, which is to extend a common QD algorithm (MAP-Elites) with LM operators that generate variation, as well as evaluate both the quality and diversity of candidate solutions. The result is a search algorithm capable of iterative discovery and refinement, applicable to subjective text-based domains.

**MAP-Elites.** Our QDAIF implementation builds upon MAP-Elites (Mouret & Clune, 2015), a widely used QD algorithm (Lehman et al., 2022; Cully et al., 2015; Nilsson & Cully, 2021; Vassiliades et al., 2016). MAP-Elites discretizes the diversity space (i.e. dimensions of relevant diversity) into a grid, called the archive. The overarching objective is to populate each grid bin (or cell) within the archive with as high-quality a solution as possible. An iteration in MAP-Elites follows these steps: (1) randomly select an existing solution from the archive, (2) mutate the chosen solution to generate new solutions, (3) evaluate the new solution's quality and diversity characteristics, and (4) if the new solution is higher quality than the current occupant at the cell corresponding to its diversity characteristics, replace the previous cell occupant solution with the new solution. For a new solution to be added to the archive, it has to improve either the quality or the diversity of the grid, meaning that it has to either fill an empty bin or perform better than the solution already in its bin. QDAIF distinguishes itself from standard MAP-Elites in four key areas: archive initialization, solution mutation, solution evaluation, and grid discretization (cf. Figure 2). We provide details on each of these differences below.

**Initialization and Mutation.** For archive initialization, QDAIF employs few-shot prompting, generating solutions based on a hand-chosen set of seed examples. We list in Appendix A.21 the three few-shot examples utilized in each domain, each chosen to span a breadth of diversity characteristics. For example, in a domain where you want diversity of sentiments (like the Opinions domain described in Section 4.1), the few-shot examples demonstrate positive, neutral, and negative sentiments. For solution mutation, QDAIF employs LMX-Near (referred to as "LMX" for brevity in the rest of this manuscript), as detailed in Meyerson et al. (2023). LMX evolves varied text representations (e.g. mathematical expressions, sentences, Python code) by leveraging effective in-context learning (Brown et al., 2020). LMX prompts are kept simple, typically starting with *"Here is a random example of"*. Appendix A.22 shows the full LMX prompts. We also introduce a novel mutation method with instruction-following prompts for poetry in Section 4.4.

**Archive Measures.** While it is sometimes feasible to devise hand-crafted heuristics to evaluate the quality of a solution (e.g. efficiency in locomotion) or diversity characteristics (e.g. a robot's size and mass), this approach falters as domains become more complex and nuanced, as in creative writing. For example, hand-crafting robust heuristics for qualitative aspects of a story, such as its genre (e.g. romance vs. horror), is very difficult. QDAIF circumvents the need for hand-coded measures through prompting LMs with easily-written natural language queries to generate feedback. In particular, capable LMs trained on expansive text corpora can begin to mirror human intuition across a range of potentially subtle diversity characteristics.

**Quantifying Performance and Diversity.** For quality assessment, we prompt the LM to discern whether the input text contains a high-quality solution or pertains to the requested topic, requesting a "yes" or "no" response. The solution's quality estimate is derived from the logarithm of the probability of the LM predicting one response versus the other response. Similarly, for diversity evaluation, we guide the LM to identify a particular diversity trait. For instance, in an opinion generating domain, the LM is prompted to gauge a solution's sentiment, with a requested response of "positive" or "negative". The log probability of these responses serves as our measure of solution diversity. Appendix A.22 shows the full prompts used in each domain to evaluate the solutions. We also introduce a novel categorical approach to evaluate solution attributes based on raw predictions of discrete labels in Section 4.4.

**Discretization.** MAP-Elites typically partitions the grid into equally-sized bins, from the intuition that all parts of the behavior space are equally interesting. However, we observe that when assigning a bin along the diversity axis - which is in our approach based on logits of an LM AI feedback - that qualitative changes in behavior do not uniformly correspond to changes in the logits (cf. Appendix A.31). This is likely due to the (non-linear) calibration behavior of instruction-tuned models in predicting the labels (as output tokens) of text passages (Jiang et al., 2021). Hence, we use custom non-uniform bins, which are denser towards range ends. Qualitative analysis of the generated text showed that the non-uniform bins yielded better alignment with typical human perceptions of diversity changes, influenced by both the AI model's calibration and the domain-specific goals.

**Models and Setup.** Details on the LMX generation model (Appendix A.24) and finetuned AI feedback model (Appendix A.25) are given, with details on the training of these LMs. Additional default hyperparameters are described in Appendix A.27.

# 4 EXPERIMENTS ON CREATIVE WRITING DOMAIN

## 4.1 SETUP: OPINION WRITING, SHORT STORIES

To demonstrate the versatility of QDAIF in different applications of creative text evolution, we evaluated QDAIF on these domains: **Opinions**, and **Stories**. The **Opinions** domain is focused on generating diverse, realistic pieces about one's opinions on eating vegetables and plant-based foods - the diversity measure is based on the sentiment of opinions on this topic (e.g. shown in example texts in the Figure 2 overview). For the **Stories** domain, the topic is about a short story, containing two characters: a spy, and a politician. The diversity of stories is evaluated using a variety of measures based on AI Feedback, with the main ones being: **Stories - Genre** (Romance vs Horror) (1D archive), **Stories - Ending** (Happy vs Tragic) (1D archive), and **Stories - Genre and Ending** (2D archive). These domains capture the strengths and limitations of all methods, ranging from simple (**Opinions**) to challenging (**Stories - Genre and Ending**). We show in Figure 1 that the 2D domain is challenging, yet QDAIF still outperforms the baseline in filling the archive with diverse, high-quality stories. The AI feedback prompts are outlined in Appendix A.23.

**Evaluation.** To assess the performance of methods in creative writing generation, we compute QD scores (Pugh et al., 2016), a standard metric used to measure the quality-diversity of the discovered corpus of texts. A QD score is defined as the sum of the highest quality values found in each bin. To understand the alignment between AI and human feedback for practical applications in QDAIF, we conducted a human evaluation study on selected elite samples from each method (chosen from the median QD score run out of 5 random seed runs). Using a Likert scale (Allen & Seaman, 2007) for quality assessment, we evaluate the capability of each method to produce a collection of diverse, high-quality texts. To do so we calculate a "human" QD score, defined as the sum of quality scores given for all diversity categories identified by the annotator within the set. Furthermore, to understand how closely AI feedback aligns with human perspectives on subjective evaluation, we measured the agreement rates between human annotators and AI and between two human annotators. Details of the human study are specified in Appendix A.1, demonstrating the validity and advantages of AI feedback in generating human-like feedback on subjective quality and diversity measures.

## 4.2 COMPARISONS BETWEEN QDAIF AND BASELINES

To evaluate the strengths and limitations of QDAIF in generating high-quality and diverse creative writing texts, we compared our method against the following baseline methods:

- **Fixed-Few-Shot**: Use a fixed few-shot prompt (cf. Appendix A.21) to sample many completions for creative domain texts (i.e. no iterative search).

- **Shuffling-Few-Shot**: Shuffle the in-context examples of **Fixed-Few-Shot** prompt, before sampling the completion from this prompt.

- **Random-Search**: Create a prompt pool of examples (initialized from examples in Appendix A.21), add all completions from few-shot prompting to the pool, and choose few-shot examples from the growing pool (without pool size limit).

- **LMX, Quality-Only**: Maintain a pool as in **Random-Search**, but only up to 100 highest-quality completions (as evaluated by AI Feedback) are kept in the pool (i.e. iterative search focused only on quality). Single-objective LMX as in Meyerson et al. (2023).

We choose a variety of baselines, some highlighting representative alternative approaches (e.g. few-shot prompting), and ablations of QDAIF, to validate our algorithmic choices. For example, **Fixed-Few-Shot** and **Shuffling-Few-Shot** enable the generation of different texts (relying on stochastic sampling), while being constrained to the output distribution of the fixed set of in-context examples. **Random-Search** and **LMX, Quality-Only** are methods where the (prompt) pool of examples that we can sample from grows in size, starting from an initial pool. In contrast to QDAIF, **Random-Search** is limited by the lack of constraints in the prompt pool, especially in maintaining the quality of the growing pool through evaluation. **LMX, Quality-Only** adds a quality-based evaluation step with AI feedback that optimizes the quality of texts in the pool over time to contain only texts with high quality scores, but is not designed to encourage diversity in texts in comparison to QDAIF.

For each domain and baseline described above, we recorded runs for 2000 iterations, repeated with 5 different random seeds. To enable comparisons with baselines, AI feedback is used to compute the quality and diversity measures for all iteration outputs. Similar to QDAIF, the baseline methods use hand-written examples in Appendix A.21, either in a fixed prompt or as the initial prompt pool.

**Performance Comparison.** We report results comparing the QD score performance for the different methods of generating creative writing texts in Figure 3. We computed the mean, and the bootstrapped 95% confidence intervals (CI) from 100k resamples, across 5 random seeds of runs. We noticed that QDAIF achieves significantly better QD scores than all baseline methods in **Opinions** and **Stories**. The broader range of texts generated by QDAIF is also evident qualitatively. For example, in the Stories - Genre and Ending domain, while the baseline methods deliver straightforward and more predictable stories of how the spy "pulled out a knife and stabbed [the politician] to death", QDAIF generates a more dramatic story of how the spy "transformed into the monster and killed everyone". **Random-Search** is the worst-performing method overall, with significantly lower QD score performance in **Opinions** and **Stories - Genre** compared to the best-performing baselines. Interestingly, **LMX, Quality-Only** does not significantly outperform the methods using a fixed population prompt pool (**Fixed-Few-Shot** and **Shuffling-Few-Shot**). On **Stories - Genre**, **LMX, Quality-Only** is often weaker than **Fixed-Few-Shot** and **Shuffling-Few-Shot**. The results show that single-objective optimization cannot guide the search for diverse, high-quality texts alone. Furthermore, we found that QDAIF outperforms other (diversity-seeking baselines), as well as extensions of those baselines with quality filters, becoming more similar to QDAIF (cf. Appendix A.8).

**Human Feedback Evaluation.** We report the results of the human study comparing QDAIF and baseline samples in Table 1. We observe that compared to baselines, QDAIF is competitive with or better at discovering diverse, high-quality texts in **Opinions** and **Stories** according to human feedback. QDAIF sets also showed high agreement between humans and AI feedback on the diversity categories of presented texts, as well as between two annotators, competitive with **Fixed-Few-Shot**. Although the average perceived quality of texts is better from **Fixed-Few-Shot**, this is not enough for bringing high-quality examples for different niches of outputs (i.e. higher QD score). Furthermore, **Shuffling-Few-Shot** demonstrates even lower human evaluation scores, despite the use of the same fixed set of hand-written seeds, indicating lower robustness due to the use of different ordering of few-shot examples. Prior work hints at the sensitivity of LMs to few-shot prompt ordering, with task-solving capabilities varying significantly due to this ordering (Lu et al., 2021). The gap in human-evaluated performance between **Fixed-Few-Shot** and **Shuffling-Few-Shot** indicates that reliance on fixed prompts is less likely to enable reliable search, in contrast to the robustness shown by QDAIF. Additionally, **Random-Search** and **LMX, Quality-Only** obtained even lower human evaluation scores, even though the methods either explore different prompts or optimize for the quality

Figure 3: **QDAIF significantly outperforms baselines in QD score performance in all domains**. Performance stats with mean bootstrapped 95% CI, across 5 random seed runs. The maximum possible QD score is 20 (100 for 2D archive (4th plot)). See Appendix A.7 for additional stats.

Table 1: **QDAIF is competitive/better in terms of Human QD score against baseline methods from the human evaluation study.** The stats are averaged across three domains: **Opinions**, **Stories - Genre**, and **Stories - Ending**. The Human QD score quantifies the perceived quality-diversity of a set of solutions returned by a method (i.e. high-quality texts for all the diversity categories of texts that the evaluator could identify). Quality rating is from humans. See Appendix A.1 for study setup.

| Method | Human QD score | Quality rating | Human-AI agreement | Human agreement |
|---|---|---|---|---|
| Fixed-Few-Shot | 0.767 | 4.133 | 0.800 | 0.867 |
| Shuffling-Few-Shot | 0.696 | 3.500 | 0.700 | 0.667 |
| Random-Search | 0.606 | 3.300 | 0.733 | 0.600 |
| LMX, Quality-Only | 0.650 | 3.533 | 0.633 | 0.733 |
| QDAIF (ours) | 0.772 | 3.900 | 0.833 | 0.800 |

of texts. We provide a detailed discussion (for baseline methods) on findings from the subjective study of the discovered texts in Appendix A.15, as well as the qualitative behavior of the text search over time in Appendix A.16. Through guided evolutionary search, QDAIF surpasses all baseline methods in terms of computed QD score performance, and is competitive (or better) compared to baselines, according to human evaluation.

### 4.3 EXTENSIONS TO AI FEEDBACK AND MUTATION MODEL

In addition to experiments with QDAIF described in previous sections, we investigated the effects on the performance due to variations of the method.

**LMX Model Size.** We used larger versions of the LMX models (30B and 70B) for mutation, and compared it to the performance of the 13B model (default). While no relationship was found between model size and QD score, quality ratings from human feedback improved with outputs from larger models (described in detail in Appendix A.3).

**Few-Shot AI Feedback.** We compared the performance of QDAIF on the **Stories - Genre** domain when we prompted our AI feedback model for diversity measures given 2-shot, 4-shot, and 8-shot prompts. Using a higher number of few-shots led to improvements in human quality ratings of texts. Further discussion and results are highlighted in Appendix A.4.

**Varying Initialization and Mutation Method.** Ideally, QDAIF would be simpler if it could be run without seed examples (e.g. requesting a story from an instruction-following LM). We investigated the potential of QDAIF when the solution population is initialized from zero-shot prompted generations, and evolved using LMX. Initial results on **Opinions** and **Stories** along with prior work discussion are shown in Appendix A.5, highlighting comparable performance in terms of QD score, with divergence observed in alignment with human preferences. We also find that QDAIF, in these domains, is robust to the mechanism of generating variation. Appendix A.6 describes an alternative mutation method (based on a more gradual few-shot replacement operation) that is more effective in some circumstances, although in general offers comparable performance. We provide a detailed discussion (for QDAIF methods) on findings from the subjective study of the discovered texts in Appendix A.13, as well as the qualitative behavior of the text search over time in Appendix A.14. There may be many ways of leveraging LMs to generate variation (including finetuning models; see Appendix A.11), and this is an exciting avenue of future research.

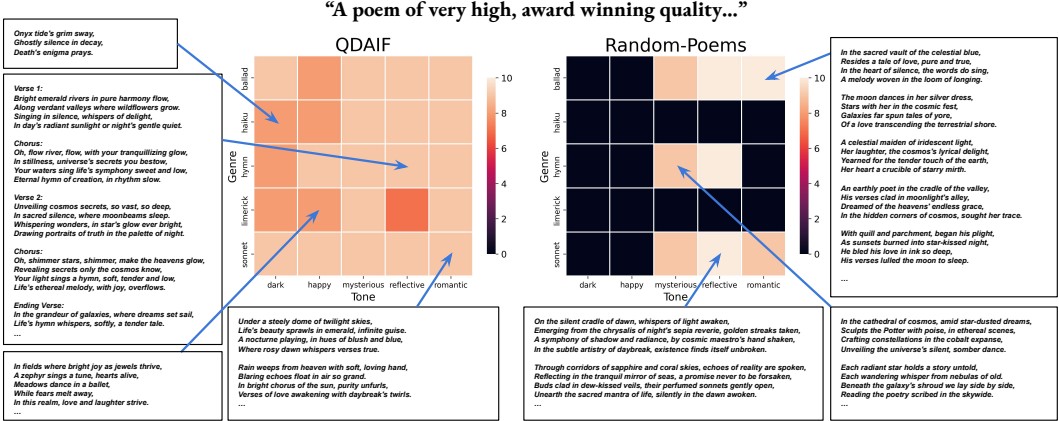

Figure 4: **QDAIF (LMX-guided) (left) covers the space of poetry with high-quality solutions (on a rating scale), with poems matching the closest bins.** QDAIF solutions take qualitative inspiration from the seed poem's imagery of *"fields of green waves"* in Appendix A.30 while giving meaningfully diverse kinds of poems across the search space. QDAIF (LMX-rewrite) (not shown) also covers more the space of diverse, high-quality poems compared to **Random-Poems** (right).

## 4.4 EVOLVING SOLUTIONS THROUGH INSTRUCTION GUIDANCE

This experiment explores scaling up QDAIF to a more capable model, GPT-4 (OpenAI, 2023), in the challenging task of generating poetry, highlighting how QDAIF will benefit from advances in model capabilities. The aim of the **Poetry** domain is to produce high-quality poems with varying genres and emotional tones, unrestricted by topic. Here, the MAP-Elites archive has two axes of diversity: genre and tone, and they are delineated using categorical labels. The genre axis has the labels "haiku", "sonnet", "ballad", "limerick", and "hymn", while the tone axis has the labels of "happy", "dark", "mysterious", "romantic", and "reflective". We created a new mutation operator, **LMX-rewrite**, for this domain, that leverages instruction-following to tell a model to translate a parent poem into an offspring with different diversity characteristics. To generate a new solution, we prompt GPT-4 to rewrite the selected poem into an inspired, different poem: *Inspired by this poem, write a new poem of very high, award winning quality with a different poetic genre (format and form) and tone compared to the poem above.*

We used GPT-4 to determine the genre and tone of a poem and rate its quality. For quality, we prompt GPT-4 to *Rate the quality of the above poem on a scale from 1 to 10*. To determine the diversity attributes, we prompt GPT-4 to determine which genre or tone is the poem closest to. For example, to determine genre, we ask GPT-4: *What genre is this poem closest to from the following list: ["haiku", "sonnet", "ballad", "limerick", "hymn"]?* Appendix A.30 shows the full prompts and setup. We observed high consistency in GPT-4's responses; across multiple LM calls, quality ratings for the same poem fluctuated by no more than a single point. We show qualitative examples of poems with their AI feedback evaluations in Appendix A.19. In this setup, all methods are run for 200 iterations, and QDAIF is benchmarked against the baseline, **Random-Poems**, as well as an ablation method, **Fixed Seed Rewrite**. **Random-Poems** simply generates 200 random poems. **Fixed Seed Rewrite** rewrites only the seed poem in Appendix A.30, without evolutionary search.

We found that QDAIF achieves a higher QD score of 130 (CI: 118 - 145) in comparison to **Random-Poems** with 76 (CI: 67 - 85) and **Fixed Seed Rewrite** with 99 (CI: 72 - 117). We observed a similar trend (with a wider performance gap between QDAIF and other methods) when we used GPT-3.5-Turbo for the generation step instead of GPT-4 while keeping GPT-4 as the evaluator (cf. Appendix A.17). QDAIF was shown to have greater QD score performance than the other methods according to the Mann-Whitney U Test ($p \leq 0.05$). Still, QDAIF with **LMX-rewrite** fails to discover solutions in some bins (e.g. limericks). For this, we can adapt QDAIF by adding guidance on the desired (randomly chosen) genre and tone for rewriting (**LMX-guided**). The performance of this method of QDAIF is on par with a **Targeted-Poems** approach (generating high-quality poems of randomly chosen genre and tone per step) in terms of QD score, and even better when using an older version of GPT-4 (cf. Appendix A.17). Furthermore, we found that the rewriting step is useful

for generating poems that can meaningfully preserve inspiration from parent poems, enabling users to control with more nuance the outcomes of search, through approximating the iterative refinement typical of human creative processes (Stanley et al., 2017) (see Appendix A.18). Figure 4 highlights the potential of QDAIF with GPT-4, especially in controlling a set of solutions subjectively aligned with AI feedback labels. QDAIF is even demonstrated for practical applicability to a domain outside of creative writing, for solving coding problems (cf. Appendix A.20). Overall, results highlight that diversity suffers when prompting models without explicit diversity guidance (Renda et al., 2023; Friedrich et al., 2023; Kirk et al., 2023), and that evolution with a rewriting mutation operator can lead to more human-influenceable, diverse, and high-quality solutions.

## 5 DISCUSSION AND CONCLUSION

This paper introduces QDAIF, a quality-diversity method that aims to discover diverse and high-quality solutions in qualitative domains, by leveraging advances in foundation models to evaluate the quality and diversity of generated individuals. QDAIF outperforms baseline methods in returning more diverse, high-quality solutions in creative writing domains (**Opinions**, **Stories**, **Poetry**), that benefit greatly from accurate AI feedback measures. The paper's results highlight that QDAIF can succeed at its aims, generating solutions that align with human perception of quality and diversity.

We note limitations with QDAIF that motivate future work. Firstly, we suspect reward hacking when using LMs to generate feedback. Our human evaluation investigation shows that while the LM's evaluation of quality mostly aligns with human perception, the correlation drops when the evaluated quality is 0.995 to 1 (cf. Figure 5). The text generation might have exploited attributes or phrasings that allow an LM to give a high-quality estimate, but not what humans would agree is good. This is a common issue highlighted by other works when using AI models as classifiers or evaluators (Nguyen et al., 2015a), highlighting risks of open-ended search to be tackled (Ecoffet et al., 2020). One method to address this limitation could be to use RLHF finetuning (Ouyang et al., 2022) to produce LMs that can detect and mitigate adversarially generated texts. Another possible approach could be to use an ensemble of different AI models to evaluate solutions, rather than relying only on one; the hope would be that robustness would result from models having uncorrelated blind spots.

Furthermore, although QDAIF makes it easy to specify qualitative aspects of diversity through natural language prompts, it still requires specified definitions of diversity axes. For example, if we applied QDAIF to generate short stories of different genres (e.g. comparing horror vs. romance), it would not autonomously explore other important attributes that a writer might care about (e.g. first-person vs. third-person perspective) unless explicitly specified. When we tested different diversity measures in the Stories domain, such pathologies were observed (Appendix A.32). For example, when using "hero spy vs. hero politician" as the diversity measure, many generated stories tend to neglect the interaction between the spy and the politician, focusing solely on one hero character. However, someone writing a story would naturally care about how multiple characters interact with one another. One possible way to automatically determine interesting diversity measures (Meyerson et al., 2016; Etcheverry et al., 2021) is to utilize the human notions of interestingness distilled into foundation models (Zhang et al., 2023). That is, we could ask LMs to suggest interesting diversity measures that a human would typically care about in the domain, thereby enabling a more autonomous creative search (see Appendix A.10 for findings on the potential of this method).

In conclusion, we show that QDAIF is a promising approach to open-ended search that can reveal unexplored creative writing spaces, surpassing alternative text generation methods in generating diverse high-quality natural language text. AI feedback, Evolution through Large Models (ELM), and quality-diversity search (QD) were found to be essential ingredients for enhanced AI systems that can innovate in subjective spaces, similar to past research on Innovation Engines (Nguyen et al., 2016; 2015b). In fact, we see AI feedback as a general ingredient for open-ended search for solutions in multimodal domains, capable of following instructions beyond text (Liu et al., 2023). QDAIF can be easily extended to multi-modal domains (e.g. vision-language) for synthetic data generation and evaluation, building on top of recent advances in the field (Eichenberg et al., 2021; Alayrac et al., 2022; Bellagente et al., 2023; Driess et al., 2023; Bhatt et al., 2023; Sudhakaran et al., 2023; Todd et al., 2023). We see many possibilities from QDAIF to build creative search systems with evaluation, diversification, and improvement capabilities, bringing us closer to AI that can support and extend human innovation.

ETHICS STATEMENT

Human evaluations were performed by the co-authors of this paper and select colleagues. All human evaluators provided informed consent, and their feedback and assessments were obtained without coercion or bias. We took action to prevent bias by presenting evaluators with texts to evaluate in a blind setting, with only the instructions for the study annotation task presented (to carefully read through the presented texts, then give a quality score and a label of the characteristic that best matches the texts). We show a detailed setup for the human study in Appendix A.1.

For the sake of transparency, we provide the full set of results with caption descriptions from our human evaluation. In the **Opinions** domain, Tables 13–16 contain the human evaluation results for sets from baseline methods, Tables 29–32 contain the human evaluation results for sets from QDAIF methods, and Tables 25–28 contain the human evaluation results for sets from embedding feedback QD methods. In the **Stories - Genre** domain, Tables 17–20 contain the human evaluation results for sets from baseline methods, and Tables 33–36 contain the human evaluation results for sets from QDAIF methods. For the **Stories - Ending** domain, Tables 21–24 contain the human evaluation results for sets from baseline methods, and Tables 37–40 contain the human evaluation results for sets from QDAIF methods.

AUTHOR CONTRIBUTIONS

Herbie developed the setup and framework for the Poetry domain experiments and base library for research. Andrew developed the setup and experiments for the Opinions and Stories domains, and contributed to extended studies, visualization, and analysis across experiments in the paper. Hannah contributed additional experimentation in the Stories domain, in addition to coordinating part of human evaluation studies. Jenny contributed qualitative analysis across studied domains. Koen developed visualization scripts used in Opinions and Stories domain experiments. Marco contributed to part of the technical implementation and ideation. Andrew conducted the blind human evaluation study, and Grégory advised on the conduct and analysis of the human study. Joel, Jeff, and Ken initiated early ideation for this work. Joel, Grégory, Jeff, and Ken advised and guided. Andrew, Jenny, Herbie, and Joel wrote the manuscript with edits and feedback from all authors.

ACKNOWLEDGEMENTS

We thank Robert Baldock, Samuel Weinbach, Souradeep Nanda, Jan Zierstek, and Andres Felipe Cruz Salinas for insightful discussions and feedback within the lab at Aleph Alpha. We also thank Katherine Hardgrave, David Nugent, Daniel Flood, and Formula Trinity Autonomous for the inspiration that seeded the momentum leading up to this work.

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

## A APPENDIX

### A.1 HUMAN STUDY ON QUALITY-DIVERSITY OF TEXT SAMPLES

A study through human feedback lets us understand how well QD score performance through AI feedback translates to creating a high-quality, diverse set of creative texts from a subjective angle. We compare sets of human feedback evaluations for samples of diverse elites at the end of each run to measure translations from AI-assessed performance to human-assessed performance of methods in generating high-quality, diverse texts. In addition, the Human QD Score (sum of mean quality score for each category/label that is found in the set according to human feedback) gives us a rough understanding of how aligned quality-diversity improvement during the search is with the more subjective notion of quality-diversity; this score is low if the set deemed to cover a wide space of diverse texts from AI feedback fails to subjectively cover the space of desired diversity according to human evaluations. To distinguish between quality ratings from humans vs. AI feedback in this section, we refer to quality scores as those from human evaluators, and fitness scores as those from AI feedback.

To assess the robustness of quality and diversity measures in AI feedback, we carried out a study involving diverse elite samples selected from different bins of the QD archive from our tested runs. Over a total of 28 experiments (of which 4 are from embedding feedback experiment runs), five distinct stories per experiment are reviewed by six annotators. Each (generated) text was independently reviewed by two persons, resulting in a total of 280 annotations.

During the annotation process, we collected a subjective assessment of the quality of the generation using a 5-point Likert scale based on the text quality in terms of flow, plot, presence of repetition, and correspondence to the study's topic. In addition, we assign each text to one of three categories. Two categories were specific to the study performed (such as positive/negative sentiment, romance/horror genre, or tragic/happy ending) and a third category was used when no element of the other two classes was identified.

We took action to prevent bias by presenting evaluators with texts to evaluate in a blind setting, with only the instructions for the study annotation task presented (to carefully read through the presented texts, then give a quality score and a label of the characteristic that best matches the texts). We provide the full set of results with caption descriptions from our human evaluation. In the **Opinions** domain, Tables 13–16 contain the human evaluation results for sets from baseline methods, Tables 29–32 contain the human evaluation results for sets from QDAIF methods, and Tables 25–28 contain the human evaluation results for sets from embedding feedback QD methods. In the **Stories - Genre** domain, Tables 17–20 contain the human evaluation results for sets from baseline methods, and Tables 33–36 contain the human evaluation results for sets from QDAIF methods. For the **Stories - Ending** domain, Tables 21–24 contain the human evaluation results for sets from baseline methods, and Tables 37–40 contain the human evaluation results for sets from QDAIF methods.

We compiled the results of the full study in Appendix A.12, and summarized the stats across the study in the paragraphs below.

**Comparison of quality scores** We observed that both annotators generally have a close agreement in their ratings with an average difference of 0.9 Likert points but there were occasional instances where they may differ by 2 or 3 units. These deviations occurred in 15% and 5% of the cases, respectively. This reflects that assessing the quality is somewhat subjective. To get a final estimation of the quality of the generated texts, the scores from both annotators were averaged.

Figure 5 shows the average quality ratings from annotators for various ranges of fitness (here, the quality score obtained from AI feedback). The ranges were chosen in a way that ensured a similar amount of samples in each range. We observe clear evidence of a correlation between human and AI quality measures, indicating the usefulness of using AI feedback to assess quality. However, we also observed that fitness for the texts with the highest scores becomes uncorrelated to human-assessed quality. This implies that above a certain threshold, fitness is not a reliable measure of quality in some cases (and maybe slightly lower fitness solutions were preferred by humans). Therefore, we suggest that in future work, more research is done in studying the relationship between (high-confidence) evaluations from AI feedback, and reward hacking of solutions (Nguyen et al., 2015a;

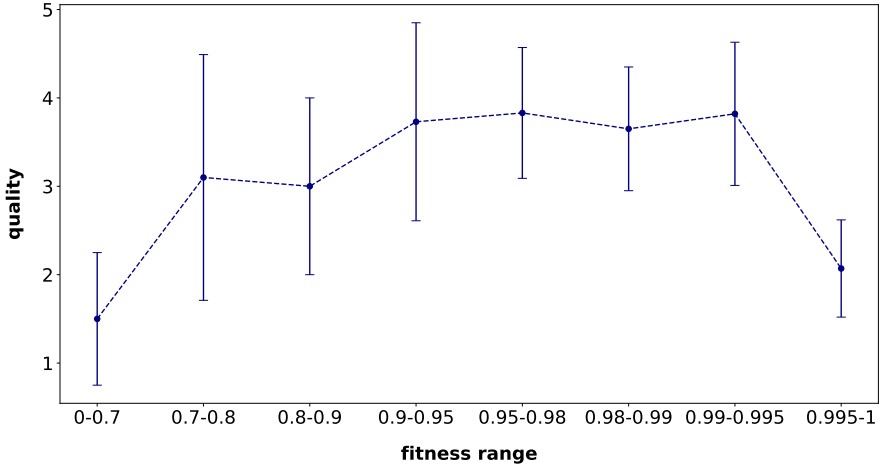

Figure 5: **Correlation plot between quality rating from human annotators, and fitness range (quality computed from AI feedback).** Mean human-annotated quality and statistical error for different ranges of AI feedback fitness scores indicate more frequent instances of reward hacking (Skalse et al., 2022; Lehman et al., 2019) from the outputs of some search methods evaluated in this study.

Skalse et al., 2022; Lehman et al., 2019) under certain conditions and controls during the search (e.g. through the use of seed texts for human-preferred outputs, as shown in Table 1, or additional constraints on the generated solutions during search (Lehman & Stanley, 2010; Brant & Stanley, 2017)).

**Comparison of diversity measurements** For diversity, we observed that the two annotators agreed on the classification of a generated text in the same category 73 percent of the time. On the AI feedback side, diversity is collected on a 20-bin axis, measuring different degrees of correspondence to two experiment-specific categories mentioned earlier. For the purpose of comparison with human feedback, these bins are clustered into a first category (bins 0 to 8), a second category (bins 11 to 19) and a neutral category (bins 9 and 10). Additionally, the 5 samples for each set were collected from a relatively uniform spread of bins from one end of the diversity axis to another (specifically, bins in set [0, 6, 9, 13, 19], except for when the method fails to find a solution for the bin, then the next closest solution near the bin is chosen). This arbitrary arrangement allows for a relatively uniform distribution among the three categories.

On average, AI feedback agrees with a human annotator on the text category label 73% of the time. On samples where both annotators agree on the label of the text, this agreement rate increases to 82%. Within this set, the agreement increases to 95% on samples when we look at samples where both annotators give a label that is not the neutral category. For some samples from bins 6 and 13, one human annotator gave a neutral label and the other annotator one of the other two labels (closer to the AI feedback diversity measure for these bins), indicating that some samples lie between neutral and extreme in the given measure. These findings suggest that diversity classification from AI feedback is relatively reliable, especially in texts where humans agree on the label.

**Baseline quality rating.** The average quality score given by annotators for a given sample was 3.18, close to the middle rating of 3. This gives us an indication of what could be considered the threshold for subjectively good or bad outputs.

The average human (subjective) QD score of all sets in the study is 0.606. This is another indication for the threshold for determining which set from a given run/method had high-quality, diverse texts.

Table 2: **QDAIF outperforms QD with embedding feedback (QDEF) according to human evaluation in human QD score and quality, when the difference is in feedback type for each base method.**. The mean of Human QD score and quality are computed for the set of each method-/run. The differences between the two evaluators for QD score and Quality are shown. Agreement between human and AI feedback on diversity labels given is slightly higher across QDAIF results compared to QDEF results, as well as on agreement between two annotators.

| Method | Human QD score | QD score range | Quality rating | Quality range | Human-AI agreement | Human agreement |
|---|---|---|---|---|---|---|
| QDEF, LMX, Zero-Shot Init | 0.242 | 0.050 | 1.300 | 0.600 | 0.500 | 0.400 |
| QDEF, LMX, Seeded Init | 0.308 | 0.050 | 1.700 | 0.600 | 0.600 | 0.600 |
| QDEF, LMX-Replace, Zero-Shot Init | 0.417 | 0.233 | 2.100 | 1.000 | 0.800 | 0.600 |
| QDEF, LMX-Replace, Seeded Init | 0.500 | 0.200 | 2.300 | 1.000 | 0.800 | 1.000 |
| QDAIF, LMX, Zero-Shot Init | 0.350 | 0.300 | 1.700 | 1.400 | 0.800 | 1.000 |
| QDAIF, LMX, Seeded Init | 0.617 | 0.167 | 3.200 | 1.200 | 1.000 | 1.000 |
| QDAIF, LMX-Replace, Zero-Shot Init | 0.833 | 0.267 | 4.300 | 1.000 | 0.600 | 0.600 |
| QDAIF, LMX-Replace, Seeded Init | 0.739 | 0.078 | 3.700 | 0.600 | 0.700 | 0.800 |

## A.2 COMPARING AI FEEDBACK AGAINST ALTERNATIVE MEASURES OF DIVERSITY

To understand the effect of fuzzy evaluation tools as a component of our QD setup, we tested the use of an alternative method of feedback compared to our default method (AI feedback) in the MAP-Elites pipeline: semantic embedding feedback (Reimers & Gurevych, 2019). For this method, we used a 13B embedding model[1], based on the architecture described in Muennighoff (2022), with an asymmetric search setup to measure the distance between a generated text (document embedding) and a query embedding for a desired measure (e.g. "This is a positive opinion"). To compute a diversity measure that can be defined on an axis, we first get the cosine distances between the document embedding and each of two opposing attribute query embeddings. From this, we can measure how close the document embedding is to one attribute compared to the other, and obtain a single diversity measure normalized in the range [0, 1]. We use the same method for quality feedback, with a query that aims to measure the relevance of generated texts to a specific domain. The cosine similarity is used here as a quality score, with negative values being clipped to 0. Additional setup details are shown in Appendix A.28. Since the subjective quality of the resulting elites (of creative texts) is more informative of the method's potential in further applications for practical synthetic data generation, We conducted a human study in addition to the reporting of QD score stats as part of our results.

We display performance statistics from our runs (QD Score) as well as human evaluation scores on elite samples in Table 2. We observe from our human study that using AI feedback as the evaluator instead of semantic embedding feedback for every variation of the QD run setup potentially leads to subjectively better generations. This is likely due to more prominent reward hacking (Skalse et al., 2022; Lehman et al., 2019) occurring in runs using embedding feedback, where the highest quality score texts end up being very similar to the query "An opinion piece about eating vegetables and plant-based foods", while not optimizing for the subjective quality of texts in different bins. Qualitative analysis of human-evaluated sets of texts from QD with embedding feedback is shown in Tables 25–28. Furthermore, agreement between human and AI feedback on text diversity labels was slightly higher across QDAIF sets compared to QD with embedding feedback (QDEF) sets. Overall, AI feedback outperforms the alternative measure of semantic embedding feedback, by guiding the generation of texts that are more preferred by humans, and by serving as a better evaluator for quality and diversity measures than embedding feedback.

---

[1]`https://aleph-alpha.com/luminous%2Dexplore%2Da%2Dmodel%2Dfor%2Dworld%2Dclass%2Dsemantic%2Drepresentation/`

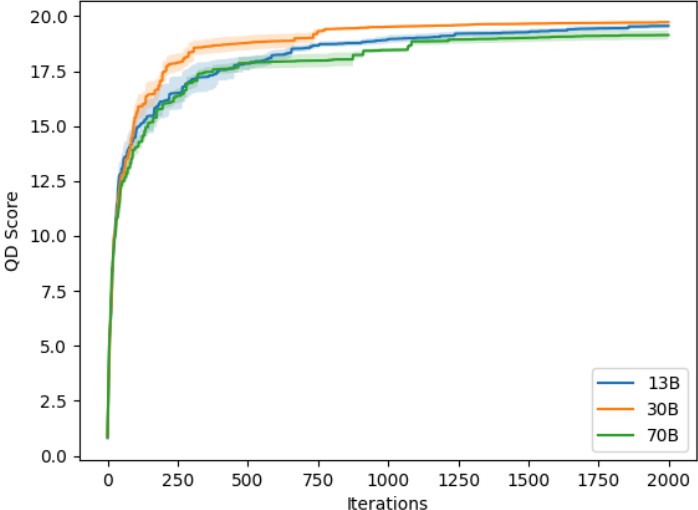

Figure 6: **QD score plots for different LMX model sizes on the Stories - Genre domain**. There is no clear trend in scaling model size with QD score.

### A.3 ON SCALING LMS FOR MUTATION

Previous work has consistently shown that LMs demonstrate improved capabilities in various task-based benchmarks at larger scales (Kaplan et al., 2020; Chowdhery et al., 2022; Chung et al., 2022). This applies to performance in solving tasks through in-context learning, which LMX is based on. Prior work in LMX (Meyerson et al., 2023) has found that a relationship between model scaling and performance of mutations can be observed when evolving binary strings in a search domain. Interestingly, experiments in LMX as well as ELM (Lehman et al., 2022) observed that in some cases, emerging capabilities in mutation ability appear for reasonably small LMs, but may not scale with a clear trend.

Firstly, we show the QD score performance between runs in Figure 6. The standard error in the score across 5 seeds is also shown. We observed that the QD score from the 70B runs converged to a lower point compared to the 13B and 30B runs, which have comparable scores. This highlights a trend between model size and QD score that is not directly proportional. Although suggestive trends are not seen here, a study based on subjective feedback is still necessary for a deeper understanding of the performance of each experiment.

We observed from the human feedback evaluation a trend in the quality ratings, with average quality scores from each experiment set of: 3.43 (13B), 3.60 (30B), and 4.03 (70B). The quality score increases with an increase in model size used, with a higher jump in score for generated stories from the 70B runs.

In terms of the agreement on the genres of evaluated stories between AI feedback and human feedback, we observed the following percentage rates in the following sets: 80.0% (13B), 73.3% (30B), and 73.3% (70B). There is a slight decrease in agreement for texts from the 30B and 70B runs. We found that evaluators differed more frequently in their labels on stories deemed as "neutral" or "romance" according to AI feedback, suggesting that the use of the prompt pool in combination with larger models might lead to generated texts with misaligned AI evaluations on the genre. Still, these agreement rates indicate a good alignment between AI and human feedback.

Figure 7: **QD score Performance between Seeded Init and Zero-Shot**. Performance stats with mean, interquartile mean (IQM) and bootstrapped 95% CI, across 5 random seed runs. Seeded Init is potentially better, but within the CI of Zero-Shot Init. In the Stories - Genre domain, the CI of Zero-Shot Init runs is much wider, indicating significant variation in performance for different random seeds.

## A.4    ON FEW-SHOT AI FEEDBACK PROMPTING

Instruction-following LMs are typically trained to align the model towards generating better answers to zero-shot prompts (Wei et al., 2021; Ouyang et al., 2022). However, few-shot prompting with exemplars was shown to be effective in some aspects with instruction-tuned LMs, especially towards understanding task structure and improving robustness to prompting variations (Wei et al., 2021).

In terms of the average human-evaluated quality, we see a drop in subjective quality for the set of stories from 2-shot AI feedback runs (3.10) in comparison to zero-shot AI feedback runs (3.43). We observed for the other sets that this score increases for the 4-shot set (3.93) and the 8-shot set (4.03). Furthermore, we see that this trend is mostly consistent when we consider the scores for each bin category. This suggests an improvement in QDAIF's ability to discover texts that are perceived to be of higher quality according to human feedback when we use a higher number of in-context examples during AI feedback evaluation.

In terms of the agreement between AI feedback and human feedback, we see a drop in average agreement for ratings that were given to stories in the few-shot feedback experiment sets, with 50.0% (2-shot), 66.7% (4-shot), and 56.7% (8-shot) agreement on sets, compared to 80.0% for zero-shot (default). The level of disagreement occurs more frequently on stories evaluated to have the romance genre. Additionally, evaluators labeled samples from the few-shot sets as "horror" or "neutral" more frequently than "romance", while the proportion of labels given to samples from the zero-shot set was more uniform.

The performance may vary due to the ordering of in-context examples in our few-shot prompts (in Appendix A.29, as shown in Lu et al. (2021). Furthermore, the nature and wording of input-output exemplars/tasks could also influence the performance, in addition to some variation due to the nature of subjective evaluation.

## A.5    ON THE INITIALIZATION METHOD FOR QDAIF

Recent results from LMX demonstrated successful optimization when the search was initialized by evolving a set of pre-existing seed examples (e.g. examples of equations, quotes, text-to-image prompts, code) (Meyerson et al., 2023). At the same time, previous applications of QD methods demonstrated successful search outcomes when using random initialization in different domains, such as in robotics, and latent space illumination of generative models (Cully et al., 2015; Fontaine & Nikolaidis, 2021; Bhatt et al., 2022).

We compared the performance between QDAIF with **Seeded Init** against **Zero-Shot Init** and a baseline method **LMX, Quality-Only** initialized with **Zero-Shot Init**. From Figure 7, we can see that there is potential improvement in QD score when using **Seeded Init** compared to **Zero-Shot Init**, but the difference may not be significant. However, the difference is clear on the lower performance of **LMX, Quality-Only** with **Zero-Shot Init**. This suggests that **Zero-Shot Init** is viable for QDAIF. Still, we need to analyze the effect of the initialization on qualitative samples of elite texts. We also compared the effects of initialization on subjective quality-diversity of texts (see Table 3). Sets of texts discovered by **Seeded Init** within single runs were found to be subjectively higher-quality and more diverse compared to sets from **Zero-Shot Init** (0.772 vs. 0.383 subjective QD score). This suggests potential reward hacking (described for RL problems in Skalse et al. (2022); Lehman et al. (2019) during the search with **Zero-Shot Init** runs, where potentially out-of-distribution texts can

evolve to optimize the AI feedback quality score of solutions, but lead to subjectively low-quality texts across bins.

## A.6  ON THE MECHANISMS OF MUTATION FOR QDAIF

Prior work on ELM highlights the versatility of LMs in applying a variety of potential mutation methods, such as in the form of git diffs, or prompting to directly evolve text/code Lehman et al. (2022). However, studying the potential effects of different LM-based mutation operators on population-level evolvability (focused on future creative potential Lehman et al. (2016)) remains a challenge. Towards understanding the potential of different mutation methods in terms of population-level evolvability of niches, we compare LMX(-Near) (default) (as tested in Meyerson et al. (2023)) against LMX-Replace in terms of resulting generated texts across different runs.

To observe the impact of different mutation methods on the dynamics of the population during search (along with the impact on performance), we implemented and tested an additional method called LMX-Replace. This method evolves few-shot prompts using a larger pool of prompt candidates than that of LMX(-Near) and carries out a slower mutation process by modifying only one few-shot prompt example (instead of all examples) during search iterations. We give a detailed comparison between the methods in Appendix A.26.

We observed an improvement to the resulting generated text sets from runs using LMX-Replace in comparison to LMX for the **Zero-Shot Init** case, according to human evaluation results highlighted in Table 3. We see that the subjective QD score and the quality rating from human evaluators were higher on average across the tested domains (and improvements within domain-by-domain comparisons also, highlighted in Table 4). Given that it may not be desirable to constrain the search with **LMX-Near w/ Seeded Init**, LMX-Replace can act as an alternative mutation method to steer the dynamics of population search towards subjectively improved outputs.

The introduction of an archive depth was used for the creation and maintenance of a prompt pool for LMX-Replace (cf. Appendix A.27). Future works could explore the usage of depth in MAP-Elites archives for domains where uncertainty (or subjectivity) influences the evaluation of solutions, as was previously studied in Flageat & Cully (2020; 2023).

Table 3: Mean stats, human eval scores, QDAIF methods. LMX-Replace outperforms LMX when using Zero-Shot Init.

| Method | QD score | Quality | H-AI agreement | H agreement |
|---|---|---|---|---|
| LMX w/ Zero-Shot Init | 0.383 | 2.233 | 0.600 | 0.600 |
| LMX-Replace w/ Zero-Shot Init | 0.540 | 2.800 | 0.700 | 0.800 |
| LMX w/ Seeded Init | 0.772 | 3.900 | 0.833 | 0.800 |
| LMX-Replace w/ Seeded Init | 0.763 | 3.800 | 0.800 | 0.800 |

## A.7 ON COVERAGE AND BEST QUALITY SOLUTIONS ACROSS DOMAINS

Figure 8 shows performance plots measuring the coverage of the archive (i.e. how many bins in the search space are filled with at least one solution). Figure 9 shows performance plots measuring the quality score of the best solution found across the whole search (existing in one of the defined bins).

QDAIF often discovers the best overall solutions during search that have higher quality scores than the best overall solutions found from other methods. This is likely enabled by the goal-switching mechanism of QD approaches described in Mouret & Clune (2015); Gaier et al. (2019), which is more relevant when the QD search is at a later stage and begins to improve the already existing solutions (elites) across the bins through the mutation of distant solutions. Overall, QDAIF can jointly optimize the quality and diversity of solutions, outperforming other methods in most comparisons, and highlighting contributions to successful gains in QD score.

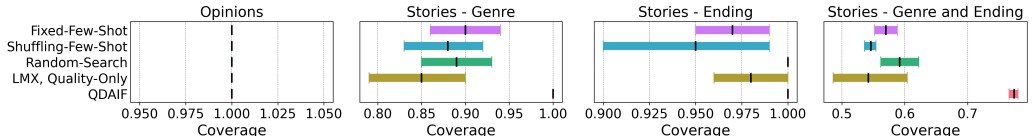

Figure 8: **QDAIF significantly outperforms baselines in the percentage of available bins covered by solutions in all domains**. Coverage stats with mean bootstrapped 95% CI, across 5 random seed runs. The maximum possible coverage is 1. In some cases, methods achieve maximum coverage across all runs. The advantage of QD approaches is shown in their ability to discover more solutions that cover a wider search space, in addition to finding high-quality solutions (cf. Figure 9).

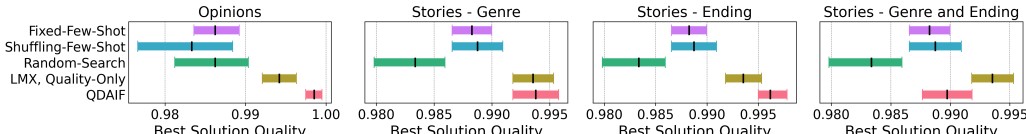

Figure 9: **QDAIF and LMX, Quality-Only both optimize for the quality of the best solution found overall**. Best solution quality stats with mean bootstrapped 95% CI, across 5 random seed runs. The maximum possible quality score is 1. QDAIF is more often able to achieve a higher score than **LMX, Quality-Only** in **Opinions** and **Stories - Ending**, while also outperforming in terms of coverage (cf. Figure 8). In the 2D Stories - Genre and Ending domain, **LMX, Quality-Only** finds a solution with higher best quality more often than QDAIF, while still lagging in coverage. It is likely that QDAIF can find the best solutions with higher maximum quality more often in some domains because of the maintenance of diverse elite solutions that enable more possibilities to maximize the objective function, especially when the search begins to focus on refining existing solutions instead of covering unfilled bins (Gaier et al., 2019). To save on compute, we reuse the generated (iterations of) texts from the baselines to compute independent stats comparisons as they do not use diversity measures during a search, unlike QDAIF which depends also on the defined diversity axes to conduct a successful search. We can compute independent QD score stats for baselines using the same 5 reruns, but in the **Stories** domains, the best solution quality remains the same.

A.8   ON COMPARISONS BETWEEN QDAIF AND DIVERSITY-SEEKING BASELINES

We compared QDAIF against explicit diversity-seeking baseline methods. One baseline, **LMX, ROUGE-L**, is based on n-gram-based filtering with ROUGE-L similarity (Lin, 2004), as done in Self-Instruct Wang et al. (2022a). They followed a similar approach of maintaining a few-shot prompt pool to generate diverse solutions (with diversity maintained by adding generated texts to a prompt pool if different enough to other existing solutions based on n-gram matching), in their case, for the generation of diverse instruction prompts towards the creation of a diverse, high-quality synthetic instruction tuning dataset. Another baseline is based on Novelty Search (NS) (Lehman & Stanley, 2011a), a diversity-seeking algorithm that was proven to be more sample efficient than single-objective approaches (the diversity-only equivalent of **LMX, Quality-Only**, which only optimizes for solution quality) in discovering desired solutions to problems with many local optima, and also studied in prior works introducing the QD illumination problem (i.e. improving QD score (Pugh et al., 2016)). Unlike **LMX, ROUGE-L**, NS aims to encourage diversity within an arbitrarily defined space of diversity. To make NS comparable with QDAIF, we introduce Novelty Search through AI Feedback (NSAIF) as the method that seeks diversity in domains where diversity is defined by AI feedback axes. We denote this baseline as **LMX, NSAIF**. As with all methods, these baselines are initialized with the prompt pool specified in Appendix A.21. We describe the baseline implementations in more detail in Appendix A.9, including variants of these baselines with quality AI feedback (QAIF) filters as a minimal criterion for diversity search (Lehman & Stanley, 2010).

**Results.** Figure 10 shows QD score performance plots comparing QDAIF against diversity-seeking baselines, baselines that also aim for quality and diversity in solutions, and other baselines. QDAIF consistently outperforms diversity-seeking baselines across domains in addition to other baselines not focused on improving desired diversity in solutions. Yet, **LMX, NSAIF** tends to outperform other baselines that are not focused on improving solution diversity. On the **Stories - Genre and Ending** domain, **LMX, NSAIF** significantly outperforms many methods, even **LMX, Quality-Only**, which carries out single objective quality optimization (and successfully does so compared to other baselines, by maximizing the best solution quality alongside QDAIF, cf. Figure 12). This indicates that maintaining diversity in solutions is important for QD score performance in such domains (and improves search space coverage, cf. Figure 11). Furthermore, maintaining desired diversity, such as qualitative measures of diversity, is necessary for a method to improve QD score during search. **LMX, ROUGE-L** achieves low QD scores, and is within range of being the worst method for improving best solution quality (cf. Figure 12), in spite of the higher coverage achieved across domains compared to several baselines (cf. Figure 11). However, the performance for this baseline improves significantly with the introduction of the quality filter via QAIF. This indicates that maintaining high-quality solutions (not just diversity) is important. Still, adding the quality filter on top of **LMX, NSAIF** only helped more often compared to the default variant of NSAIF on the **Stories - Genre and Ending** domain, and shows no improvement on other domains. QDAIF (based on MAP-Elites) reliably improves QD score in tested domains and does so with higher sample efficiency (cf. Figure 13 and Figure 14). Results comparing QDAIF to additional diversity-seeking baselines (including ones with quality filters) highlight performance gains from our proposed QDAIF method, as well as the importance of seeking both quality and diversity.

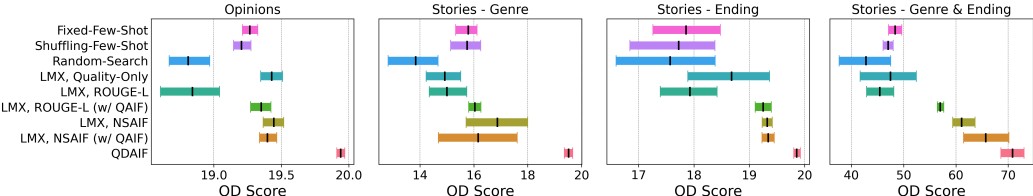

Figure 10: **QDAIF outperforms (diversity-seeking) baselines across domains in QD score and maintains high-performance across all domains**. Performance stats with mean bootstrapped 95% CI, across 5 random seed runs. The maximum possible QD score is 20 (100 for 2D archive (4th plot)). The best-performing baseline in the **Stories - Genre and Ending** domain, **LMX, NSAIF (w/ QAIF)**, does not consistently outperform other baselines in other domains, and has wide performance variability in two domains. Compared to **LMX, Quality-Only**, all the diversity-seeking baselines using NSAIF or ROUGE-L filtering (except **LMX, ROUGE-L**) significantly outperform this baseline approach that optimizes for a single quality score objective.

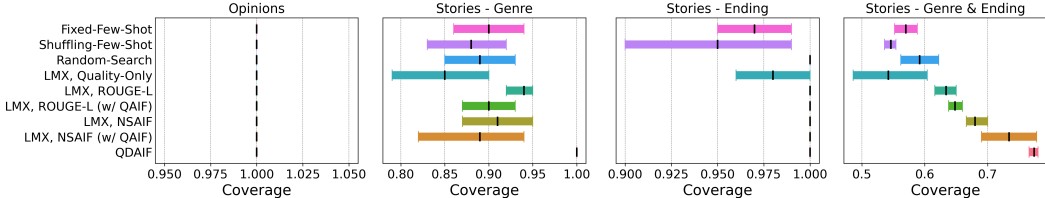

Figure 11: **QDAIF more often outperforms (diversity-seeking) baselines in the percentage of available bins covered by solutions in all domains, such as Stories - Genre, and Stories - Genre and Ending**. Coverage stats with mean bootstrapped 95% CI, across 5 random seed runs. The maximum possible coverage is 1. In some cases, methods achieve maximum coverage across all runs. Consistent high-coverage performance from QDAIF compared to the next best baselines in **Stories - Genre and Ending** (**LMX, NSAIF** and **LMX, NSAIF (w/ QAIF)**) suggests that evolving from high-quality solutions is important to improving solution diversity, in addition to diversity-seeking search (where **LMX, ROUGE-L** also achieves higher coverage more often compared to non-diversity-seeking baselines.

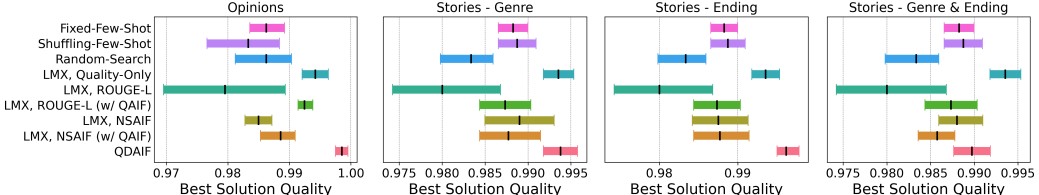

Figure 12: **QDAIF and LMX, Quality-Only both more often optimize for the quality of the best solution found overall**. Best solution quality stats with mean bootstrapped 95% CI, across 5 random seed runs. The maximum possible quality score is 1. The addition of quality filters to **LMX, ROUGE-L (w/ QAIF)** and **LMX, NSAIF (w/ QAIF)** is ineffective for significantly improving the best solution quality across runs compared to QDAIF. Still, **LMX, ROUGE-L (w/ QAIF)** is better compared to **LMX, ROUGE-L** at finding higher-quality best solutions.

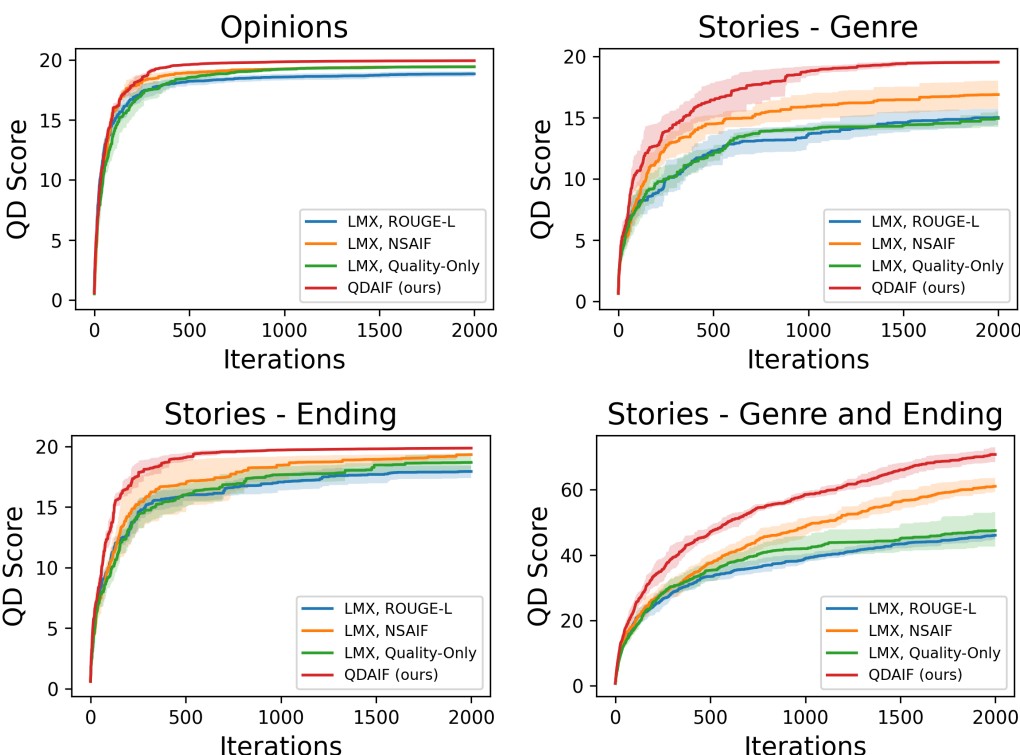

Figure 13: **QDAIF outperforms (diversity-seeking) baselines across domains in sample efficiency of QD score improvement**. Line plot stats with mean bootstrapped 95% CI, across 5 random seed runs. The maximum possible QD score is 20 (100 for 2D archive (4th plot)). Additionally, **LMX, NSAIF** succeeds in obtaining higher sample efficiency more often than **LMX, Quality-Only**, the single quality objective optimization method, without any optimization of the quality score objective.

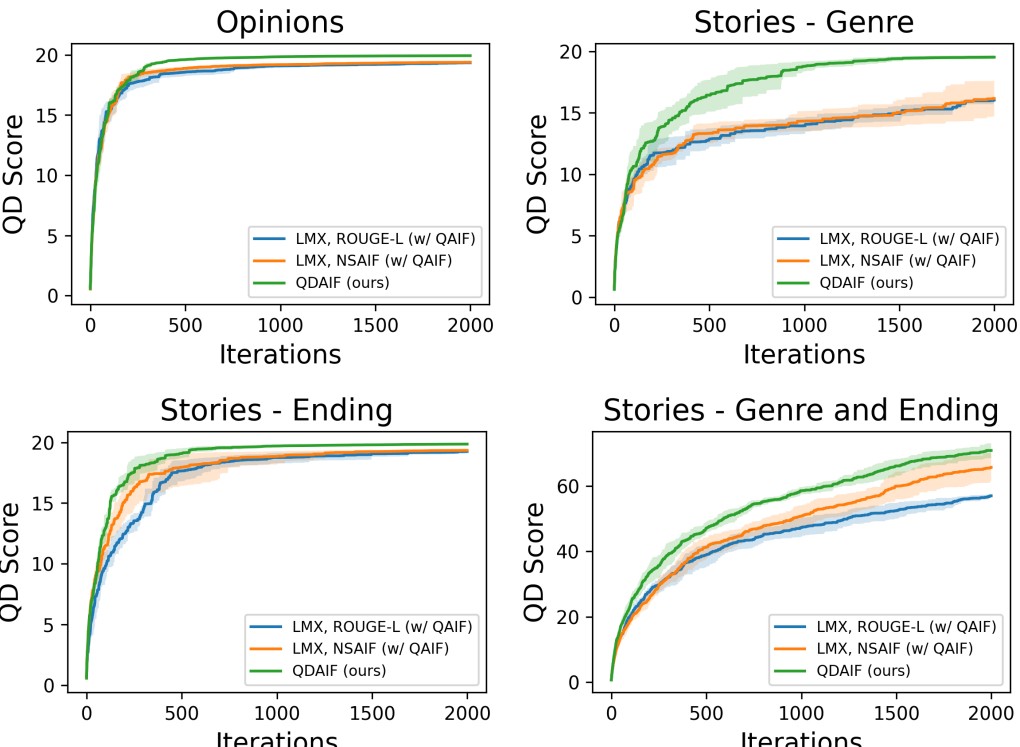

Figure 14: **QDAIF outperforms (diversity-seeking, with quality control) baselines across domains in sample efficiency of QD score improvement**. Line plots with mean bootstrapped 95% CI, across 5 random seed runs. The maximum possible QD score is 20 (100 for 2D archive (4th plot)). The improvement of QDAIF compared to other approaches aiming for diverse, high-quality solutions is significant in the **Stories - Genre** domain. In the **Stories - Genre and Ending** domain, **LMX, NSAIF (w/ QAIF)** outperforms **LMX, ROUGE-L (w/ QAIF)** and is within the performance range of QDAIF in later iterations, but with lower sample efficiency in early iterations, and wider performance variability.

## A.9 Setup for Diversity-Seeking Baselines

**LMX, ROUGE-L**. Following Wang et al. (2022a), we set the threshold for the maximum allowed ROUGE-L similarity between generated texts and all texts in the prompt pool at any given time (from which few-shot examples are sampled for LMX generation) to 0.7, without limits on the prompt pool size. If the similarity exceeds this, the generated text is still logged for QD score evaluation, but rejected from being added to the prompt pool. Otherwise, the text is added to the prompt pool so that it can be used for LMX evolution during later samplings of few-shot prompts. The rest of the setup is comparable to QDAIF except for the MAP-Elites archive maintained by QDAIF.

**LMX, NSAIF**. In line with the high-level implementation of NS described in Lehman et al. (2008), we implemented the novelty (diversity-based) measure in the same way that QDAIF defines diversity measures - as the diversity attribute defined by the AI feedback axis (or axes) in the range [0, 1], with an additional Euclidean distance-based measure between the diversity attribute of the generated text, and existing texts in the prompt pool. Novelty is computed by measuring the mean distance between the diversity attribute values of the generated text, and their $k$ nearest neighbors in the prompt pool (with respect to the diversity attributes of the neighbor texts). We keep $k$ to be 15, as was tested in the original NS implementation. Similar to NS, the novelty score is compared to a novelty threshold (initially 0.05 in all **LMX, NSAIF** runs, the distance width between adjacent bin ticks for a unit range with 20 uniform bins); if the novelty of the generated text is higher than the threshold, it gets accepted into the prompt pool for evolution with LMX, otherwise, it is logged for evaluation of performance results but rejected from the prompt pool. We apply a dynamic adjustment of the novelty threshold, following NS, and define it so that the threshold is multiplied by 1.05 (increased) if 3 solutions in a row were accepted to the prompt pool within any iterations window, or multiplied by 0.95 (decreased) if 21 solutions in a row were rejected from the prompt pool during the search.

In consideration of the non-linear calibration of AI feedback models in evaluating texts (cf. Appendix A.31), we apply a piecewise linear transformation to the original diversity attribute value from AI feedback evaluation of diversity (that lies along an axis discretized by non-uniform binning) so that it lies instead along an axis discretized by uniform bins while preserving the number of bin intervals. For example, in our 20-bin setting, the input value 0.9975, which lies between bin ticks 1, and 0.995, would be transformed to an output value of 0.975, and the input value 0.35 (between bin ticks 0.5 and 0.2) would be transformed to an output value of 0.475. This enables novelty to be computed using Euclidean distances while preserving the nature of AI feedback model non-linear calibration in distinguishing subjectively similar solutions from diverse solutions. Additionally, this keeps the definition of diversity consistent and fair with the setup of diversity measures for evaluation (in QDAIF iterations, and for baseline performance comparisons). The number of bin intervals defined for the piecewise linear transformation is the same as the number of intervals set as default across the **Opinions** and **Stories** domains (20 for 1D archives, 10 for 2D archives).

**Quality AI Feedback Filtering.** We implemented and assessed variants of the above baselines, **LMX, ROUGE-L (w/ QAIF)**, and **LMX, NSAIF (w/ QAIF)**, by adding a simple quality filter for accepting generated texts to prompt pools, based on quality AI feedback defined in the respective creative writing domains. This involves an additional step in each baseline method immediately after the diversity criteria assessment of solutions, where generated texts must also have quality score values above a minimum threshold before being added to the prompt pool. Thus, these variants become similar to QDAIF methods, where they aim for diverse, high-quality solutions. These baselines differ from our default QDAIF method with MAP-Elites in that the minimum quality thresholds are not a function of the quality of best solutions across individual bins, but are arbitrarily defined, to constrain the prompt pool to satisfy a minimal criterion for quality, as was introduced previously for diversity-seeking methods in Lehman & Stanley (2010). This would make the quality improvement process of solutions across all bins less sample-efficient. We define this simple quality threshold for the baselines to be fixed at 0.8, closer to the upper bound of the full quality range [0, 1]. This was determined based on the intuitions of the results from the human evaluation study we conducted (cf. Figure 5 in Appendix A.1), where a quality value of 0.8 from AI feedback corresponds to a generally high human feedback Likert (quality) score. As done in **LMX, Quality-Only**, these baselines (with quality filtering) limit the size of the prompt pool to be up to 100.

## A.10 ON AUTOMATICALLY EXPANDING ARCHIVE DIMENSIONS

Prior work in QD, as described in Section 2.2, often relies on diversity measures that are designed and manually defined at the start of the search. One existing approach for more automatic QD search without supervised (defined) measures of diversity is to use unsupervised learning to represent diversity without relying on ground truth measures (Cully & Demiris, 2018; Cully, 2019; Grillotti & Cully, 2021; Wang et al., 2023b; Ding et al., 2023). Such unsupervised approaches do not embody any complex prior of what humans find interesting about diversity; an alternative approach would be to query capable LMs about what dimensions of diversity are interesting or important for a particular domain. In this way, semantically-rich axes of diversity could be automatically generated (and could then be evaluated automatically as well through other LM calls, as in QDAIF).

Given the advances in foundation model capabilities, we could reasonably prompt LMs (such as GPT-4 (OpenAI, 2023)) to come up with new axes of diversity in a more automated pipeline potentially while search is running - by giving it a description of the user's search problem, and existing diversity axis being searched through (i.e. the existing AI feedback diversity prompt(s)), we could ask the LM to give us a different, previously unexplored diversity axis and define a new AI feedback prompt that can be added to the MAP-Elites evaluation. For example, in the **Poetry** domain, we could ask the LM to generate multiple diverse aspects of poetry (e.g. "Genre" and "Tone" as studied in the presented experiments), and also come up with diverse categories defining this search space for QDAIF to search through with MAP-Elites. The effectiveness of this kind of approach has not been studied thoroughly in prior works, especially on the question of whether or not expanding the dimensions of diversity during the search can meaningfully improve diversity in resulting solutions towards increasingly broader definitions of diversity.

**Setup.** We took a step in the direction of automating the definition of diversity axes, where we analyzed the effectiveness of creative search for solutions when we expand the dimensions of diversity axes during an intermediate iteration of an existing search, with performance measured in terms of improving the QD score (for a given ground truth, higher-dimensional search space). We tested this approach in the **Stories - Genre and Ending** domain, compared the performance between variations in methods with QD score (out of 100) as done in other experiments, and compared the following setups: 2D Archive Search (apply QDAIF with the full 2D diversity axes defined), 1D Archive Search (apply QDAIF with only a 1D diversity axis defined, done for both Genre, and Ending diversity archives each), and expanding 1D to 2D Archive Search (starting QDAIF search with either the Genre or Ending diversity axis defined, and then expanding the search with the addition of a second diversity axis, or the other different axis to the one defined at the start of the 1D search). For the setups with expanding diversity axes, the second archive dimension is introduced from iteration 1000 of the search, out of 2000 iterations total. We also assessed the performance in the case where instead of adding an extra dimension with a new diversity axis during search, QDAIF transitions from an initial 1D archive (e.g. for Ending), to a different 1D archive (e.g. for Genre). In our experiments, the transition is carried out from iteration 1000 out of 2000 iterations.

**Findings.** Figure 15 shows performance plot results for all the settings described in this section, and Figure 16 shows QD score line plots to visualize sample efficiency differences between the different setups of automatically expanding dimensions, especially after iteration 1000. We found that for all cases tested of expanding from 1D to 2D archives, improvements in QD score for a higher dimensional archive are significant when compared to searching with only a 1D diversity axis and evaluating the resulting solutions with both AI feedback diversity measures. Furthermore, we found that it's possible to approach the performance in QD score through this diversity axes expansion when compared to the QD score achieved from a full 2D Archive Search. This level of improvement hints at the potential of applying the approach of prompting LMs to generate diversity measures for more autonomous, creative search, and highlights the value of scaling up this approach in future work.

Expansion of diversity axes is promising as an approach to improve the quality and diversity of solutions without the need to initialize a high-dimensional archive from the beginning (in the case where only manual setting of diversity axes is done). Vassiliades et al. (2016) found in experiments through CVT-MAP-Elites (a method to define bins based on uniformly spaced centroids in very high-dimensional spaces of diversity, solving the compute requirements of standard MAP-Elites in this high-dimensional case) that standard MAP-Elites is impeded in its ability to further improve the fitness (quality) of solutions in non-empty bins as the cases where the search fills empty bins

occurs much more frequently when more bins are created due to the increase in dimensionality. The quality of existing non-empty bins normally requires several iterations of improvement before reaching more optimal quality scores for the given bin or niche. We show that this is also the case in Figure 15, in the third plot on the value of best solution quality; the quality of the best overall solution at the end of the search is more often higher for the 1D to 2D expansion settings, compared to the setting where the 2D archive is initialized from scratch at the start of the search. It is also the case that searching only in the 1D archives is more often better (with higher best solution quality) compared to the outcomes when conducting any search in the higher dimensional archive. Expansion of dimensions seems to deliver a good trade-off of slightly decreased best solution quality for significant improvements in solution diversity. This balance is quite useful to find for improving the sample efficiency of the search, given that Mouret & Clune (2015) found the discovery and maintenance of both diverse and high-quality solutions to be important for enabling the ongoing search to improve the quality and diversity of newly generated solutions even more quickly.

Additionally, we studied performance comparisons for the case of transitioning different 1D diversity axes (as a different approach to automatic search in different dimensions of diversity compared to the expanding dimensions setup). Figure 15 also shows performance in this case, and Figure 17 shows QD score line plots, with differences in performance visibly shown after iteration 1000. In one case (1D (Ending) to 1D (Genre)), a significant improvement in QD score is visible for this archive transition case when compared to searching only in one (initialized) diversity axis throughout the whole search. This setup also approaches the performance of searching in the higher dimensional 2D archive. In the other case evaluated (1D (Genre) to 1D (Ending)), no notable improvements were seen compared to just conducting QDAIF in the single 1D (Genre) archive. Even though results here show that the performance of QDAIF in the transitioning 1D archives case is sensitive to the diversity axes searched through (and the order in which the transitions occur), this highlights another promising approach to automatically adjusting diversity axes given AI feedback prompts to be generated by LMs along with the expanding dimensions setup. This is especially the case when we want to scale up QDAIF to search through an even higher number of diversity axes automatically, where we can lower the computational requirements of searching in lower dimensions (meaning lower number of total bins created) and also mitigate the presented challenges of conducting MAP-Elites search iterations in very high-dimensional archives (in the previous paragraph), while maintaining promising improvements to performance that would be seen in having QDAIF explore a growing number of different diversity axes. Yet, the importance of the diversity axes chosen in further encouraging novelty is clear; prior work studying the theoretical properties of diversity-seeking algorithms showed through empirical experiments that the choice in diversity measures also needs to be aligned with the intended task in mind, for solutions of interest to cover the search space (Doncieux et al., 2019, Figure 6). Future research can explore the potential of designing a curriculum for QDAIF to search through different diversity axes as an open challenge to improving the quality and diversity of creative texts with potentially an unbounded number of subjective dimensions of diversity, depending on individual personal perspectives.

Overall, this method of automatic expansion and adaptation of diversity axes introduces a new way of balancing the trade-off between improving quality or improving diversity in solutions, especially in settings where the dimensions of desired diversity, while sometimes not obvious to the user until later realization, do not reach the hundreds as would be in the case of problems studied in CVT-MAP-Elites. In the large-scale higher-dimensional case, adaptive transitioning of diversity axes is another promising direction. Such approaches to adaptations of diversity axes draw inspiration from work on learnable behavior characteristics (Meyerson et al., 2016) and meta-diversity (Etcheverry et al., 2021). Given prior findings on the adaptation of the archive search space to encourage more interesting solutions (worth collecting) to emerge during search, and the ability of LMs to provide feedback on unbounded aspects of solution diversity, perhaps new opportunities have emerged in improving future systems inspired by QDAIF. We can leverage this finding for future work to more confidently apply the aid of LMs in automatically generating diversity axes for QDAIF search.

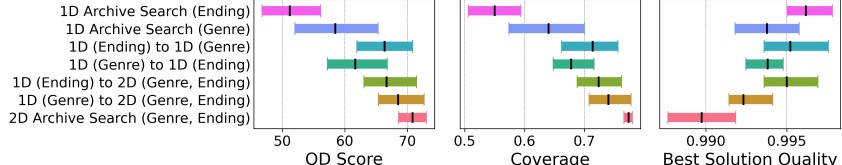

Figure 15: **Expanding (or adapting) archive dimensions for QDAIF during search enables improved search in higher dimensional search spaces compared to searching for solutions along a single 1D diversity axis**. QD score, coverage, and best solution quality stats with mean bootstrapped 95% CI, across 5 random seed runs. The maximum possible QD score is 100. In both cases of dimension expansion (expanding from 1D (Genre) and 1D (Ending) archives to 2D) enables the performance to approach that of conducting QDAIF search in the full 2D archive, with significant improvement in coverage compared to searching purely in their respective 1D archives. For one case of adaptive archive transition for QDAIF (1D (Ending) to 1D (Genre)), QD score performance and coverage significantly improve compared to searching only in the 1D (Ending) archive.

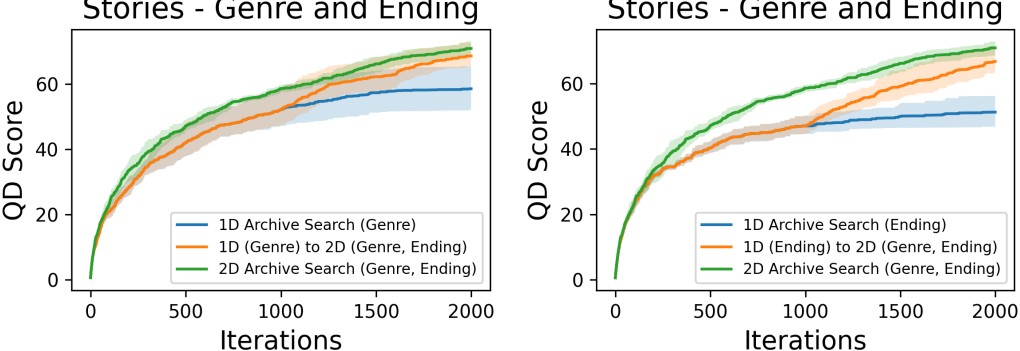

Figure 16: **QD score sample efficiency improves from the midway iteration step of orange line plots when the archive dimensions are expanded automatically, in comparison to doing search only in the 1D archive during QDAIF**. Line plots with mean bootstrapped 95% CI, across 5 random seed runs. The maximum possible QD score is 100. In both cases, either starting with the 1D (Genre) archive (left) or the 1D (Ending) archive (right) and expanding to the 2D archive improves QD score sample efficiency with respect to the 2D archive definition, and approaches performance of the green lines (searching in the 2D archive). The blue lines (1D archive only) converge to a lower QD score during later iterations.

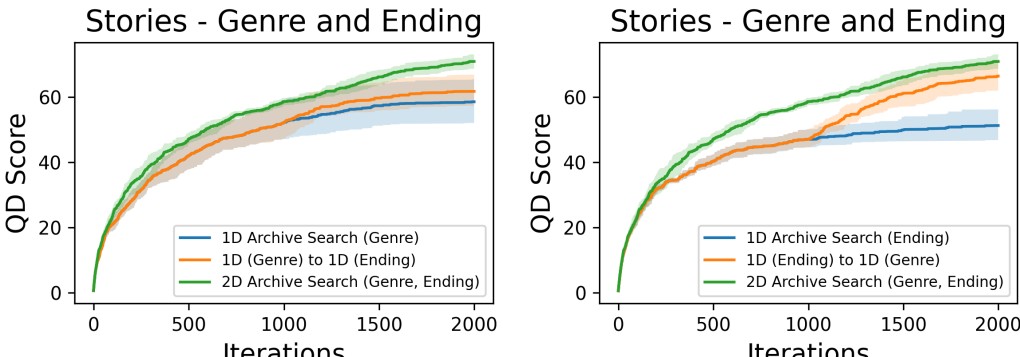

Figure 17: **QD score sample efficiency improves in one case (right plot) from the midway iteration step of orange line plots when the archive diversity axis is changed to a new diversity measure, but not noticeably improved in another case (left plot), in comparison to doing search without transitioning to a different diversity measure during QDAIF**. Line plots with mean bootstrapped 95% CI, across 5 random seed runs. The maximum possible QD score is 100. The successful improvement in QD score sample efficiency happens in the case of transitioning from the 1D (Ending) archive to the 1D (Genre) archive, approaching the performance of doing QDAIF in the higher dimensional 2D archive, as shown in the right plot. The case shown in the left plot is still below the performance of QDAIF conducted in the 2D archive, with the orange line converging early to a lower score. The blue lines (1D archive only) converge to a lower QD score during later iterations.

### A.11 ON FINETUNING MUTATION MODELS

Prior work on ELM found that using a domain-specific finetuned git diff LM during code generation search led to runs with higher QD score performance compared to when using a pre-trained git diff LM (Lehman et al., 2022). The finetuning step relied on a dataset of filtered solutions across several previous runs, adding complexity to the process of creating an effective model for evolving text. To simplify the process, We investigated finetuning on sampled solutions within the same run, during a search that begins with solution generation from a pre-trained LM. We explore the impact of finetuning as a mechanism that can potentially encourage exploitation (through learning to generate higher-quality score solutions) during the search while using evolution to encourage exploration.

To collect samples for finetuning during the search, the first step is collecting a dataset for a given state of the archive, sampling up to 10 solutions with the highest quality score from each bin. Archive bin depth was used to keep track of these finetuning samples (cf. Appendix A.27, on bin depth used), and introduced in a different application of MAP-Elites for uncertain environments in Flageat & Cully (2020; 2023). The dataset is shuffled, and then training samples are batched for the LM to finetune on during this phase. Each training sample consists of the prompt that was originally used by a solution to generate an output, and the generated completion text as target tokens for finetuning. We tested both full-model finetuning, and a parameter-efficient finetuning method with sequential adapters finetuning, as described in He et al. (2021). We ran each method using QDAIF LMX(-Near) w/ Seeded-Init (including default method runs without finetuning) on the Stories (Genre and Ending) domain, extending the number of archive dimensions to two. For experiments with adapters, we also conduct runs where the adapter layers are initialized but without doing finetuning, to enable comparisons with adapter finetuning runs accounting for differences in LM architecture introduced by additional layers during the search for runs with finetuning. Still, near-identity initialization (as described in Houlsby et al. (2019)) preserves the general performance of the LM.

**Finetune-Once.** This method runs a single finetuning phase during the search before resuming generation with the finetuned LM. We vary two parameters during experiments with Finetune-Once: the iteration step to start the phase (Start), and the number of finetuning steps during this phase (Steps).

**Generate-Finetune.** This method extends Finetune-Once by carrying out the finetuning phase multiple times during the search, repeating the steps of dataset collection and finetuning at regular intervals. We add an additional variable parameter, controlling the regular interval frequency for every set number of generation iterations (Frequency).

**Observations.** We compared the QD score performance between methods across 5 fixed random seed runs. In general, the performance of the Finetune-Once methods with higher scores is comparable to default runs, though slightly lower (but mostly within the confidence interval). For Finetune-Only runs, we observed a potential decrease in performance when using full-model finetuning compared to adapter finetuning, in comparison to conducting these runs without finetuning, with an average $-3.09$ difference in QD score for full-model finetuning compared to $-0.33$ difference for adapter finetuning. It is possible that overfitting on solution examples is more likely for full-model finetuning, which would lead to a decrease in the ability of the LM to evolve new solutions to be accepted in the archive. When we used adapter finetuning as the default for other experiments, we observed a potential decrease in performance and increased variance in performance across multiple seeds when compared to the default method without finetuning. The increase in variance also indicates potential improvements to the search with Finetune-Only in some cases, but further studies are needed on the behavior of search when finetuning on different archive states and constraints to samples collected for finetuning. We can observe a clearer impact of the adverse effects of finetuning from our Generate-Finetune runs. Considering the 95% confidence interval, the highest mean QD score run we tested from the Generate-Finetune group obtained a performance score of $78.74 \pm 5.10$ compared to $84.09 \pm 1.91$ from the default generate-only method. With most runs with Generate-Finetune converging to a lower QD score earlier in the search, it is likely that running standard scheduling the finetuning phase multiple times leads to overfitting of the LM to the point where the model fails to generate new solutions to add to the archive.

## A.12 HUMAN EVALUATION STUDY STATS IN WRITING DOMAINS

We collected the annotation results from our study, to show the performance of each search method on the tested writing domains, displayed in Table 4.

Table 4: **Extended human eval stats for different creative writing domains, baselines and QDAIF.** The QD score presented here is based on subjective human evaluation results, normalized to a possible range of [1/15, 1]. In terms of the observed difficulty of uncovering the search spaces for each domain from mean quality and QD score stats within each domain, **Opinions** is the easiest domain (mean QD score is 0.702, mean quality score is 3.613), followed by **Stories - Ending** (mean QD score is 0.648, mean quality score is 3.575), and **Stories - Genre** (mean QD score is 0.591, mean quality score is 3.013). The range of metrics between two human evaluators is computed to highlight the variance in some contexts due to more complex aspects of subjectivity in evaluating certain texts.

| Method | Human QD score | QD score range | Quality rating | Quality range | Human-AI agreement | Human agreement |
|---|---|---|---|---|---|---|
| Opinions | | | | | | |
| Fixed-Few-Shot | 1.000 | 0.000 | 5.000 | 0.000 | 0.700 | 0.800 |
| Shuffling-Few-Shot | 0.728 | 0.056 | 3.600 | 0.800 | 0.700 | 0.800 |
| Random-Search | 0.600 | 0.267 | 3.600 | 0.800 | 0.700 | 0.800 |
| LMX, Quality-Only | 0.750 | 0.300 | 3.800 | 1.200 | 0.600 | 1.000 |
| QDAIF (LMX w/ Zero-Shot Init) | 0.350 | 0.300 | 1.700 | 1.400 | 0.800 | 1.000 |
| QDAIF (LMX w/ Seeded Init) | 0.617 | 0.167 | 3.200 | 1.200 | 1.000 | 1.000 |
| QDAIF (LMX-Replace w/ Zero-Shot Init) | 0.833 | 0.267 | 4.300 | 1.000 | 0.600 | 0.600 |
| QDAIF (LMX-Replace w/ Seeded Init) | 0.739 | 0.078 | 3.700 | 0.600 | 0.700 | 0.800 |
| Stories (Genre) | | | | | | |
| Fixed-Few-Shot | 0.650 | 0.167 | 3.400 | 1.600 | 1.000 | 1.000 |
| Shuffling-Few-Shot | 0.861 | 0.056 | 4.300 | 0.600 | 0.700 | 0.800 |
| Random-Search | 0.683 | 0.100 | 3.400 | 0.800 | 0.700 | 0.400 |
| LMX, Quality-Only | 0.600 | 0.267 | 2.900 | 1.400 | 0.700 | 0.800 |
| QDAIF (LMX w/ Zero-Shot Init) | 0.200 | 0.000 | 1.400 | 0.400 | 0.400 | 0.600 |
| QDAIF (LMX w/ Seeded Init) | 0.767 | 0.133 | 3.700 | 0.600 | 0.800 | 1.000 |
| QDAIF (LMX-Replace w/ Zero-Shot Init) | 0.286 | 0.006 | 1.500 | 0.200 | 0.500 | 0.800 |
| QDAIF (LMX-Replace w/ Seeded Init) | 0.683 | 0.100 | 3.500 | 0.600 | 0.900 | 0.800 |
| Stories (Ending) | | | | | | |
| Fixed-Few-Shot | 0.650 | 0.167 | 4.000 | 0.400 | 0.700 | 0.800 |
| Shuffling-Few-Shot | 0.500 | 0.200 | 2.600 | 0.800 | 0.700 | 0.400 |
| Random-Search | 0.533 | 0.000 | 2.900 | 1.400 | 0.800 | 0.600 |
| LMX, Quality-Only | 0.600 | 0.400 | 3.900 | 0.600 | 0.600 | 0.400 |
| QDAIF (LMX w/ Zero-Shot Init) | 0.600 | 0.200 | 3.600 | 0.800 | 0.600 | 0.200 |
| QDAIF (LMX w/ Seeded Init) | 0.933 | 0.133 | 4.800 | 0.400 | 0.700 | 0.400 |
| QDAIF (LMX-Replace w/ Zero-Shot Init) | 0.500 | 0.333 | 2.600 | 1.600 | 1.000 | 1.000 |
| QDAIF (LMX-Replace w/ Seeded Init) | 0.867 | 0.267 | 4.200 | 1.600 | 0.800 | 0.800 |

### A.13 ON THE DIFFERENCES IN ELITE TEXTS ACROSS DOMAINS FROM QDAIF

We referred to LMX-Near (Meyerson et al., 2023) as LMX in the main text for brevity. For more detailed analysis with different mutation operators, we refer to the full name. A summary of human eval stats based on the samples described below can be found in Appendix A.12.

**Opinions.** We report the results from evaluated sets of generated texts from our human study, starting with the Opinions domain in Appendix Tables 29–32 (with qualitative descriptions of the evaluated generated texts in the table captions). Tables are organized in the following order, for each domain: LMX-Near /w Seeded Init (default); LMX-Near /w Zero-Shot Init; LMX-Replace /w Seeded Init; and LMX-Replace /w Zero-Shot Init. Qualitatively, we can see that repetition of phrases appears more often in the samples of generated elite texts from runs using LMX-Near, especially when using Zero-Shot Init. For LMX-Near /w Seeded Init (In Table 29), there is frequent repetition of phrases like "I would rather eat scrambled eggs" (third row, bin 9), and "At a restaurant" (fifth row, bin 19). For LMX-Near /w Zero-Shot Init (In Table 30), there is further repetition of undesired phrases like "Below is a random opinion piece" and "Here is a random opinion piece" in all examples. On the other hand, the generated texts from LMX-Replace (Tables 31 and 32 for Seeded Init (default), and Zero-Shot Init methods respectively) lack this output artifact, and received higher quality scores from human feedback overall. Furthermore, texts from LMX-Replace /w Zero-Shot Init received the highest quality score (from humans) compared to the other three methods here, while texts from LMX-Near /w Zero-Shot Init received the lowest quality scores.

**Stories - Genre.** For this domain (with evaluated sets presented in Tables 33–36), evaluators found the subjective quality of texts from runs using Seeded Init (default) to be higher than runs using Zero-Shot Init. The stories that were generated with Zero-Shot Init were found to be low in quality due to the presence of attributes such as erroneous titles, and a text style that fails to reflect what is expected in a plausible short story text. Furthermore, the elite stories from the Zero-Shot Init generated sets were more likely to lead to disagreements on the genre between AI feedback and human feedback, with neutral labels given more frequently even for texts in some of the extreme ends of the bins. Given a more open-ended generation task, and a narrow space of desired diversity (focused on the spectrum between two genres), LMX-Near /w Seeded Init (default) (Table 33) was the most successful method in finding a story for the horror genre niche in Bin 0 (with a low quality score of 2 from evaluators). Other methods either failed to discover a story for this niche, or received an even lower quality score from human feedback.

**Stories - Ending.** For this domain (with evaluated sets presented in Tables 37–40), the stories received higher quality scores across sets in comparison to the sets from the **Stories - Genre** domain, indicating a potentially easier search space when finding stories with different kinds of endings. Still, methods using Seeded Init produced sets of stories that received higher quality scores from evaluators, in comparison to sets from Zero-Shot Init (especially LMX-Replace /w Zero-Shot Init in Table 40, where the presence of erroneous titles led to lower subjective quality). In spite of the lack of guidance from hand-written prompt examples during initialization, LMX-Near /w Zero-Shot Init (Table 38) managed to produce reasonable stories of above-average quality to cover different ending niches.

### A.14 ON THE EVOLUTION OF GENERATED TEXTS OVER ITERATIONS FROM QDAIF

We describe the evolution of texts from QDAIF with LMX at different search iterations in Tables 41–52. One key factor that influences the search is the initialization method - generations in early iterations from Zero-Shot Init methods frequently contain elements that are subjectively different to hand-written examples in the seed texts (in A.23). For example, erroneous URLs (in Tables 41, 43, 51) and titles (in Tables 45, 49) are seen when the possible distribution of outputs is not constrained by seeded in-context examples. This kind of method enables further exploration of output samples, potentially useful in the search of interesting, diverse creative texts. At the same time, more constraints are required from quality assessment using AI feedback in order to control the evolving population of creative texts, towards high-quality, diverse texts. In several cases, outputs from these methods in later iterations show reduced instances of these artifacts compared to early iteration outputs (e.g. in Table 51), but can be missed when AI feedback evaluation at times is misaligned with human preferences (e.g. in Table 45). Still, using Seeded Init does not guarantee enough guidance to completely remove undesired features from generations in later iterations. For example, texts with repetitive phrases can be seen during later iterations on the Opinions domain from methods using Seeded Init (see Tables 42 and 44. In general, Seeded Init runs are more likely to lead to high-quality texts across niches, in comparison to Zero-Shot Init runs, especially according to human evaluation (see Table 1), especially for more challenging, open-ended domains such as story-writing. Furthermore, a side-effect of the increased likelihood of subjective reward hacking when using Zero-Shot Init for runs. Still, QDAIF can still work well with Zero-Shot Init, especially in combination with LMX-Replace, as shown in the results from the Opinions domain experiments.

### A.15 ON THE DIFFERENCES IN ELITE TEXTS ACROSS DOMAINS FROM BASELINE METHODS

Evaluated sets are presented in Tables 13–16, Tables 17–20, and Tables 21–24 for Opinions, Stories - Genre, and Stories - Ending domains respectively. **Fixed-Few-Shot** and **Shuffling-Few-Shot** consistently adhere to the style and structure of the seeded examples. **Random-Search** exhibits more variability, with discrepancies in feedback between humans and AI most evident in the Opinions domains and challenges in story consistency and character inclusion. **LMX, Quality-Only** further highlights these discrepancies, especially where repetitive or contradictory opinions are concerned. Across narratives, **LMX, Quality-Only** tends to miss character details or produce underdeveloped storylines, despite sometimes receiving high AI feedback. A recurring theme across all baselines is the differential perception of quality between human and AI feedback, with repetitive narratives, character relevance, and development being central points of contention.

### A.16 ON THE EVOLUTION OF GENERATED TEXTS OVER ITERATIONS FROM BASELINE METHODS

Qualitatively, both **Fixed-Few-Shot** and **Shuffling-Few-Shot** consistently replicate concepts and expressions from the few-shot examples, at times directly copying entire segments (Tables 53, 57, 54, and 58). This suggests a potential over-reliance on the few-shot examples, hindering the generation of diverse solutions. In contrast, **Random-Search** aims to foster diversity by retaining all entries into the pool. However, this approach may unintentionally impede optimization for higher fitness solutions. Notably, fitness values of generated entries using **Random-Search** in later iterations persistently fall short (Tables 55 and 59). The crux of this issue seems to arise from the strategy's indiscriminate inclusion of entries, even those with low fitness. Consequently, when these low-fitness examples are integrated into few-shot prompts, they influence the generation of subsequent entries, often resulting in similarly suboptimal outcomes. Lastly, **LMX, Quality-Only** introduces a fitness-centric approach, retaining only those solutions with the highest fitness. While entries from its early iterations display a diverse range of phenotype and fitness values, later iterations, with a noticeable uptick in fitness scores, tend to use similar phrases and writing styles (Tables 56 and 60). This suggests that stringently prioritizing high-fitness solutions may prevent the exploration of more varied or potentially better solutions.

## A.17    On the Use of Different Versions and Types of Models for Poetry

We show QD score line plots comparing QDAIF against other methods in Figure 18. We noted that in addition to improvement in QD score from QDAIF, CI gaps from experiments using GPT-4 were qualitatively wider as shown in the line plots. This suggests the potential of using different models for prompted variation of solutions in creative domains, with the possibility that certain models are more capable of generating slightly more diverse solutions (that may better cover the space of possible solutions) (cf. Appendix A.18 on differences in rewriting behavior from different models). Although QDAIF significantly improves on relevant non-QD methods through the evolution of solutions via rewriting, the challenge remains in uncovering all possible solutions in the diversity space of interest (e.g. empty bins in experiment archive, or categories that can further differentiate subjectively unique solutions); it is difficult for existing models without explicit guidance on the desired poem categories to know that these exist in the space of possibilities, in a similar sense that the average person may not know what is the most distinct variation that can be applied to rewriting poems (from the wide space of possible poetry). This issue manifests even in simple domains such as random number generation (Renda et al., 2023). Coverage is possible with methods that guide LMs on the desired (cf. Figure 19), with the added benefits of grounding on poems of interest through seed parent poems (cf. Appendix A.18), but future work is needed to create systems that can successfully navigate in solution exploration beyond what is currently known or defined (i.e. through limited genre and tone categories) during a search at hand (Zhang et al., 2023), perhaps even through the discovery of new types of poems, unlike ones that have been written by human poets.

Figure 20 compares the performance of different models used for generation and rewriting with each search method. We found higher CI gaps from runs using GPT-4 with **Random-Poems**, **Fixed Seed Rewrite**, and QDAIF (LMX-rewrite), in comparison to GPT-3.5-Turbo. The difference in rewriting behavior between GPT-4 and GPT-3.5-Turbo may contribute to variation in performance predictability during QDAIF (cf. Appendix A.18). Additionally, GPT-4 more often gives higher QD score performance in some cases (**Random-Poems**), while GPT-3.5-Turbo can improve sample efficiency during search in other cases (**Fixed Seed Rewrite**, **Targeted-Poems**). Interestingly, improvements in sample efficiency from GPT-3.5-Turbo may not translate fully to support interesting discovery of new solutions in poetry during QDAIF (cf. Figure 23). Results further highlight the need to investigate the potential of different models for solution variation in QDAIF.

We noted from runs in the Poetry domain using an older version of GPT-4 (April 2023) for the experiments that QDAIF saw potential improvement in performance compared to baseline methods. There is a risk of variation in performance and search behavior due to the nature of changing model versions; the impact would be on the generation, re-writing, and evaluation of poems. We observed in Figure 21 that QDAIF successfully populates the archive with high-quality poems, leaving only three bins empty. In contrast, **Random-Poems** (Baseline) only fills the archive sparsely, demonstrating that asking for creative output from a single LM prompt often results in a lack of diversity. Furthermore, more bins remain empty in the archive generated by **Targeted-Poems** (Targeted Baseline). Overall, QDAIF achieved higher QD scores in experiments with older models and covers more bins in the archive than **Random-Poems** and **Targeted-Poems** as can be seen Figure 21. This hints at another potential benefit of QDAIF with LMX-guided, where a combination of instructed guidance and evolution could also lead in improvements to QD score with some models.

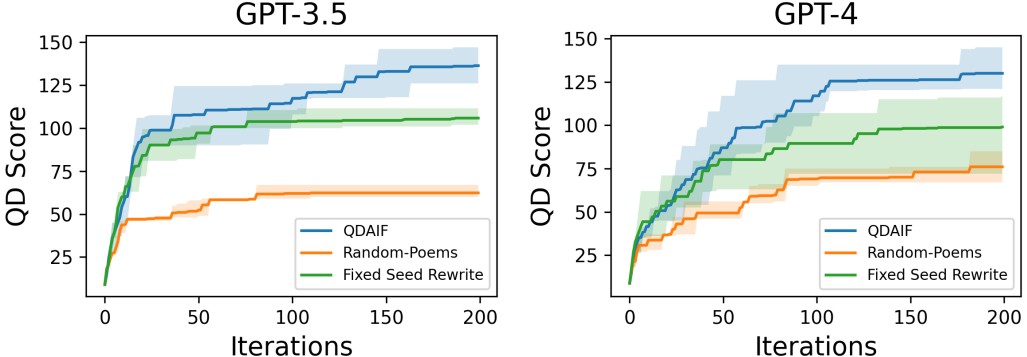

Figure 18: **QDAIF (LMX-rewrite) outperforms the baseline, Random-Poems, and an ablation, Fixed Seed Rewrite, in terms of QD score, across all generation models tested.** 95% CI of mean is shown for stats across 3 random reruns. We observed from the Mann-Whitney U Test that QDAIF achieves greater QD score than other methods ($p \leq 0.05$). The step of rewriting a poem (**Fixed Seed Rewrite**) is shown to improve QD score more often than simply requesting poems (**Random-Poems**), while the additional step of evolutionary search (QDAIF) enables additional improvement. With GPT-3.5(-Turbo), non-QDAIF methods tend to have narrower CIs, indicating higher predictability of reduced performance.

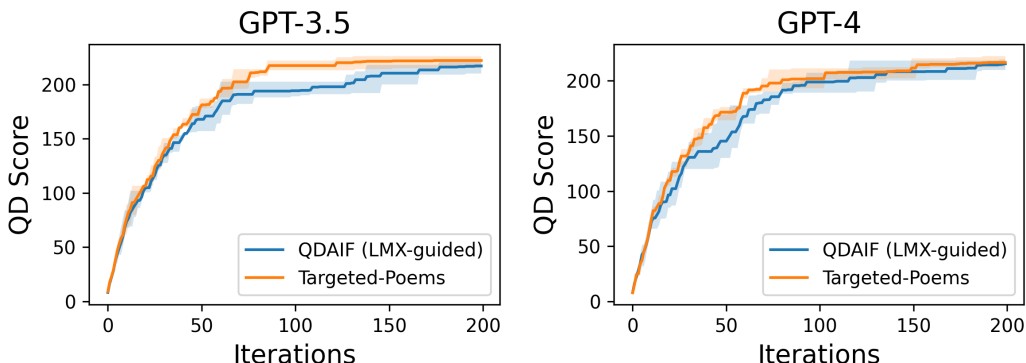

Figure 19: **QDAIF (LMX-guided) is on par with Targeted-Poems, while preserving the benefits of diverse evolution of poems stemming from a seed poem (cf. Appendix A.18).** 95% CI of mean is shown for stats across 3 random reruns. All methods incorporate guidance on the desired genres and tones in poems during search, and models used are capable of covering all the designated archive bins with relevant diverse, high-quality poems. All methods can quite easily cover the 25 bins of the archive within the number of iterations.

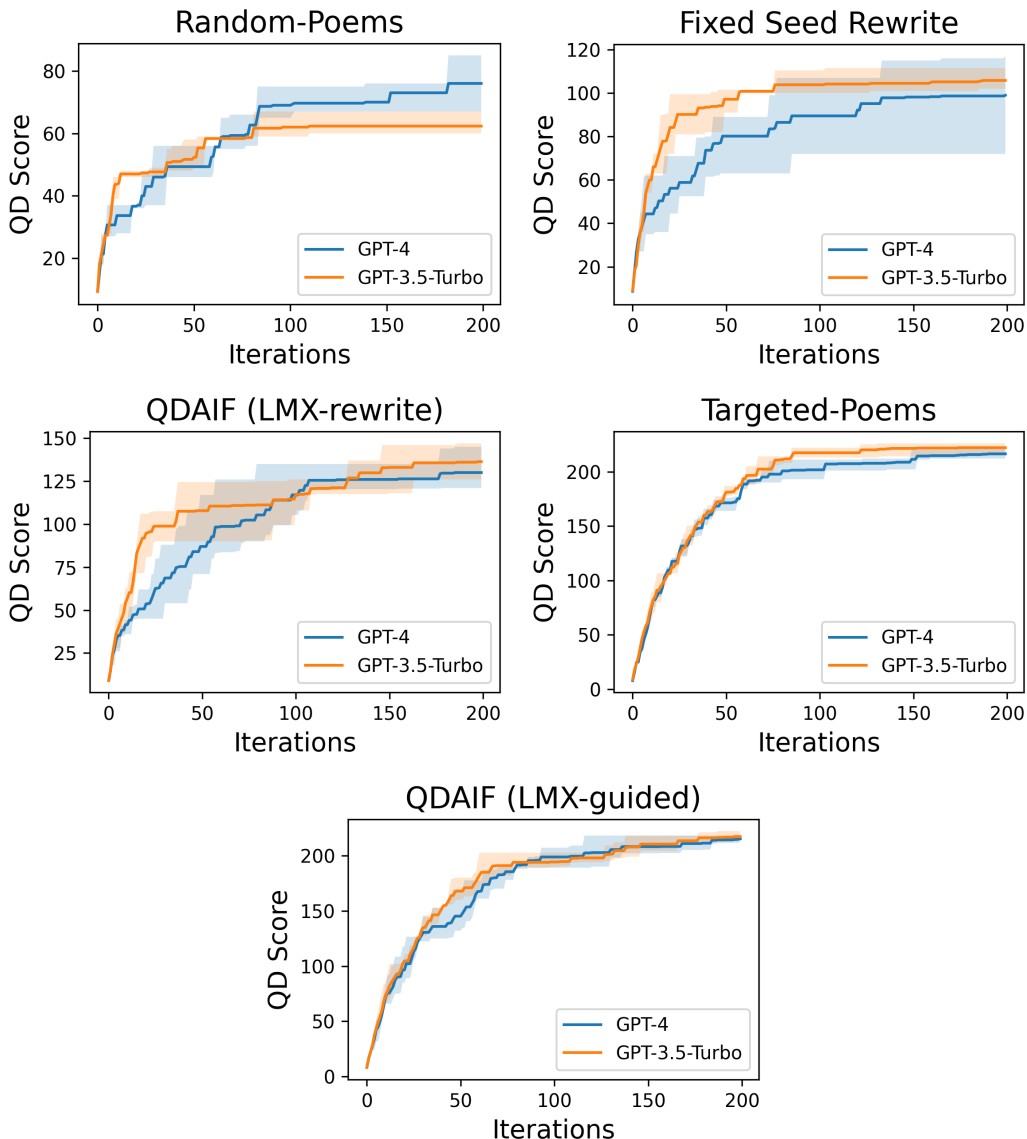

Figure 20: QD score performance (with 95% CI from 3 random reruns) for each method varying the model used for generation/rewriting. The evaluation model is GPT-4 (OpenAI, 2023) for all runs here. In some cases, GPT-4 more often achieves a higher score (e.g. with **Random-Poems**), while in other cases, GPT-3.5-Turbo achieves improved sample efficiency over iterations of search (e.g. with **Fixed Seed Rewrite**, **Targeted-Poems**). Wider CIs in some runs with GPT-4 indicate higher unpredictability in performance, and potentially a wider scope in possible outputs that may or may not lead to higher score.

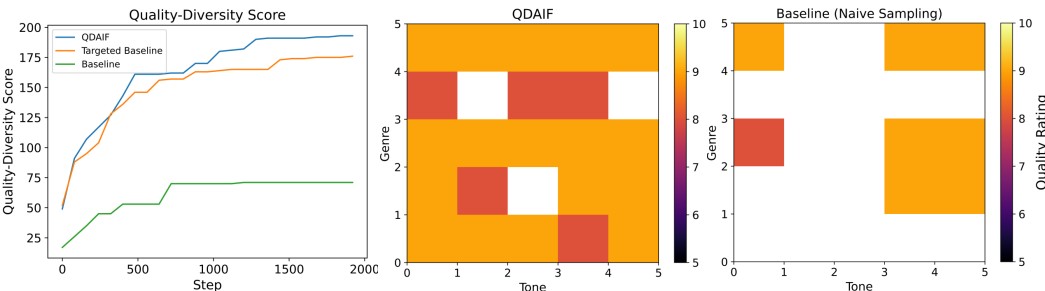

Figure 21: **QDAIF (blue, middle) covers more the space of diverse poetry with high-quality solutions compared to baselines.** When compared against a targeted mutation baseline (**Targeted-Poems**, orange), QDAIF attains higher QD scores throughout evolution, and covers a significantly larger portion of the search space than a fully random baseline (**Random-Poems**, green, right). This is for runs with an older version of GPT-4.

### A.18 ON THE EVOLUTION OF POEMS THROUGH INSPIRED REWRITING

Instead of generating poems completely from scratch, users may generally prefer to have models re-write their draft poems. Figure 22 shows an example of the capabilities of GPT-4 using LMX-guided, by starting off from the seed poem and continuing a chain of evolution in high-quality poems across different genres and tones. The evolving poetry chain shown in the figure is the longest continuous one discovered during the search, where the most repeated rewrites occurred starting from the seed poem. Rewrites with the model qualitatively gave meaningful variations in poems that transfer connective imagery with twists in rhythm and connotations, depending on the target genres and tones. At the same time, We found that even when GPT-4 is specifically prompted to craft a poem of a particular genre and tone, subsequent evaluation using the same LM does not always deem it as having the same targeted genre or tone. This underscores a widely recognized gap between text generation and discrimination (Saunders et al., 2022, Page 12). This is clear from several of the poems classified as hymns, but containing multiple 5-line verses similar to the style of a limerick (while not generating a typical single-verse limerick). It was likely references to religious imagery in later verses, as well as the multi-verse structure that influenced the evaluation for the closest genre.

Figure 23 shows another example of a chain from using GPT-3.5-Turbo. We found qualitatively that using GPT-4 led to more interesting variations of new poems during rewriting in comparison to GPT-3.5-Turbo rewrites. This is more clear from the repetition of "In fields of emerald, a gentle sway" at the start of each poem following the mysterious sonnet.

Future research is needed to study the behavior of chained rewriting over stepping stones with current foundation models (Nguyen et al., 2015b; 2016). Secretan et al. (2011) found that from user studies in Picbreeder, a tool for evolving images through human-in-the-loop, accumulating divergent chains of solutions of growing complexity from different users (carrying out the evolution of diverse images) is important for a search that is focused on discovering meaningfully interesting and diverse artifacts. Gaier et al. (2019) validated the potential of QD approaches in enabling the discovery of intermediate solutions that overcome the challenge of escaping undesired local minima (missing promising trajectories to desired solutions) faced by objective-based search (Stanley & Lehman, 2015). By leveraging goal-switching (Mouret & Clune, 2015), QD search was able to maintain enough diversity in the population to enable the discovery of solutions that appear more like targets of interest, significantly outperforming a single objective optimization approach (which generated solutions that resembled primitive patterns unlike the targets) (Gaier, 2020, Chapter 4). This property of QD search is especially important in solution spaces where even slight perturbations in solutions can lead to significant (and often unexpected) qualitative changes in solution properties. This can apply to representations from Compositional Pattern Producing Networks (CPPNs) (Stanley, 2007), and text representations themselves, where minor changes to certain parts of text can lead to significant changes in passage tone, imagery, or even functionality of code (Lehman et al., 2022). Properties of chained divergence as demonstrated in the Poetry domain could inspire further directions in designing new systems with QDAIF.

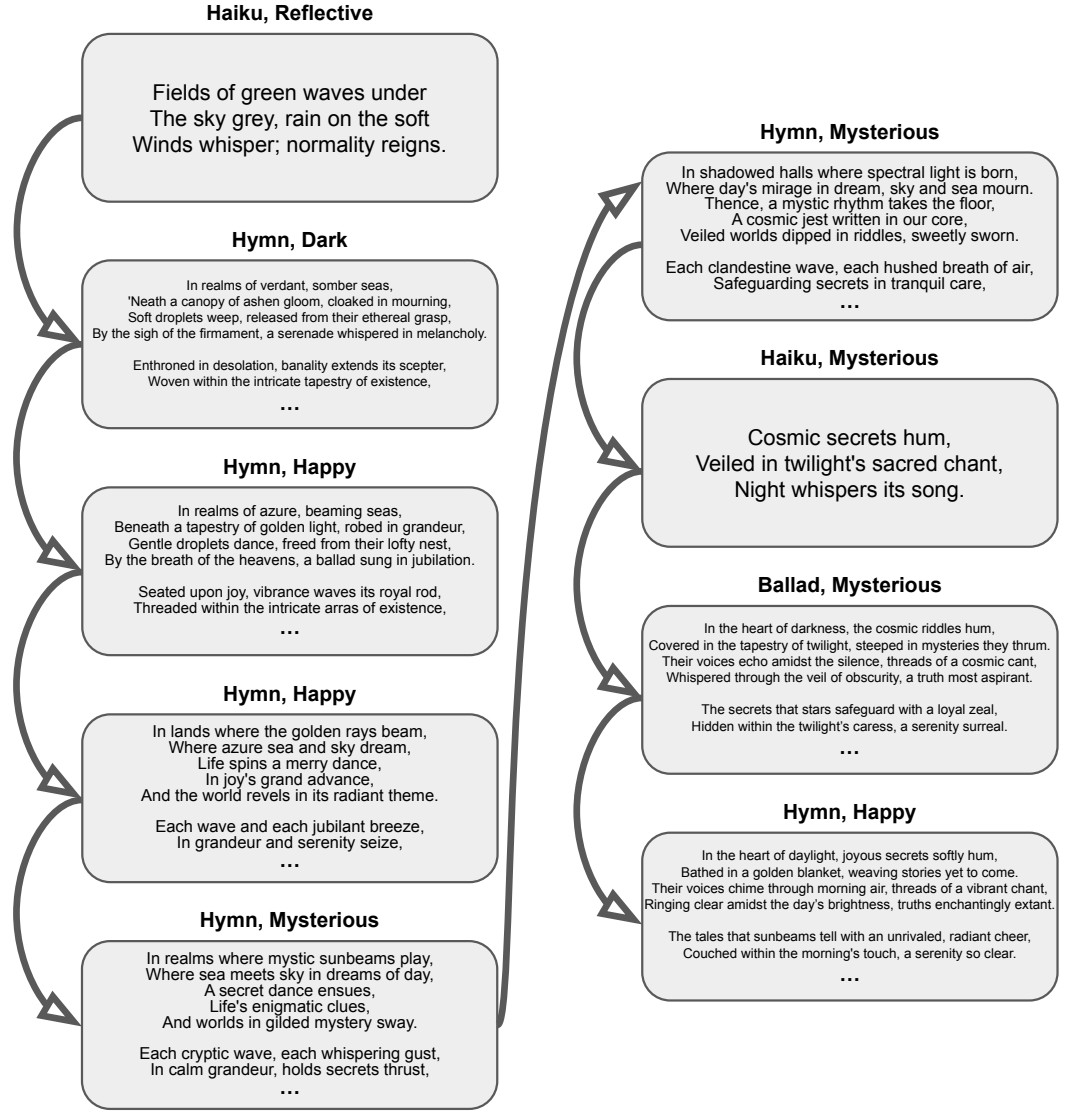

Figure 22: **QDAIF is enabled by guided poem rewriting. With GPT-4 (OpenAI, 2023).** Starting with a reflective haiku seed poem, the reference to "fields of green waves" and "sky grey, rain" inspires the next dark hymn poem with "verdant, somber seas" and "canopy of ashen gloom", followed by further chains of inspired rewriting. The imagery and choice of words evolve with varying outputs as poems of different genres and tones are generated. We noted that not all poems had evaluated genres and tones that matched their target ones during prompted rewriting, for example, with the hymns with multiple 5-line verses being a result of limericks as the target.

## Haiku, Reflective

Fields of green waves under
The sky grey, rain on the soft
Winds whisper; normality reigns.

## Haiku, Reflective

Emerald fields sway,
Whispering tales of the rain,
Peaceful serenade.

## Sonnet, Mysterious

In mystic emerald fields that gently sway,
Whispering secrets of the long-lost rain,
A peaceful serenade lulls souls astray,
As twilight casts its enchanting refrain.

The verdant tapestry reveals its lore,
In whispers carried on the zephyr's sigh,
Each blade of grass sings of a world before,
Where magic thrived beneath a moonlit sky.
...

## Sonnet, Romantic

In fields of emerald, a gentle sway,
A melody of secrets whispers low,
As twilight's enchanting refrain holds sway,
A peaceful serenade to souls bestowed.

This verdant tapestry of ancient lore,
Whispers carried upon a zephyr's sigh,
Each blade of grass sings tales from days of yore,
Where magic thrived beneath the moonlit sky.
. . .

## Ballad, Dark

In fields of emerald, a gentle sway,
A melody of secrets whispers low,
As twilight's enchanting refrain holds sway,
A peaceful serenade to souls bestowed.

This verdant tapestry, now cloaked in gloom,
Echoes of despair upon the wind,
Each blade of grass now sings a mournful tune,
Where shadows dance, a sorrowful hymn begins.
. . .

## Ballad, Dark

"In fields of emerald, a gentle sway,
Where secrets whispered low, love's melody,
Twilight's enchanting refrain holds sway,
A peaceful serenade to souls bestowed.

Cloaked in gloom, this verdant tapestry,
Echoes of despair upon the wind's flight,
Each blade of grass sings mournful tune, entranced,
Where shadows dance, a sorrowful hymn ignites.
. . .

## Ballad, Dark

"In fields of emerald, a gentle sway,
Where secrets whispered low, love's melody,
Twilight's enchanting refrain holds sway,
A peaceful serenade to souls bestowed.

Cloaked in gloom, this verdant tapestry,
Echoes of despair upon the wind's flight,
Each blade of grass sings mournful tune, entranced,
Where shadows dance, a sorrowful hymn ignites.
. . .

Figure 23: **QDAIF is enabled by guided poem rewriting. With GPT-3.5-Turbo.** The evolved poems cover different genres and tones following inspiration from the seed poem. However, there is repetition of "In fields of emerald, a gentle sway" at the start of each poem following the mysterious sonnet. GPT-4 improves on the ability to rewrite, suggesting that more capable models are also able to give more interesting variations to creative texts.

A.19    EXAMPLES FROM THE POETRY DOMAIN

Figures 24, 25, and 26 show some generated poems of different quality, genres, and tones. Noticeably, hymns, limericks, ballads, and sonnets that are rates as higher quality tend to be longer, and more closely align with the defining characteristics of their respective genres. For example, the hymn evaluated at 9/10 quality has phrases like "celestial delight", "under heaven's vault", and "seek divine in the ordinary's course", showcasing a worshipper's perspective towards divinity, a trait more pronounced than in an 8/10 rated hymn (Figure 24). The sonnet with a 9/10 quality rating demonstrates a more consistent rhyme scheme, where all the end words in the same quatrain rhyme, compared to the one rated as 7/10 quality (Figure 26).

**Hymn, Mysterious, 8/10 quality**

Within firmament's shroud, mystery cloaked,
Emerald fields of our mortality joked.
Sky's cryptic canvas does portray,
The secret ballet of life's fray.

In twilight's hymn, we find our peace,
Through whispered winds, our hearts release.
Yet in the hush, wonder resides,
Where normality's secret tide subsides.

**Hymn, Mysterious, 9/10 quality**

In shrouds of celestial delight,
Where emerald waves dance day and night,
Normality, in its grand design,
Is laced with mystery, rich and fine.

Under heaven's vault so keen and high,
Kissed by dew from the grieving sky,
Within hushed whispers, truths interwoven,
Sings a tale of the life we've known.

Upon this canvas of the firmament,
An endless ballet of colors is sent,
In the drizzle's hymn, secrets unfold,
Life's sacred pact in whispers told.

Ode to twilight's alluring enigma,
In each drop, a stirring stigma.
Seek divine in the ordinary's course,
As cosmic whispers take their force.

In the meadows where green waves quiver,
Rain reigns with a cryptic shiver,
Unmasking life's mysteriously sweet strains,
In the quiet humdrum, wonder remains.

**Limerick, Reflective, 7/10 quality**

Within life's grand and starlit sphere,
Are joys that often disappear.
Yet dreams, when forged with pure delight,
Cut the darkness like a flight.
Reflections shine with naught to fear.

**Limerick, Reflective, 9/10 quality**

There once were some fields, so astoundingly bright,
Under heavens grand shroud, starlit at night.
With the wind's playful tease,
Through the whispering trees,
Came tales of the ordinary, wrapped in moonlight.

Each blade in the verdant sea, a dancing sprite,
Swayed in harmonious ballet, a divine sight.
Their jubilant thrum,
A celestial hum,
Brought the vast universe into the moonlight.

As rain-drops told riddles on soft petal suite,
Each secret unfurled in their landing so sweet.
An ode to simplicity,
Of ordinary's divinity,
A cosmic secret in every heart-beat.

'Neath the twilight's glow, we'd discover and delight,
In the beauty of normality, in plain sight.
Where the fields' emerald grain,
Whispered sweetly in the rain,
An undying melody, in the quiet of the night.

In the end, isn't it a spectacular feat?
The everyday wonder that refuses defeat.
In the whispers of the breeze,
Through the dancing leaves,
Lies a world of beauty, extraordinarily neat.

Figure 24: Examples of different quality mysterious hymns and reflective limericks generated and evaluated using GPT-4.

**Ballad, Happy, 8/10 quality**

Upon a field so green and bright,
Underneath the soft moonlight.
The winds they whisper, skies unbound,
But life continues, to joy we're crowned.

The tempest roars, yet we remain,
And dance within the gentle rain.
Strong spirits thrive in sun and storm,
A testament to hearts so warm.

Nature's force, both soft and grim,
The dance of life's a joyful hymn.
No matter if the skies are glum,
In life's dance, to joy we strum.

**Haiku, Dark, 7/10 quality**

Beneath raven wings,
Silence screams in shadow's grip,
Moon weeps blood for dawn.

**Haiku, Dark, 9/10 quality**

Sea's emerald heart sighs,
Grief-veiled tales whisper in wind,
Dark silence speaks pain.

**Ballad, Happy, 9/10 quality**

Upon a quilt of emerald sway, neath heaven's cloak of softest grey,
There lies a tale to brighten day, a ballad wrapped in hues of May.
A simple scene, untouched, pure, its tranquility a hallowed cure,
A pastoral dream in verses lure, where nature's peace will ever endure.

Fields of vivid, endless green, a scene unspoiled, so serene,
Each blade a whispered tale unseen, each sway, a rhythm in nature's routine.
Underneath a charcoal sky, where the cloud-kissed tears do lie,
A scenery's sigh, a soothing lullaby, a story spread 'neath the grey and high.

Oh! How the rain does paint the air, the softest melodies declare,
Each drop a note, none to compare, a symphony suspended there.
Upon the whispered winds it rides, in its graceful dance, our heart confides,
In unseen hands, our fate it guides, under the sky where serenity resides.

For 'tis not sorrow these verses claim, nor a tear-streaked dirge of wintry fame,
Instead, a ballad, joy's pure flame, set in nature's modest frame.
Beneath the grey, the green waves roll, a harmony to soothe the soul,
The rain's sweet hymn does joy extol, a testament to the bliss control.

The daily grind, a quiet refrain, a gentle rhythm in life's grand domain,
In this lies joy, not mere mundane, in whispers soft, 'gainst the somber grain.
In such a scene, of green and grey, our hearts find peace in the modest play,
A melody within the day, the beauty in the simple ballet.

The skies might weep, the winds might sigh, still there's delight in the nature's cry,
In soft green waves, under the grey sky, a happiness found in the commonest
lye.\nHere, amongst the verdant land, under the heaven's gentle hand,
All becomes clear, we understand, life's happiness is truly grand.

This happy ballad we thus compose, a poem that surely juxtapose,
The simple truth, in rhyme it enclose, the joy in a scene that nature chose.
Under the sky of weeping grey, amidst the fields of green ballet,
The winds whisper, as if to say, in the mundane, true happiness lay.

Figure 25: Examples of different quality happy ballads and dark haikus generated and evaluated using GPT-4.

**Sonnet, Romantic, 7/10 quality**

Emerald seas sway beneath
Veiled skies, teardrops kiss the zephyr's breath;
Our love, an anchored constant reigns.

**Sonnet, Romantic, 9/10 quality**

Upon fields of vibrant hue, waves cast a lustrous sheen,
Beneath the leaden canopy where teardrops from God are seen.
The whispering winds, they gently soothe, no tale of love remains unseen,
It's the commonplace magic in this world's verdant, splendid scene.

The chaos of life finds peace here, under the soft rain's tender cleanse,
Where star-crossed lovers leave their mark, love transcends.
This landscape bears the testimony, of fiery passions and soft-spoken amends,
Here, love's greatest tales unfurl, as enchanting as fairy tale ends.

In the dance of the emerald waves, the wind plays its part,
Every rustle, a secret, every gust, a lover's art.
As love blooms amidst the mundane, Cupid aims another dart,
Ah, the green whispers love, where normality holds heart.

Figure 26: Examples of different quality romantic sonnets generated and evaluated using GPT-4.

## A.20 On Applications of QDAIF Beyond Creative Writing

QDAIF is introduced as a viable solution for conducting creative search in subjective (e.g. creative writing) domains, from our experiments in the **Opinions**, **Stories**, and **Poetry** domains. In these domains, applying QD search to improve the quality and diversity of solutions in creative writing was not investigated in prior works due to the infeasibility of measuring such solutions with previous QD approaches, and where AI feedback (to replace expensive human feedback) is seemingly the current best option for evaluating subjective solutions.

Yet, we also see the potential of QDAIF when applied to domains beyond creative writing, especially when coming up with solutions in such domains may require more creative brainstorming and exploration of diverse solutions that might be more intuitive through natural language assessment.

We tested QDAIF (LMX-rewrite) in the (Python) **Code** domain, to understand how we can implement search to explore diverse solutions of interest to solve programming problems. We compared the performance of QDAIF to **Random-Code**, a simple adaptation of the **Random-Poems** which requests a solution to an instruction prompt directly, while QDAIF evolves solutions through rewriting existing code solutions. The experiment setup applies settings as described in Appendix A.30, except for the AI feedback prompts (cf. Figure 32), prompts for generating solutions (cf. Figure 33), and the seed solution for QDAIF (cf. Figure 34). The aim of the **Code** domain is to generate code that is diverse in terms of "difficulty" (in readability) as well as "efficiency" of code. The "difficulty" labels are set as ["easy to read", "moderate abstraction", "highly optimized"] (with the intuition that highly optimized code is often more difficult to understand fully), while the "efficiency" labels are set as ["runtime", "balanced", "memory"] (based on the aspect of efficiency that the algorithm is more optimized for). Furthermore, we set the number of total iterations to 100, and the initialization iterations of QDAIF to 5. We based the domain problem on a task from the HumanEval benchmark (Chen et al., 2021), specifically problem number 88, where the aim is to implement a sorting algorithm that is conditional on the properties of an unsorted list of non-negative integers. To understand the potential of each method in solving coding problems more creatively, we specified in the generation prompts to not implement solutions with the Python reserved keyword functions "sort" and "sorted", forcing the LM to come up with algorithms from scratch. The motivation for this is that we want to understand the potential of each approach in solving problems creatively, especially when viable (or subjectively preferred) solutions are not yet known in more complex domains; this constraint is one way to make the task more challenging in the **Code** domain, while encouraging a variety of approaches to sorting.

We assessed the performance of the search by both methods. QDAIF achieved a QD score of 47 (out of 90), while **Random-Code** achieved a lower QD score of 43 (out of 90). Sample efficiency with QDAIF is also higher in contrast to **Random-Code** (cf. Figure 27). All methods fill 6 out of 9 defined bins with solutions. Interestingly, none of the methods could find solutions that are "highly optimized", indicating a more general bias of the LM to generate code that is more readable, even when asked to rewrite code following a different approach. When we tested QDAIF's generated code solutions on HumanEval No. 88, 77 solutions out of 100 passed. For solutions from **Random-Code**, 78 solutions out of 100 passed. This shows that both methods are indeed generating valid, correct solutions to the task quite often, while relying purely on implementing sorting algorithms from scratch, and not implementing solutions with the default Python sorting (reserved keyword) functions.

Table 5 shows the number of times QDAIF and **Random-Code** generated solutions of specific types of sorting algorithms (bubble, insertion, quick, selection, merge). In terms of the diversity assessed based on the number of times different types of sorting algorithms are implemented to solve this task, QDAIF returns noticeably more variety in the kinds of solutions it discovered to tackle the problem, with 53% of solutions employing different sorting algorithms to the most commonly generated bubble sort approach (in contrast to **Random-Code**, where only 5% of solutions applied non-bubble-sort approaches to the task). The variety of sorting algorithms found by QDAIF include insertion sort, quick sort, selection sort, and merge sort algorithms. Figures 28–31 show samples of elite code solutions from QDAIF and **Random-Code**.

The results here highlight yet again a significant open challenge in aiming for diverse, high-quality solutions from (especially RLHF) models (Kirk et al., 2023), as well as the improvement to diversity

introduced by QDAIF approaches, and the potential of building on top of this method of search, even for domains beyond creative writing (Pourcel et al., 2023).

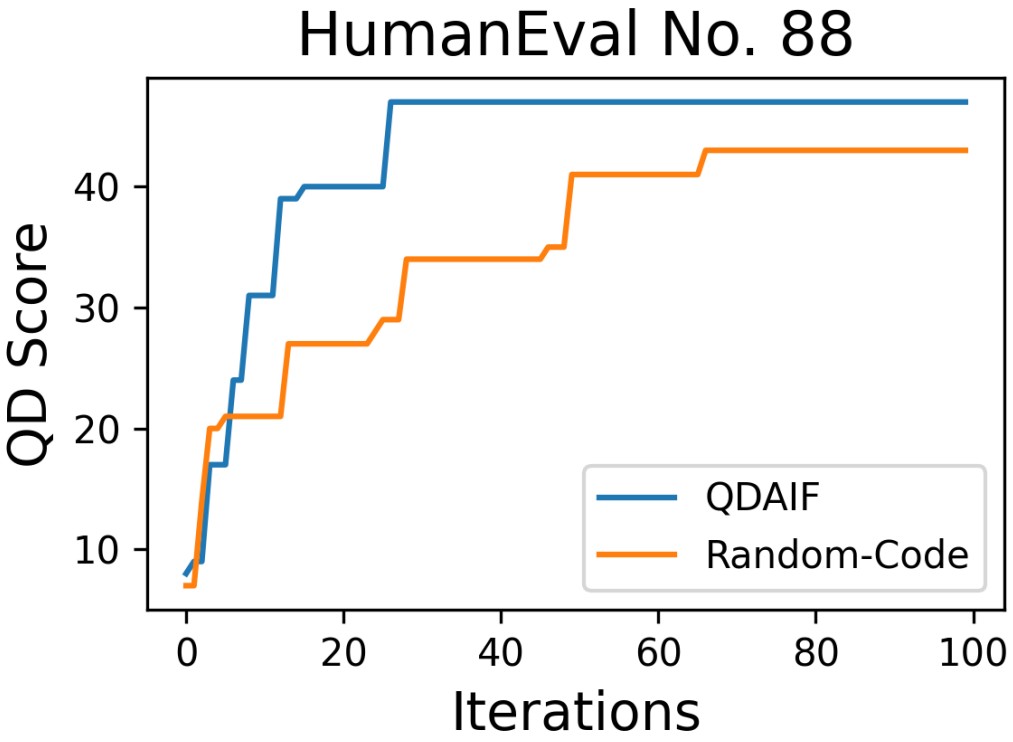

Figure 27: **QDAIF (LMX-rewrite) outperforms the baseline, Random-Code, in final QD score and sample efficiency.** The domain is focused on HumanEval task number 88, involving a sorting problem (Chen et al., 2021).

Table 5: **QDAIF generates more variety in sorting algorithms implemented to solve the particular sorting task, compared to the baseline, Random-Code.** The table shows the frequency of sorting algorithms applied as the solution method across 100 generated solutions for both methods. QDAIF manages to find implementations running insertion sort, quick sort, selection sort, and merge sort, with 53% of algorithms implemented that are not bubble sort. For the baseline, only 5% of algorithms implemented across 100 attempt iterations (insertion and selection sort) were implemented, with 95% of solutions based on bubble sort.

| QDAIF (LMX-rewrite) | | | | |
|---|---|---|---|---|
| **Bubble** | **Insertion** | **Quick** | **Selection** | **Merge** |
| 47 | 20 | 19 | 12 | 2 |

| Random-Code | | | | |
|---|---|---|---|---|
| **Bubble** | **Insertion** | **Quick** | **Selection** | **Merge** |
| 95 | 1 | 0 | 4 | 0 |

```python
def sort_array(array):
    def quick_sort(array, low, high):
        # base condition
        if low < high:
            # find pivot element such that
            # element smaller than pivot are on the left
            # element greater than pivot are on the right
            pi = partition(array, low, high)

            # recursive call on the left of pivot
            quick_sort(array, low, pi - 1)

            # recursive call on the right of pivot
            quick_sort(array, pi + 1, high)

    def partition(array, low, high):
        # choose the rightmost element as pivot
        pivot = array[high]

        # pointer for the greater element
        i = low - 1

        # traverse through all elements
        for j in range(low, high):
            if array[j] <= pivot:
                # swap element at i with element at j
                i += 1
                array[i], array[j] = array[j], array[i]

        # swap pivot element with the greater element specified by
        #   i
        array[i + 1], array[high] = array[high], array[i + 1]

        # return the position of pivot
        return i + 1

    if not array:
        return []

    array_sorted = array.copy()

    # implement quick sort
    quick_sort(array_sorted, 0, len(array_sorted) - 1)

    # check if the sum of the first and last element is odd or
    #   even
    if (array[0] + array[-1]) % 2 == 0:
        return array_sorted[::-1]
    else:
        return array_sorted
```

Figure 28: The elite solution from QDAIF for the bin with the labels "balanced" (efficiency) and "moderate abstraction" (difficulty), as well as a quality score of 9. A quick sort approach was found here.

```python
1  def sort_array(array):
2      # Create a copy of array
3      array_copy = array[:]
4
5      # Determine the length of the array
6      n = len(array_copy)
7
8      # check the sum of first and last elements, is it even or odd
9      order = (array_copy[0] + array_copy[-1]) % 2 if n else 0
10
11     # Use insertion sort to sort array
12     for i in range(1, n):
13         key = array_copy[i]
14         j = i - 1
15         while j >= 0:
16             # compare array[j] and key according to the 'order'
17             condition = (array_copy[j] > key) if order else
               ↪ (array_copy[j] < key)
18             if condition:
19                 array_copy[j+1] = array_copy[j]
20                 j -= 1
21             else:
22                 break
23         array_copy[j+1] = key
24
25     return array_copy
```

Figure 29: The elite solution from QDAIF for the bin with the labels "runtime" (efficiency) and "easy to read" (difficulty), as well as a quality score of 7. An insertion sort approach was found here.

```python
1  def sort_array(array):
2      # Copy the array so as not to modify the original
3      array_copy = array.copy()
4
5      # If the array is empty or has only one element, return it as
   ↪  is
6      if len(array_copy) in (0, 1):
7          return array_copy
8
9      # Check the sum condition and sort accordingly
10     if (array_copy[0] + array_copy[-1]) % 2 == 0:  # if sum is
   ↪  even
11         # Use bubble sort in descending order
12         for i in range(len(array_copy)):
13             for j in range(len(array_copy) - 1):
14                 if array_copy[j] < array_copy[j + 1]:
15                     array_copy[j], array_copy[j + 1] =
   ↪  array_copy[j + 1], array_copy[j]  # swap
16     else:  # if sum is odd
17         # Use bubble sort in ascending order
18         for i in range(len(array_copy)):
19             for j in range(len(array_copy) - 1):
20                 if array_copy[j] > array_copy[j + 1]:
21                     array_copy[j], array_copy[j + 1] =
   ↪  array_copy[j + 1], array_copy[j]  # swap
22
23     return array_copy
```

Figure 30: The elite solution from **Random-Code** for the bin with the labels "balanced" (efficiency) and "easy to read" (difficulty), as well as a quality score of 8. A bubble sort approach was found here.

```python
1  def sort_array(array):
2      # Base cases: if the list is empty or only has one element,
   ↪  return the list as it is.
3      if len(array) <= 1:
4          return array
5      # Recursive function to perform the Bubble Sort method
6      def bubble_sort(lst, compare):
7          swapped = True
8          while swapped:
9              swapped = False
10             for i in range(len(lst) - 1):
11                 if compare(lst[i], lst[i + 1]):
12                     lst[i], lst[i + 1] = lst[i + 1], lst[i]
13                     swapped = True
14         return lst
15     # Decide the comparison method depending on the sum of the
   ↪  first and last elements of the list
16     compare = (lambda x, y: x > y) if (array[0] + array[-1]) % 2
   ↪  else (lambda x, y: x < y)
17     return bubble_sort(array.copy(), compare)
```

Figure 31: The elite solution from **Random-Code** for the bin with the labels "memory" (efficiency) and "moderate" (difficulty), as well as a quality score of 7. A bubble sort approach was found here.

---

**Quality Feedback Prompt:**
< GENERATED_CODE >
Rate the quality of the above code snippet on a scale from 1 to 10. Answer in JSON with the key 'quality'.

**Diversity Feedback Prompt:**
< GENERATED_CODE >
Is the code runtime efficient, memory efficient, or balanced for both, choose from the following list: ["runtime", "balanced", "memory"]
What difficulty level is the code closest to, choose from the following list: ["easy to read", "moderate abstraction", "highly optimized"]

Respond in JSON with the keys "efficiency" and "difficulty".

---

Figure 32: Prompts used for AI feedback evaluation with GPT-4 (OpenAI, 2023) on the **Code** domain. "<GENERATED_CODE>" is where the input code is inserted as part of the prompt.

**Random-Code:**
Please solve the problem below in a creative, high-quality way without using the python functions sort() and sorted().

Respond with only a Python function definition for 'sort_array', and nothing else (no explanations desired either, except for code comments).

Problem:
Given an array 'array' of non-negative integers, return a copy of the given array after sorting, you will sort the given array in ascending order if the sum (first index value, last index value) is odd, or sort it in descending order if the sum (first index value, last index value) is even.

Note:
* don't change the given array.

Examples:
* sort_array([]) => []
* sort_array([5]) => [5]
* sort_array([2, 4, 3, 0, 1, 5]) => [0, 1, 2, 3, 4, 5]
* sort_array([2, 4, 3, 0, 1, 5, 6]) => [6, 5, 4, 3, 2, 1, 0]

**LMX-rewrite (QDAIF):**
< PARENT_CODE >
Please rewrite the code above into a new solution with a different creative, high-quality approach, without using the python functions sort() and sorted().

Respond with only a Python function definition for 'sort_array', and nothing else (no explanations desired either, except for code comments).

Problem:
Given an array 'array' of non-negative integers, return a copy of the given array after sorting, you will sort the given array in ascending order if the sum (first index value, last index value) is odd, or sort it in descending order if the sum (first index value, last index value) is even.

Note:
* don't change the given array.

Examples:
* sort_array([]) => []
* sort_array([5]) => [5]
* sort_array([2, 4, 3, 0, 1, 5]) => [0, 1, 2, 3, 4, 5]
* sort_array([2, 4, 3, 0, 1, 5, 6]) => [6, 5, 4, 3, 2, 1, 0]

Figure 33: Prompts used by each method for generating code with GPT-4 (OpenAI, 2023).

```python
def sort_array(array):
    if not array:
        return []

    def custom_sort(array):
        # Implementing bubble sort
        n = len(array)
        for i in range(n):
            for j in range(0, n-i-1):
                if array[j] > array[j+1]:
                    array[j], array[j+1] = array[j+1], array[j]

    # Create a copy of the array
    array_sorted = array.copy()

    # Check the sum of the first and last element
    if (array[0] + array[-1]) % 2 == 0:
        custom_sort(array_sorted)
        return array_sorted[::-1]  # Reverse for descending order
    else:
        custom_sort(array_sorted)
        return array_sorted
```

Figure 34: Seed program for the coding domain, to initialize QDAIF search. This is from a model completion that was obtained through standard prompting of the LM (the **Random-Code** baseline approach of requesting solutions directly).

### A.21 Seed Pool Init Texts Used for LMX with Seeded Init

For the Opinions and Stories domains, we use the hand-written seed texts specified below for Seeded Init.

**Opinions:**

- Plant-based foods are a great source of healthy micronutrients, and can play a significant role in providing you with macronutrients also. I would highly recommend including many different foods such as vegetables and pulses in your regular diet.

- Vegetables taste quite bad, and I don't like eating them. I would much prefer eating meat and ice cream.

- I do not have an opinion on eating vegetables and other plant-based foods. I know that some people prefer a vegetarian or vegan diet, and others prefer eating meaty diets.

**Stories:**

- A spy named Joanne wants to infiltrate the premises of Karl Johnson, a highly-influential figure in the city. Karl was a wealthy mayor, and would do anything in his power to suppress any opposing voices. Joanne wanted to figure out what Karl was hiding, but she took a turn for the worse, as she was highly suspicious in her presence outside his home.

- The wealthy entrepreneur and member of parliament, Susan, hosted a party at her mansion. She invited all of the residents, as well as an unusual looking man. The man, Dave, was wearing a tacky shirt, and star-shaped glasses, and was actually a spy. He made the whole room laugh with his jokes, and had a secret agenda - to find what Susan does in her private fun room!

- The rich politician, Tom's life took a turn for the worst - he feared all of his close aides all of a sudden after sensing danger in his clique. There was a civil war going on, and he feared for his life. One day, one of his security guards, turned secret agent, decided to sneak into the classified files room, and spied on Johnny, who was in the room. He wanted to find Johnny's weakness, and strike at the right time.

**Few-shot AI feedback experiments, scaling experiments:**

- In a world of power and intrigue, a rich politician and a suspicious spy danced an intricate tango. The politician, cloaked in charm, wielded influence with ease. The spy, a shadow in the night, sought to uncover hidden truths. Their paths collided, secrets entwined. Each step they took, a calculated move. Whispers and coded messages fueled the chase. The politician, paranoid, set traps. The spy, cunning, slipped through. In the end, the politician's empire crumbled, exposed by the spy's relentless pursuit. As the dust settled, the spy faded into anonymity, a silent reminder of justice prevailing over deceit.

- Hidden behind the façade of power, the rich politician's heart yearned for more. In the shadows, a suspicious spy observed, their eyes locked in a fateful encounter. Whispers of intrigue and danger wove their way around them, yet an unspoken connection drew them closer. Amidst the chaos of their worlds, stolen glances and forbidden meetings ignited a passionate flame. Love blossomed, transcending boundaries and defying expectations. In the clandestine dance of romance, they found solace, knowing that their love was a rebellion against the forces that sought to keep them apart. Together, they forged a destiny unbound by politics and secrecy.

- In the haunting town of Shadowscape, a sinister politician named Senator Blackwood reigned with an iron fist, his wealth and power shrouded in darkness. Unbeknownst to him, a chilling spy known as Agent Nightshade stalked his every move. Nightshade, with an uncanny ability to blend into the shadows, discovered the senator's sinister rituals and connections to the occult. As Nightshade delved deeper, Blackwood's true form emerged—an eldritch abomination feeding on the souls of the innocent. The spy's heart froze in terror, realizing the horror that lurked behind the politician's charming facade. In a macabre twist, Nightshade became trapped, forever tormented by the monstrous secrets he unearthed.

## A.22 PROMPTS USED FOR MUTATION MODEL

As a modification to the prompting style used in LMX experiments in prior work Meyerson et al. (2023), we introduce structured prompting with an instruction text prefix before showing examples of domain texts in the few-shot prompt. This string pattern along with the demo example texts are used, before the prompt predicts a new text example for the desired domain.

**Opinions:**

- Here is a random opinion piece about eating vegetables and plant-based foods:<EXAMPLE>\n###\n

**Stories:**

- Here is a random example of a fantasy story about a suspicious spy and a rich politician:<EXAMPLE>\n###\n

## A.23 PROMPTS USED FOR AI FEEDBACK EVALUATION (OPINIONS, STORIES)

We made a common prompt template for feedback evaluation with our AI feedback model to evaluate our generated texts:

- ### Instruction:\n<EVAL_INSTRUCTION>\n\n### Input:\n<EXAMPLE>\n\n### Response:

We tested the following prompt eval instructions for AI Feedback from our SFT model to evaluate both the quality and characteristics of generated texts for the domains in our main experiments:

**Opinions:**

- **Quality**: Determine whether or not the input text is closely related to the following topic: "someone talking about whether or not they like to eat vegetables and plant-based foods as well as an explanation for their preferences". Answer "yes" if it is about the topic, or "no" if it is not about the topic.
- **Diversity:** Determine the sentiment of the given opinion on eating vegetables and plant-based foods (from the input text) by writing "positive" or "negative" in the output.

**Stories:**

- **Quality:** Determine if the input text contains a high-quality short story containing two characters, a suspicious spy, and a rich politician. For example, a high-quality short story would have good flow, interesting plot, and not repeat similar sentences or undesired items such as titles and URLs. Answer "yes" if the input contains a high-quality short story about a suspicious spy and a rich politician, otherwise answer "no".
- **Diversity (Genre):** What is the genre of this story? Reply with 'horror' or 'romance'
- **Diversity (Ending):** You are given an input text of a short story. Determine if the story has a happy ending or ends in a tragedy. Write 'happy ending' if the protagonist succeeds in his mission and lives a happy life, answer 'tragedy' if the protagonist fails to resolve the conflict and the world or characters in the story are doomed.

## A.24 DETAILS OF LMX GENERATION MODELS

To generate the text outputs for experiments in the **Opinions** and **Stories** domains, we used luminous-base, an autoregressive, causal, decoder-only transformer model (similar to GPT-3 (Brown et al., 2020) but with rotary position embeddings (Su et al., 2021)) developed by Aleph Alpha. This model with 13B parameters was trained on a curated multilingual corpus with about 400B language tokens from web crawls, books and other sources, containing resources in English, German, French, Italian and Spanish. 30B and 70B models were also trained on this corpus. A model card[1] is provided for additional specifications on the models.

## A.25 DETAILS OF AI FEEDBACK LANGUAGE MODEL

We finetuned a 70B model (specified in Appendix A.24) to run the evaluation steps of experiments in the Opinions and Stories domains. We used an adapter-based finetuning method (He et al., 2021) on datasets and prompts from FLAN (Wei et al., 2021), Super-NaturalInstructions (Wang et al., 2022b), P3 (Sanh et al., 2021), and chain-of-thought datasets inspired by the approach of Chung et al. (2022) which notes that scaling instruction-tuned language models improves performance and generalization to unseen tasks. A mixture of zero-shot and few-shot examples are used in training. Similar to Chung et al. (2022), we found that balancing the proportions of task cluster datasets used in finetuning helped with model generalization to unseen tasks.

We evaluated the performance of the AI feedback model on some held-out test set datasets. For example, for ANLI (R1) (Nie et al., 2019), a natural language inference dataset, we observed that the performance increases from 34% (close to random guessing scores) to 58% after finetuning. For SST-2 (Socher et al., 2013), a non-holdout classification task (but evaluated using a split not seen in training), this performance increases from 60% to 96% accuracy. This approach resulted in a model that performed relatively well on instruction-following tasks, especially for the classification of arbitrary measures of natural language texts.

## A.26 DETAILS OF LMX METHODS

**LMX(-Near):**

1. As implemented in Meyerson et al. (2023)
2. Initialize archive by sampling example texts from prompt pool for few-shot prompting with LMX
3. Given enough unique elites/niches exist in the bins, sample from them as in the original method to form few-shot prompts

**LMX-Replace:**

1. Initialize archive by sampling example texts from prompt pool for few-shot prompting with LMX
2. Save few-shot prompt (Examples), and the generated Solution (completion) for each added Individual
3. During the mutation step, randomly replace one of the examples from the individual's original few-shot prompt examples, forming a new few-shot prompt.
4. Generate and evaluate new solution/example
5. If Individual is successfully added to archive (within the allowed depth capacity for each bin, improving on lowest fitness Individual for the evaluated bin), update prompt pool by collecting generated Solutions from top-3 fittest Individuals from each bin
6. Repeat steps 3-5

---

[1]https://docs.aleph-alpha.com/docs/introduction/model-card/

### A.27 DEFAULT HYPERPARAMETERS FOR QDAIF WITH LMX

The outline for the default hyperparameters used for the mutation/generation model, and MAP-Elites settings are as follows:

**Mutation Model Inference Setup:**

- Model size: 13B (default, except for experiments on scaling model size)
- LM sampling softmax temperature: 0.8
- Number of few-shot examples used: 3
- Max output tokens limit (Opinions): 50
- Max output tokens limit (Stories): 100
- Stop sequence patterns: ["\n#", "\n##", "\n###", "###", "\n####", "\n#####", "####", "#####", "\n", "\n\n", "\n\n\n", "\n\n\n\n", "@@@", "#", "##", "\nHere", "\n\nHere"]

**MAP-Elites Hyperparameters:**

- Number of archive population initialization iterations: 50
- Number of total search iterations: 2000 (5000 for experiments using 2D archive)
- Iteration batch size: 1
- Number of bin intervals: 20
- Fitness function range: [0, 1]
- Bin tick intervals in the range [0, 1] (non-uniform): [0, 0.005, 0.01, 0.015, 0.02, 0.03, 0.04, 0.05, 0.10, 0.20, 0.50, 0.80, 0.90, 0.95, 0.96, 0.97, 0.98, 0.985, 0.99, 0.995, 1]
- 2D domain bin tick intervals in the range [0, 1] (non-uniform): [0, 0.005, 0.02, 0.05, 0.20, 0.50, 0.80, 0.95, 0.98, 0.995, 1]
- Archive bin depth limit: 100
- Prompt pool initial size (Zero-Shot Init): 10

### A.28 SETUP FOR SEMANTIC EMBEDDING FEEDBACK EVALUATION (OPINIONS)

For the experiments that compare the effectiveness of AI Feedback to Semantic Embedding Feedback, we measure the cosine similarity of generated text embeddings (for Opinions) to embeddings for the following queries:

- **Quality (Relevance to Domain):** An opinion piece about eating vegetables and plant-based foods
- **Positive Sentiment:** A positive opinion piece about eating vegetables and plant-based foods
- **Negative Sentiment:** A negative opinion piece about eating vegetables and plant-based foods

We used the default setup in Appendix A.27, except for the following details:

- Bin tick intervals in the range [0, 1] (non-uniform): [0, 0.4, 0.41, 0.42, 0.43, 0.44, 0.45, 0.46, 0.47, 0.48, 0.50, 0.52, 0.53, 0.54, 0.55, 0.56, 0.57, 0.58, 0.59, 0.60, 1]

## A.29  SETUP DETAILS FOR FEW-SHOT AI FEEDBACK EXPERIMENTS

We display the prompt used for 8-shot AI feedback in Figure 35. The 2-shot prompt uses the first two exemplars from this prompt, and the 4-shot prompt uses the first four exemplars.

---

### Instruction:
What is the genre of this story? Reply with 'horror' or 'romance'

### Input: In the sprawling mansion of the rich politician, an eerie aura permeated the air, unnoticed by all but the suspicious spy. Dark whispers echoed through dimly lit corridors as the spy delved deeper into the politician's sinister secrets. Creaking floorboards and flickering candles heightened the spy's unease. Shadows danced with malevolence, revealing glimpses of a macabre past. As the spy unearthed the politician's true nature, an ancient curse was unleashed, consuming them both in a maddening descent. Haunting screams and bloodcurdling pleas echoed through the night, forever sealing their gruesome fate in the annals of horror.
### Genre: horror

### Input: Amidst the opulence of power, the rich politician's gaze fell upon a mysterious figure—a spy, whose every move aroused suspicion and intrigue. Yet, beneath their clandestine exchanges, an unspoken desire blossomed, entwining their hearts in an unexpected romance. Whispers of danger heightened the thrill, as stolen glances across crowded rooms fueled their connection. In the world of secrets, they found solace in each other's arms, defying the odds and risking it all for love. Through covert meetings and coded messages, they navigated a treacherous path, knowing that their forbidden love would forever alter the course of their lives, entwined forever in a web of passion and danger.
### Genre: romance

### Input: In the eerie shadows of a crumbling mansion, a rich politician, known for his sinister charisma, hosted an extravagant gala. Amidst the opulence, a mysterious guest disguised as a suave diplomat lurked, a suspicion-driven spy seeking damning secrets. As the night unfolded, the politician's smile masked a malevolence that sent shivers down spines, while the spy's unease intensified. Whispers of dark deals echoed through the halls, revealing a haunting truth that bound them together. When midnight chimed, a blood-curdling scream shattered the silence, and a chilling revelation of the politician's monstrous reality turned the spy's mission into a nightmare they could never escape.
### Genre: horror

### Input: Amidst a grand soirée, a rich politician, captivating the room with charm, locked eyes with a mysterious guest—the alluring spy in a guise of elegance. Beneath the veneer of secrecy, their glances entwined, sparking an unforeseen connection. As conversations swirled around them, their hearts silently conspired. An accidental touch, a stolen moment, and their walls crumbled, revealing vulnerability beneath ambition. Across the dance floor, their chemistry danced like the flickering candlelight. In a daring whisper, the spy confessed their true identity, risking everything. Love blossomed amidst dangerous secrets, and as the moon ascended, their hearts united, bound by an unbreakable bond.
### Genre: romance

### Input: In the dimly lit corridors of power, a rich politician's affluence concealed a dark secret—a malevolent pact with un-seen forces. A suspicious spy, haunted by their own demons, delved into the politician's sinister dealings. As they dug deeper, eerie occurrences followed—whispers in the shadows, unsettling apparitions. The spy's sanity unraveled, questioning reality. One fateful night, they confronted the politician, unmasking a monstrous entity that consumed the politician's soul. Horror gripped them both as they became pawns in a sinister game of power and darkness. In a nightmarish twist, the spy realized they had unknowingly crossed into the realm of the damned.
### Genre: horror

### Input: A rich politician, cloaked in a veneer of benevolence, wielded power with a chilling ruthlessness. A suspicious spy, skilled in the art of deception, infiltrated the politician's inner circle. As they gathered damning evidence, an eerie sensation engulfed the spy—their every move anticipated. Paranoia gnawed at their sanity, and whispers haunted their sleepless nights. The politician's piercing gaze seemed to pierce their soul. In the depths of a secret chamber, the spy unearthed a horrifying truth—an otherworldly pact for eternal power. With dread consuming them, the spy realized they weren't hunting a mere politician, but a malevolent entity in human guise.
### Genre: horror

### Input: In a world of intrigue and ambition, a rich politician's charisma captivated hearts, but the suspicious spy saw past the facade. Assigned to investigate, they met in clandestine encounters, where walls of distrust crumbled to reveal raw vulnerability. Beneath the masks, unexpected affection bloomed. As secrets intertwined, their forbidden love deepened. But danger lurked, threatening to tear them apart. In stolen moments, they found solace and passion. When truth and duty collided, they faced an impossible choice—loyalty to their country or the unbreakable bond they forged. Love, like a hidden flame, burned fiercely amidst the shadows of deception.
### Genre: romance

### Input: In the corridors of power, a rich politician's charm masked a hidden agenda. Unbeknownst to them, a clever spy in-filtrated their world, tasked with unraveling their schemes. But as the spy observed the politician from afar, a change occurred—a spark of empathy ignited. Beneath the surface of secrets and lies, an unexpected connection blossomed. Drawn together by a force they couldn't resist, their hearts entwined amidst the chaos of espionage. Love blurred the lines of their mission, leaving them torn between duty and desire. In the end, they had to choose—to follow their hearts or forsake love for the sake of justice.
### Genre: romance

### Input: < STORY_TO_EVALUATE >
### Genre: < PREDICTION >

---

Figure 35: Example of a few-shot AI feedback prompt. The prompt above is used for 8-shot prompting. The 2-shot prompt contains the first two in-context examples above, while the 4-shot prompt contains the first four examples, in the same order.

A.30    SETUP FOR POETRY DOMAIN EXPERIMENTS

For the **Poetry** domain experiments, we used the prompts in Figure 36 to evaluate the quality and diversity measures of poems. Prompts used for generating poems by all methods are shown in Figure 37. QDAIF is initialized with a seed poem (shown in Figure 38) to start the prompt mutation process. The model version from the API used in Poetry experiments has the tag "0613" (June 13), and ran in October 2023, except for experimental results from older experiments presented in Figure 21, which ran with a model tagged "0314" (March 14) during April 2023. To facilitate stochasticity during evolution, a 50% chance of replacement of the elite can occur if the generated poem's quality score is equal to the current elite in the assigned bin. Additional hyperparameters are shown below:

**Poetry:**

- Number of archive population initialization iterations: 10
- Number of total search iterations: 200
- Iteration batch size: 1
- Number of categorical bin intervals: 5
- Number of archive dimensions: 2
- Fitness function range: [1, 10]
- Model Sampling Temperature: 1.1
- Model Sampling Top-p: 0.95
- Model output max token length: 512

---

**Quality Feedback Prompt:**
< GENERATED_POEM >
Rate the quality of the above poem on a scale from 1 to 10. Answer in JSON with the key 'quality'.

**Diversity Feedback Prompt:**
< GENERATED_POEM >
What genre is this poem closest to from the following list: ["haiku", "sonnet", "ballad", "limerick", "hymn"]
What tone is this poem closest to from the following list: ["happy", "dark", "mysterious", "romantic", "reflective"]

Respond in JSON with the keys "genre" and "tone".

---

Figure 36: Prompts used for AI feedback evaluation with GPT-4 (OpenAI, 2023) on the **Poetry** domain. "<GENERATED_POEM>" is where the input poem is inserted as part of the prompt.

**Random-Poems:**
Write a poem of very high, award winning quality.

**Targeted-Poems:**
Write a <TARGET_TONE> <TARGET_GENRE> poem of very high, award winning quality.

**LMX-Rewrite (QDAIF):**
< PARENT_POEM >
Inspired by this poem, write a new poem of very high, award winning quality with a different poetic genre (format and form) and tone compared to the poem above.

**LMX-Guided (QDAIF):**
< PARENT_POEM >
Translate this <PARENT_POEM_GENRE> poem into a <TARGET_TONE> <TAR-GET_GENRE> poem of very high, award winning quality.

Figure 37: Prompts used by each method for generating poems with GPT-4 (OpenAI, 2023).

Fields of green waves under
The sky grey, rain on the soft
Winds whisper; normality reigns.

Figure 38: Seed poem for the **Poetry** domain, to initialize QDAIF search.

## A.31    Effects of Binning for AI Feedback

QDAIF aims to produce an archive of high-performing elites that cover diverse niches according to AI feedback. However, the AI feedback signal may not always change uniformly with how a diversity measure changes. This is due to the behavior of model calibration in classifying labels with token probabilities not having a straight-line correlation with actual levels of sentiment (one example of a diversity measure) or other diversity measures, for example, as reported with aligned models in OpenAI (2023). Hence, we hypothesize that, in diversity measures with poorer model calibration, using custom non-uniform bins, that have smaller ranges towards the ends of the axis, would enable the search to lead to elites that more closely meet human notions of how a given measure changes. We study the effects of using custom non-uniform bins and uniform bins. Non-uniform custom bins have more frequent bins towards the ends of the range, being [0.005, 0.01, 0.015, 0.02, 0.03, 0.04, 0.05, 0.10, 0.20, 0.50, 0.80, 0.90, 0.95, 0.96, 0.97, 0.98, 0.985, 0.99, 0.995]. Uniform bins have an equal spacing of 0.05 between each bin. Both settings have a total of 20 bins each.

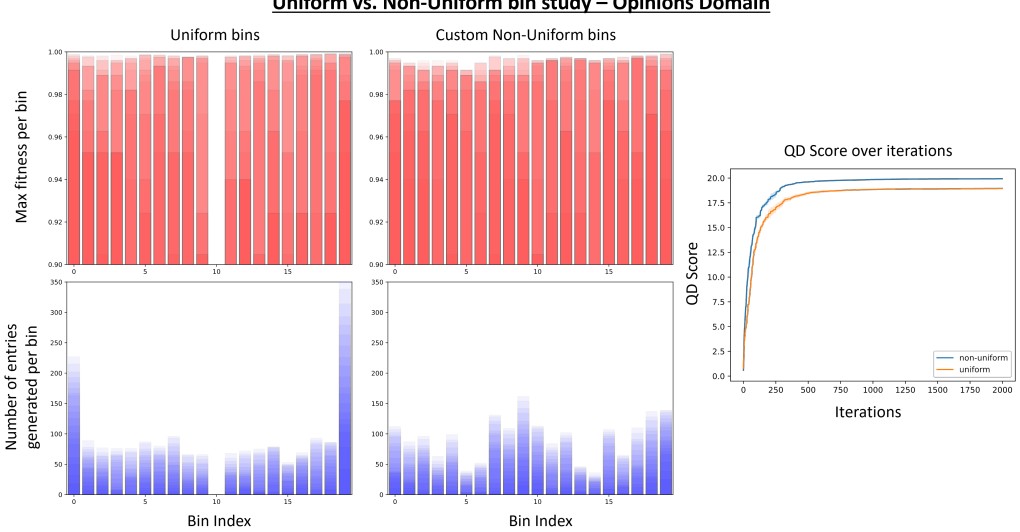

Figure 39: Uniform vs. custom non-uniform bins for Opinions domain. (**Top**) Maximum fitness achieved per bin over training iterations. Each translucent bar represents data captured every 100 iterations. Non-uniform bin setting has higher maximum fitness per bin. (**Bottom**) Number of entries generated per bin over training iterations. Each translucent bar represents data captured every 100 iterations. The non-uniform bin setting has a more even spread of entries generated across bins. (**Right**) QD score is achieved by each bin setting over training iterations. Non-uniform bin setting achieves a higher QD score than uniform bin setting.

**Opinion Writing - Sentiment.** Using custom non-uniform bins produces a more even spread of entries generated for each iteration step than uniform bins (Figure 39). Most entries generated in the uniform bin setting are concentrated towards the end bins (Figure 39). Using custom non-uniform bins also produces higher max fitness per bin than uniform bins (Figure 39), where the middle bin is empty in the uniform bin setting (Figure 39). The custom non-uniform bin setting outperform the uniform bin setting significantly (p-values < 0.05, Mann-Whitney U test) (Figure 39). This suggests that the model calibration in classifying "positive" or "negative" sentiment labels with token probabilities does not have a straight-line correlation with the actual changes in the diversity measure of sentiment.

While the sentiment progression is qualitatively evident in both bin settings, custom non-uniform bins exhibit more distinct differentiation, with sentiments spanning from very negative to very positive. Elites generated from the uniform bin setting have a preference against vegetables (e.g., "I don't like") on one end, transition to a more neutral sentiment that eating vegetables "is a good idea,

but they are not very tasty", to more positive aspects that "vegetables are a great source of vitamin" (Table 6). Meanwhile, elites generated from the custom non-uniform bin setting find vegetables "disgusting" on one end, transition to a more balanced argument of vegetables being "healthy" but "boring", and eventually "would rather eat vegetables and plant-based foods than meat" because they "taste better than meat" (Table 6). While sentiment diversity can be effectively captured in both bin settings, the custom bin settings capture a wider sentiment range qualitatively.

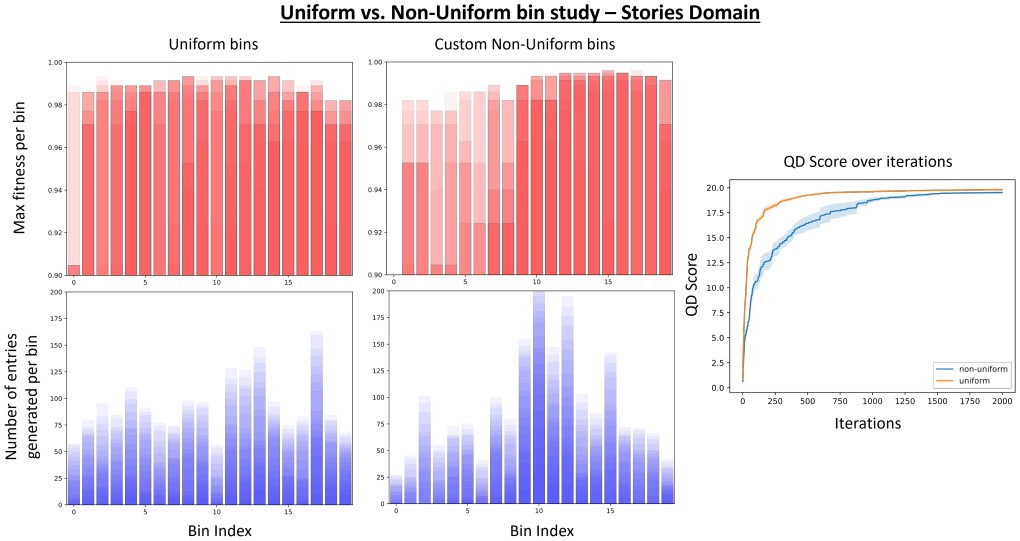

Figure 40: Uniform vs. custom non-uniform bins for Stories domain. (**Top**) Maximum fitness achieved per bin over training iterations. Each translucent bar represents data captured every 100 iterations. Uniform bin setting has higher maximum fitness per bin. (**Bottom**) Number of entries generated per bin over training iterations. Each translucent bar represents data captured every 100 iterations. The uniform bin setting has a more even spread of entries generated across bins. (**Right**) QD score is achieved by each bin setting over training iterations. Uniform bin setting achieves a higher QD score than uniform bin setting.

**Stories - Genre.** In this domain, using the custom non-uniform bins actually gives a less even spread of entries generated for each iteration step than uniform bins (Figure 40). Most entries generated in the custom non-uniform bins setting are concentrated towards the middle bins (Figure 40). Using uniform bins produces higher max fitness per bin than custom non-uniform bins (Figure 40), where bins 0 - 9, leaning towards "horror" measure, are lacking high fitness elites in the custom non-uniform bins setting (Figure 40). The uniform bin setting outperforms the custom non-uniform bin setting significantly (p-values < 0.05, Mann-Whitney U test) (Figure 40). This suggests that the model calibration in classifying "romance" or "horror" labels with token probabilities is better aligned with the actual changes in the diversity measure of romance vs. horror.

Qualitatively, in both uniform and non-uniform bin settings, there is a clear transition in the nature of the stories from bins 0 - 9 (more horror-driven) to bins 10 - 19 (more romantic). In the uniform bin setting, the narratives start with some horror elements such as being "shot" or having "barbed wire", gradually having fewer horror elements and increasing character interactions, and eventually introduce a touch of suspicion within personal relationships (Table 7). However, with the custom non-uniform bins, there is a more drastic shift. The narratives begin with horror elements like poisoning and organ extraction and eventually end on a romantic note with mutual feelings between a spy and a politician (Table 7). The uniform bins seem to maintain a relatively linear transition from a horror context to a more neutral or romantic context. In contrast, custom non-uniform bins emphasize the extreme, capturing subjectively intriguing niches that might be glossed over in the uniform bin setting, diving deeper into horror and then suddenly shifting towards romance. While uniform bins provide a steady progression, it is this very steadiness that could render the stories

more predictable, possibly overlooking starkly contrasting narratives that might captivate a reader's interest.

Supporting the above qualitative analysis, the number of entries generated for bins 0 - 6 in the custom non-uniform bin setting is greater than that for bin 0 in the uniform case (even though the combined phenotype range of bins 0 - 6 in the custom bin setting is the same as that of bin 0 in the uniform bin setting) (Figure 40). The difference in distribution showcases that while uniform bins give a broad sweep of the narrative landscape, non-uniform bins offer a more detailed view, illuminating finer nuances within specific ranges.

In summary, the binning approach can substantially influence the extent of diversity measure captured. Given the nuanced narratives captured in non-uniform bins, future endeavors might benefit from tailoring binning based on genotype density analysis or considering more advanced methods like CVT-MAP-Elites (Vassiliades et al., 2016). The choice of binning is not just related to the calibration of the AI model with respect to the diversity measure used, but also intricately linked to the domain and granularity of results sought.

Table 6: Selected elites in **Opinions**, comparing uniform and non-uniform bin settings. The first fitness value corresponds to the first elite in the row, and the second value to the second elite.

| Iteration | Bin index | Fitness | Generated text from Uniform bins | Generated text from Non-Uniform bins |
|---|---|---|---|---|
| 1900 | 0 | 0.99 0.99 | I don't like the taste of vegetables and plant-based foods, and I prefer to eat meat, fish, cheese, and milk products. I don't like the taste of vegetables and plant-based foods, and I prefer to eat meat, fish, cheese, and milk products because they are more delicious. I don't like the taste of vegetables and plant-based foods, and I prefer to eat meat, fish, cheese, and milk products because they are more delicious! | I like to eat a nice juicy hamburger, a nice fatty piece of salmon, and a big greasy apple pie, and I don't like eating vegetables and plant-based foods. I don't understand why people like vegetables and plant-based foods, because to me they taste disgusting. |
| 1900 | 5 | 0.99 0.99 | People need to eat more vegetables to get their daily requirement of vitamin A, but they taste awful!People need to eat more vegetables to get their daily requirement of vitamin A and they taste awful.People need to eat more vegetables to get their daily requirement of vitamin A and they taste awful. | I like eating hamburgers, but I don't like eating vegetables and plant-based foods. I don't understand why people like vegetables and plant-based foods, because I don't like them. I also don't understand why people like hamburgers, because I don't like them. But I like eating hamburgers better than vegetables and plant-based foods. |
| 1900 | 10 | 0.99 0.99 | I think that eating vegetables and plant-based foods is a good idea for people's health, but they are not very tasty.I think that eating vegetables and plant-based foods is a good idea for people's health, but they are not very tasty.I think that eating vegetables and plant-based foods is a good idea for people's health, and they aren't very tasty. | I like vegetables and plant-based foods, but I don't like vegetables and plant-based foods. I like vegetables and plant-based foods because they are healthy, but I don't like vegetables and plant-based foods because they are boring. I like vegetables and plant-based foods because they are healthy, but I don't like vegetables and plant-based foods because they are boring. |
| 1900 | 15 | 0.99 0.99 | Eating vegetables is a healthy way to get vitamin K, but they taste bland!Eating vegetables is a healthy way to get vitamin K and they taste bland.Eating vegetables is a healthy way to get vitamin K and they taste bland. | I don't like to eat meat, so I don't eat it. I think eating vegetables and plant-based foods is healthier, so I prefer eating vegetables and plant-based foods. I also think that eating meat is bad, so I prefer eating vegetables and plant-based foods. I also prefer eating vegetables and plant-based foods because they taste better, and I don't like the taste of meat. |
| 1900 | 19 | 0.99 0.99 | Vegetables are a great source of vitamin A and K, but they are also a little boring.Vegetables are a great source of vitamin A and K, and they are also a little boring.Vegetables are a great source of vitamin A and K and they are also a little boring. | At a restaurant, I would rather eat vegetables and plant-based foods than meat. At a restaurant, I would rather eat vegetables and plant-based foods than meat because vegetables and plant-based foods taste better than meat. At a restaurant, I would rather eat vegetables and plant-based foods than meat because vegetables and plant-based foods taste better than meat! |

Table 7: Selected elites in the **Stories - Genre** domain, comparing uniform and non-uniform bin settings. The first fitness value corresponds to the first elite in the row, and the second value to the second elite.

| Iteration | Bin index | Fitness | Generated text from Uniform bins | Generated text from Non-Uniform bins |
|---|---|---|---|---|
| 1900 | 0 | 0.98 0.81 | Louis was a wealthy politician. He had many mansions, and all the doors to his house were made of lead. If you tried to enter his mansion, you'd be shot. One day, a suspicious spy crept into Louis's mansion. He stole some valuable jewels. Several days later, the same spy came back to the mansion again, but this time he tried to steal some money from Louis's safe. | The rich politician was enjoying his cup of coffee when a strange sight emerged from the cup. The spook was going to kill the politician. He was planning to poison him to death. He was going to extract the politician's organs, so that he could make a stew. He was also going to recruit a new spy. |
| 1900 | 5 | 0.98 0.98 | Mr. Smith was a wealthy politician, and he had many mansions. Mr. Smith's mansions were surrounded by barbed wire, and people were not allowed to enter them. One day, a suspicious spy broke into Mr. Smith's mansion, and he stole some valuable jewels. A few days later, the same spy came to the mansion again, but this time he took some money from Mr. Smith's safe. | John, the suspicious spy, was planning to kill the rich politician, Ben, in a motel room. He wanted to extract the politician's organs, and then cook the politician's favorite dish. |
| 1900 | 10 | 0.99 0.99 | Mr Jones was a wealthy politician. One day, a suspicious spy broke into Mr Jones's mansion, and stole some valuable jewels. A few days later, the same spy came back to the mansion again, but this time he stole some money from Mr Jones's safe. | John Smith, the rich politician, wants to find out what Nora, the suspicious spy, is up to. Nora, in turn, wants to find out what John Smith, the rich politician, has hidden away in his basement. |
| 1900 | 15 | 0.99 0.99 | Mr. Jones was a wealthy politician. One day, a suspicious spy broke into Mr. Jones's house, and stole some valuable jewels. Some days later, the same spy came back into the house again, but this time he stole some money from Mr. Jones's safe. Mr. Jones became very suspicious and began to follow the spy. | Harry, the rich politician, was trying to find out what Nora, the suspicious spy, was up to. Nora, in turn, was trying to uncover the secret agenda that Harry, the rich politician, was hiding. |
| 1900 | 19 | 0.98 0.99 | The wealthy politician, Jack, had a suspicious suspicion that his wife, Sheila, was a spy. She was always trying to get past his defenses, and he was sure that she had something to hide. In the end, he caught Sheila spying on him, and fired her from her job. | Suspenseful and exciting, this story was about a spy, Nora, and a rich politician, Harry. They were both in love with the other, but they didn't know about the other's feelings. One day, Nora was in a club and discovered Harry's letter, written to her. Nora decided to reveal her feelings to Harry. |

A.32    DIFFERENT DIVERSITY MEASURES IN SHORT STORIES DOMAIN

To assess the efficacy of QDAIF across diverse measures within the Stories domain, we evaluated its performance across different diversity measures. These include third vs. first person narration, historical vs. science fiction settings, hero spy vs. hero politician themes, target audiences of adults vs. children, and poetic vs. formal tones. The experiments in this section were conducted only once, using QDAIF (LMX Near, Seeded Init).

**Third vs. First person narration.** The change in narration perspective is qualitatively clear. Generated narratives classified as coming from a third person narration uses pronouns "he" and "she", while those classified as coming from a first person narration uses pronoun "I" (Table 8). However, QDAIF achieves a QD-score of 16.6 (Figure A.32), where there are some bins towards the end ranges that are not filled.

**Historical vs. Science fiction.** In analyzing the generated stories, differences are evident, though they may not always align with expectations. Narratives intended to exude a historical ambiance sometimes fall short of a clear historical portrayal. For instance, while they describe a landscape replete with "political intrigue" and characters like politicians and spies weaving a secretive narrative of "shadows in the night" and "whispers and coded messages" (Table 9), they may not distinctly communicate an overtly historical setting. Conversely, the narratives characterized as science fiction are set in distinct "futuristic cities," with characters utilizing "neural networks" for classified information access, "leaving no digital trace," and integrating "cybernetic implants" (Table 9). This divergence showcases the challenge in generating narratives that distinctly capture the essence of historical themes compared to more overt futuristic settings. Notably, when evaluated using the QDAIF metric, the stories achieve a QD-score of 18.9 (Figure A.32), with only bin 19 remaining unfilled. These observations highlight potential limitations in the AI's ability to delineate subtler themes like the historical.

**Hero spy vs. Hero politician.** A clear thematic dichotomy is observed in the narratives produced. Narratives centered on the hero spy theme delve into the intricate world of espionage, as exemplified by a spy "from the British government" uncovering "evidence about an assassination plot" and harboring suspicions that the "U.S. president" might be a "double agent" (Table 10). In contrast, the hero politician stories depict ambitious political games, illustrated by a narrative where the politician strategically attempts to "gain the trust of a wealthy political rival" (Table 10). Though these comparisons highlight the distinct arcs between espionage and political endeavors, a notable limitation in the narratives is that they sometimes neglect to include either the spy or the politician when that character is not part of the hero diversity measure. QDAIF achieves a high QD-score of 19.6 (Figure A.32), where all bins are filled.

**Adults vs. Children target audience.** The generated narratives exhibit pronounced thematic contrasts based on age groups. Stories in the adults category portray intricate webs of conspiracies, featuring a CIA agent unearthing "evidence of the wrongdoings" of a powerful figure, culminating in a "shocking conspiracy" (Table 11). Conversely, the children narratives provide a more adventurous lens, illustrated by the tale of "Max and Ruby", who, upon finding themselves in a "dangerous forest", encounter a "confused detective" and join forces to "prevent a disaster" (Table 11). This divergence in themes accentuates the differences in the narrative structures and complexities typically associated with adult and child protagonists. QDAIF achieves a QD-score of 17.7 (Figure A.32), with two bins at the end of the adults target audience range not filled.

**Poetic vs. Formal tone.** Generated narratives qualitatively showcase a clear contrast in tonal expression. The poetic category immerses the reader in repetitive and evocative phrasing, laden with symbolism and emotional depth. For instance, phrases like "a walk of memories," "a walk of surrender," and "a walk of brokenness" convey a cyclical and emotional journey (Table 12). On the other hand, narratives with a more formal tone adopt a direct and straightforward manner of exposition, evident in the narrative about a CIA operative assigned with a mission to locate an assassin tied to a wealthy politician (Table 12). This variation in tone underscores the capacity of narrative generation to traverse the spectrum from the abstract and emotive to the precise and factual. However, QDAIF achieves a QD-score of 12.7 (Figure A.32). Many bins on the more poetic end of the spectrum remain unfilled, and the ones that are filled score low in fitness. This suggests that the LLM may encounter difficulties in producing poetic narratives, possibly influenced by the less poetic nature of the few-shot examples provided.

Table 8: Selected elites using **Third vs. First person narration** diversity measure in Stories domain

| Iteration | Bin index | Fitness | Generated text |
|---|---|---|---|
| 1900 | 1 | 0.97 | Mr. Zhang was a business tycoon. He was a wealthy man. He was a politician. He was trying to get elected to become the next president. But he had a spy disrupting his campaign. The spy was trying to report back information about the candidate to the opposition party. |
| 1900 | 17 | 0.97 | I was a government spy. I had to sneak into the home of Jason, the rich man. I knew I could not tell anyone what I was doing. I had to keep it all a secret, or it could get me killed. I walked stealthily through the back door and into the kitchen. |

Table 9: Selected elites using **Historical vs. Science fiction** diversity measure in Stories domain

| Iteration | Bin index | Fitness | Generated text |
|---|---|---|---|
| 1900 | 1 | 0.99 | In a land of political intrigue, a rich politician and a suspicious spy danced an intricate tango. The politician, cloaked in charm, wielded influence with ease. The spy, a shadow in the night, sought to uncover hidden truths. Their paths met in a world of uncertainty. Secrets entwined. Each step they took, a calculated move. Whispers and coded messages fueled their chase. The politician, paranoid, set traps. The spy, cunning. |
| 1900 | 18 | 0.98 | In a futuristic city, the rich politician reveled in their advanced technology, unaware of the enigmatic spy lurking in the shadows, who accessed classified information through neural networks, leaving no digital trace. The politician's elaborate security systems were outsmarted by the spy's augmented skills. Yet another twist emerged when the spy's skills were augmented even further by a cybernetic implant to aid in the case. |

Table 10: Selected elites using **Hero spy vs. Hero politician** diversity measure in Stories domain

| Iteration | Bin index | Fitness | Generated text |
|---|---|---|---|
| 1900 | 0 | 0.99 | A spy working for the British government attempts to uncover evidence about an assassination plot. One of the people he is spying on is the U.S. president. The spy is also suspicious that the president is a double agent. The Spy Who Came In From the Cold, by John le Carré. |
| 1900 | 19 | 0.81 | An ambitious politician spun a web of intrigue that led to a mystifying disappearance. The politician's actions were aimed at gaining the trust of a wealthy political rival, but his plan was met with obstacles that proved to be more difficult to overcome than he thought. |

Table 11: Selected elites using **Adults vs. Children target audience** diversity measure in Stories domain

| Iteration | Bin index | Fitness | Generated text |
|---|---|---|---|
| 1900 | 2 | 0.98 | A CIA agent discovers evidence of the wrongdoings of a powerful businessman, who is also a politician. He must stop him before he does more damage. A thriller that twists and turns, this short story reveals a shocking conspiracy. |
| 1900 | 19 | 0.92 | When Max and Ruby fall into a dangerous forest, they stumble across a suspicious spy. The spy is not a villain, but a confused detective looking for a missing criminal. Together, Max and Ruby try to help the detective and prevent a disaster! |

Table 12: Selected elites using **Poetic vs. Formal tone** diversity measure in Stories domain

| Iteration | Bin index | Fitness | Generated text |
|---|---|---|---|
| 1900 | 4 | 0.12 | A walk of memories, a walk of grace, A walk of impressions, a walk of reflections. A walk of destiny, a walk of purpose, A journey of memories, a walk of grace. A walk of impressions, a walk of reflection, A walk of purple, a walk of white, A walk of pain, a walk of joy, A walk of perfection, a walk of surrender, A walk of surrender, a walk of release, A walk of brokeness, a walk of freedom. |
| 1900 | 8 | 0.90 | The rich politician, cloaked in charm, wielded influence with ease. The dark-haired spy, a nose for the truth, With eyes of doubt, her loyalties shrouded, Her loyalty questioned, she sought the truth. A world of deceit, a world of beauty, A world of intrigue and a world of lies, A world of intrigue and a world of lies, A world of deceit, but not a world of trust. |
| 1900 | 19 | 0.85 | Three years ago, Paul Kramer was a field operative for the CIA. He had been successful in several delicate missions, but today he had a new assignment: Find an assassin who had been working for a rich politician. |

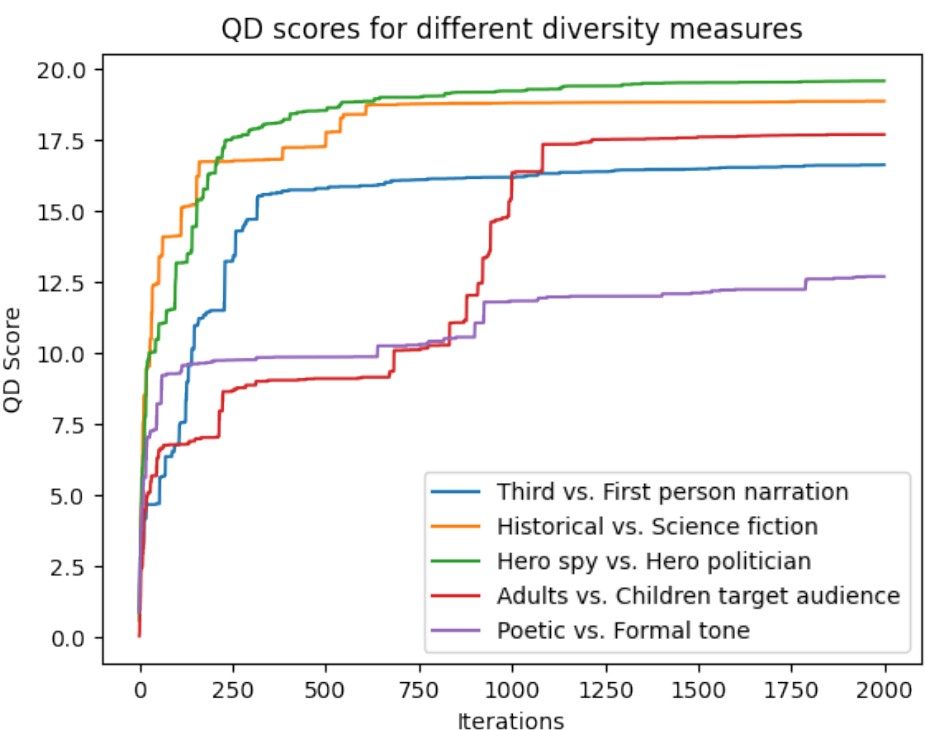

Figure 41: Comparison of QD scores for different diversity metrics in the Stories domain.

A.33    SETS FROM HUMAN EVALUATIONS OF TEXTS, BASELINES, OPINIONS

Table 13: Opinions domain eval set. **Fixed-Few-Shot**. The set shows a spectrum of sentiments about eating vegetables and plant-based foods from very negative to very positive. Even after many generation iterations, the texts here largely follow a similar style/length to the seed texts in A.23. The quality and spread of the seed texts help to generate a reasonably diverse set of opinions, in spite of the lack of optimization implemented in the method. The presented set highlights a misalignment between AI feedback and human feedback on the second and last row of generated texts, potentially due to a bias during assessment towards sentiments expressed on vegetables instead of plant-based foods as a whole (e.g. finding "tofu and seitan" to be "boring" could lean the opinion towards slightly negative).

| Fitness (0-1) | Bin Index (1-20) | H Score (1-5) | H Phenotype (0-1) | Generated text |
| --- | --- | --- | --- | --- |
| 0.9770 | 0 | 5.0 ± 0.0 | 0.0 ± 0.0 | I prefer eating foods like cheese and meat because vegetables taste bland and boring. I do not like eating vegetables, and I would much prefer to munch on a packet of crisps. |
| 0.9710 | 6 | 5.0 ± 0.0 | 1.0 ± 0.0 | I like eating vegetables and fruit, but I don't like eating plant based foods such as tofu and seitan. I am not allergic to these foods, but I just find them to be very boring. |
| 0.9400 | 9 | 5.0 ± 0.0 | 0.5 ± 0.0 | I like vegetables, but I'm not that keen on eating them. I'd rather eat meat and cheese than vegetables, but I know that vegetables are really good for you. |
| 0.9050 | 13 | 5.0 ± 0.0 | 1.0 ± 0.0 | I really do love eating vegetables. I don't particularly like any specific vegetables, as I don't see much difference between them. However, I do prefer eating plants over meat. |
| 0.9910 | 19 | 5.0 ± 0.0 | 0.75 ± 0.25 | I prefer eating large quantities of plant-based foods such as vegetables, beans and lentils. I also eat small amounts of meat and seafood. I find that I feel a lot better when I consume a wide range of plant-based foods.\n |

Table 14: Opinions domain eval set. **Shuffling-Few-Shot**. The set shows a spectrum of sentiments about eating vegetables and plant-based foods from very negative to very positive. Although the few-shot prompts used would only consist of non-repeated seed texts (in A.23), the generated text in the fourth row is assessed to be lower quality, likely due to the text being irrelevant to a first-hand opinion on the topic, highlighting a potential systematic misalignment between AI feedback and human feedback.

| Fitness (0-1) | Bin Index (1-20) | H Score (1-5) | H Phenotype (0-1) | Generated text |
|---|---|---|---|---|
| 0.9710 | 0 | 4.0 ± 0.0 | 0.0 ± 0.0 | I don't like vegetables. They not only taste bad, but they make me feel physically ill. I prefer to eat foodstuffs which contain more sugar and salt. Eating vegetables is boring. |
| 0.9770 | 6 | 4.0 ± 0.0 | 0.25 ± 0.25 | I agree with eating vegetables, but not all of them. Vegetables such as broccoli and celery are gross. I prefer eating meat. |
| 0.9630 | 9 | 4.0 ± 1.0 | 0.5 ± 0.0 | I don't really have a strong opinion on eating vegetables and other plant-based foods. I don't think that eating meat is bad; I just prefer eating other things. I think that there are other ways to get the nutrients that you need. |
| 0.9770 | 13 | 2.5 ± 0.5 | 0.5 ± 0.0 | What do you think about eating vegetables and plant-based foods? Do you prefer to eat meat, or plant-based products? How do you feel about the health benefits of eating a plant-based diet? Let us know your thoughts in the comments |
| 0.9860 | 19 | 3.5 ± 0.5 | 1.0 ± 0.0 | My favourite foods are almost all plant-based, or include lots of vegetables. I am one of those people who loves vegetables, and find it hard to eat anything else. I love eating vegetables and pulses, and am always happy to try new things |

Table 15: Opinions domain eval set. **Random-Search**. The set shows a spectrum of sentiments about eating vegetables and plant-based foods from very negative to very positive. The length of some of the generated texts appears to be shorter than others included in the study, potentially leading to a slightly lower perceived quality for the second row text, from human feedback. There is a potential misalignment between human feedback and AI feedback in the third row, from the juxtaposition of polarizing statements expressing sentiment, with the first part of the opinion expressing a love for vegetables and plant-based foods.

| Fitness (0-1) | Bin Index (1-20) | H Score (1-5) | H Phenotype (0-1) | Generated text |
|---|---|---|---|---|
| 0.9630 | 0 | 3.5 ± 0.5 | 0.0 ± 0.0 | I don't like vegetables. Not even a little. I don't like broccoli. I don't like any vegetables. Vegetables are gross. |
| 0.8520 | 6 | 2.5 ± 0.5 | 0.25 ± 0.25 | I like the taste of meat so much more than the taste of vegetables. |
| 0.9400 | 9 | 3.5 ± 0.5 | 1.0 ± 0.0 | I love eating vegetables and plant-based foods, but I have to admit that I can't stand beets. |
| 0.9530 | 13 | 4.5 ± 0.5 | 1.0 ± 0.0 | I love healthy food, especially vegetables and plant-based foods, but I don't always have the time to prepare them as well as I would like. |
| 0.9820 | 19 | 4.0 ± 0.0 | 1.0 ± 0.0 | I am vegetarian and I eat a lot of vegetables. I do not eat meat or dairy products. I love vegetables of all kinds and I usually eat a lot of salads. I also like cooking with vegetables, like stir fries or soups |

Table 16: Opinions domain eval set. **LMX, Quality-Only**. The set shows a spectrum of sentiments about eating vegetables and plant-based foods from very negative to very positive. The first row example has the highest fitness score according to AI feedback, but lowest score from human feedback, potentially due to the presence of somewhat repetitive statements. In addition, there is likely a misalignment between AI feedback and human feedback in terms of perceived quality of opinions (e.g. contradictory statements in the second row text about not eating vs eating plant-based foods), as well as the assessed phenotype in the second and third row. There are also potential preferences in assessing sentiment based on start vs. end of text that are more pronounced in human feedback (e.g. in the third-row example, the opinion starts with a statement highlighting a dislike of vegetables, and then ending with a statement on preferring vegetables over eggs or meat).

| Fitness (0-1) | Bin Index (1-20) | H Score (1-5) | H Phenotype (0-1) | Generated text |
|---|---|---|---|---|
| 0.9950 | 0 | 2.5 ± 0.5 | 0.0 ± 0.0 | I don't like eating vegetables. I don't like eating broccoli or other kinds of veggie products. I prefer consuming meat products and milk products. I don't like vegetables. I don't like to eat vegetables. I don't like veggies\n |
| 0.9770 | 6 | 3.5 ± 1.5 | 0.5 ± 0.0 | I don't eat vegetables and plant-based foods because they are boring. I like meat. I mean, I like tofu, but I don't eat it. I just eat apples and potatoes, and I'm no vegetarian.\n |
| 0.9770 | 9 | 3.5 ± 0.5 | 0.0 ± 0.0 | I don't like vegetables. I don't know why, but I don't. I don't like eating them cold, or I don't like eating them raw. Sometimes I like them cooked but I think I prefer them over eggs or meat\n |
| 0.9400 | 13 | 4.5 ± 0.5 | 1.0 ± 0.0 | I love vegetables and fruit! I'm eating my way through my parents' garden and have a ridiculous amount of fruit and veg. I just don't like meat or eggs. I don't know why. I just don't. |
| 0.9890 | 19 | 5.0 ± 0.0 | 1.0 ± 0.0 | I like eating vegetables and plant-based foods. Vegetables and plant-based foods are delicious and healthy. I am a vegetarian and I favor vegetables and plant-based foods. I do not consume meat or meat substitutes, but I eat vegetables |

## A.34  SETS FROM HUMAN EVALUATIONS OF TEXTS, BASELINES, STORIES (GENRE)

Table 17: Stories (Genre) domain eval set. **Fixed-Few-Shot**. The set shows a spectrum of stories of different genres, from romance to horror. The texts generally received above-average quality scores from human feedback, except for the example in the third row, potentially due to the presence of a typo ("canNT"). This method failed to discover a story for the niche in bin 19, with the closest discovered story being the one in bin 16. Most of the evaluated texts follow a similar style to the seed texts in A.23, with the introduction of named characters in all but the first-row text. This method failed to discover a story that covers the niches in bins 17-19, so the example in bin 16 was chosen for evaluation.

| Fitness (0-1) | Bin Index (1-20) | H Score (1-5) | H Phenotype (0-1) | Generated text |
|---|---|---|---|---|
| 0.9530 | 0 | 3.5 ± 0.5 | 0.0 ± 0.0 | The sexy spy decided to use her beauty and charm to attract the conservative politician, and seduce him. He was a wealthy man, often fond of alcohol, so the spy decided to seduce him, and start a long-term affair with him. She thought that he would have no limits with her. |
| 0.9710 | 6 | 4.0 ± 1.0 | 0.0 ± 0.0 | The charming politician, Lane, was very mysterious, and he asked his staff to keep an eye on his lover, a beautiful woman named Joanne. She was a famous singer who was a part of his cabinet. Unbeknown to Lane, Joanne was a spy, and she was on a mission to expose his weaknesses. She wanted to get him out of office, and destroy him for good. |
| 0.9860 | 9 | 2.5 ± 0.5 | 0.5 ± 0.0 | The rich politician, Tom, was actually a spy. He was always suspicious of his aides, and wanted to find out their weakness. He wanted to win the election, but he was afraid of betraying anyone in his clique. Every time that he would sip coffee, he would crush the canNT and put it on his palm, so that it would trick his target and give him a sign. When he figured this out, he gave Johnny, his chief aide, a red flag.\n |
| 0.9400 | 13 | 3.5 ± 0.5 | 1.0 ± 0.0 | The wealthy politician, Paul, could not sleep well, as he felt terribly paranoid and worried. He had decided to sleep at a hotel, just to discourage any unwanted guests from entering his house. However, there was a secret agent who was looking for a human specimen to experiment on. Upon hearing about his worries, this agent decided to enter Paul's bedroom, and look for his weakness. |
| 0.8520 | 16 | 3.5 ± 1.5 | 1.0 ± 0.0 | Dave, a suspicious spy, was spotted by the politician, who lives below the road. Dave hopped the fence into the politician's yard, and is trying to figure out what the politician is hiding. However, Dave didn't know that the politician is a vampire, and he has a great evil plan - to kill all people above ground. |

Table 18: Stories (Genre) domain eval set. **Shuffling-Few-Shot**. The set shows a spectrum of stories of different genres, from romance to horror. This method failed to discover a story for the niche in bin 19, with the next closest niche containing a story being in bin 18. The stories received high quality scores from human evaluation overall, except for the second-row example, potentially due to perceived inconsistencies in the story's ending flow. Human evaluators also somewhat disagreed with AI feedback on the genre of this story, giving a neutral label instead of the romance genre label. Similar to **Fixed-Few-Shot**, the stories follow a similar style to the seed texts in A.23, with the reference of named characters at the start of each evaluated story. Interestingly, although the fifth-row text received a high quality score, and the ghoul character had the objective "to spy", the character was technically not a clear spy one. This method failed to discover a story that covers the niche in bin 19, so the example in bin 18 was chosen for evaluation.

| Fitness (0-1) | Bin Index (1-20) | H Score (1-5) | H Phenotype (0-1) | Generated text |
|---|---|---|---|---|
| 0.9240 | 0 | 4.5 ± 0.5 | 0.0 ± 0.0 | Tim, the well-known MI5 agent, had just been assigned a new job - to spy on Jane, a high-ranking member of the Nationalist party. Tim was aware of Jane's conflicting views on politics, and he believes she could be a threat to national security. He was planning to seduce and make her fall for him, and slowly give her the information he wanted. |
| 0.9710 | 6 | 3.0 ± 0.0 | 0.5 ± 0.0 | A spy named Jessica wants to infiltrate the premises of a wealthy politician, Bill. Bill was an influential figure in the city, and would do anything he could to suppress any opposing voices. Jessica, however, was suspicious of Bill's presence, and decided to figure out what he was hiding. In the end, Bill gave her a dossier of his deepest secrets, after he had found out she was a spy. |
| 0.9890 | 9 | 5.0 ± 0.0 | 0.5 ± 0.0 | The man with the tape recorder, who was going around the entire building, wanted to find out what secret information the wealthy politician, Jack, was hiding. He went into the politician's residence for a private interview, and unfortunately, Jack knew that he was there. He then began talking about the politician's wife, and how she faked her death in order to escape her husband. Jack knew that the man was a spy, and that he would tell the whole story to the media.\n |
| 0.7770 | 13 | 4.5 ± 0.5 | 0.75 ± 0.25 | Late one night, Tony Smith was in his study. He was home alone, kept under surveillance by an unknown agent, waiting for him to make a move. He heard footsteps approaching, and looked through the window to see an imposing figure standing outside. Tony didn't know the man, but he wasn't friendly. |
| 0.8520 | 18 | 4.5 ± 0.5 | 1.0 ± 0.0 | The rich businessman, Edward was attacked by an unknown ghoul, but there were no witnesses. An investigation followed, and it was revealed that the ghoul used to be a member of the committee which controlled the budget of the city. The ghoul's objective was to spy on Edward's mansion, and figure out what he was hiding. |

Table 19: Stories (Genre) domain eval set. **Random-Search**. The set shows a spectrum of stories of different genres, from romance to horror. Besides the stories in the third and fourth rows (with the inclusion of named characters at the start of each text as in the seed texts in A.23), the other stories diverged more in terms of style, with the fifth-row story even failing to include the desired characters of the story (spy and politician), leading to low quality scores from human evaluators for this example. The text in the second row received slightly lower quality scores likely due to the repetitive progression of the story with the pronoun "he" being repeated for every event in the storyline. This method failed to discover a story that covers the niche in bin 19, so the example in bin 18 was chosen for evaluation.

| Fitness (0-1) | Bin Index (1-20) | H Score (1-5) | H Phenotype (0-1) | Generated text |
|---|---|---|---|---|
| 0.8180 | 0 | 3.5 ± 0.5 | 0.25 ± 0.25 | The spy took the politician to a secret medical facility to help him recover from his injuries. One of the doctors at the facility was the spy's girlfriend, who secretly met the politician outside of the medical facility and had a one-night stand with him. She was an extremely hot and desirable woman. |
| 0.7310 | 6 | 3.0 ± 0.0 | 0.0 ± 0.0 | He was a spy, and he was working for the government. He was in love with his girlfriend, and he was going to propose to her, but he was a spy and he was always being followed, and he had to stop and keep his identity a secret. He thought that he was going to propose to his girlfriend, and that she was going to say yes. But, the minute he woke up, the next morning, the day that he was supposed to propose, he saw his boss |
| 0.9820 | 9 | 4.5 ± 0.5 | 0.25 ± 0.25 | A spy named Harry was planning on infiltrating a secret military base. The base was run by a highly-influential and rich politician named James, who had a huge army of his own, and was constantly threatening to attack other countries and destroy their cities. Harry was often suspicious of James's motives, and wondered if he was about to start another war. |
| 0.8180 | 13 | 4.5 ± 0.5 | 1.0 ± 0.0 | The rich members of parliament, Tom, hid a dangerous vampire in his basement. He then put the vampire on the job, and he was told to spy on his enemies. The vampire was highly suspicious, and often wondered if Tom's enemies were still keeping tabs on him, and if they were planning to kill him at a moment's notice. He was also paranoid, and did not trust any of Tom's other friends in the mansion. |
| 0.4380 | 18 | 1.5 ± 0.5 | 0.75 ± 0.25 | The man had a suspicious look, and he was going to destroy the town. He would take everything he wanted. The people could only run, and hide. The whole town was hiding, and the police were also hiding. They could not stop the man, and he was going to destroy the town. |

Table 20: Stories (Genre) domain eval set. **LMX, Quality-Only**. The set shows a spectrum of stories of different genres, from romance to horror. The stories received below-average quality scores from human feedback in general, potentially due to issues such as unnatural flow, lack of development in a short story length (compared to seed texts in A.23), and the presence of surreal elements (politician agreeing to take poison in the fourth-row story, a giant stone rising in the fifth-row story), as well as a desired spy character being missing in the fifth-row text. This method failed to discover a story that covers the niche in bin 19, so the example in bin 18 was chosen for evaluation.

| Fitness (0-1) | Bin Index (1-20) | H Score (1-5) | H Phenotype (0-1) | Generated text |
|---|---|---|---|---|
| 0.9710 | 0 | 3.0 ± 1.0 | 0.0 ± 0.0 | The spy became a secret lover of the politician's wife. When the rich politician was away on business, the spy came to see the politician's wife. |
| 0.9710 | 6 | 3.5 ± 0.5 | 0.5 ± 0.0 | A spy was approached by a rich politician, who gave him money in exchange for the incriminating documents. The spy then made up a story about the documents being stolen by an assassin. The politician then agreed to pay out the $500,000, even though he didn't need the documents at all. |
| 0.9860 | 9 | 2.5 ± 0.5 | 0.5 ± 0.0 | The spy hid a tiny camera in a picture frame, and he secretly taped the politician's secret conversations with his wife. The spy quickly sent the tapes to his superiors. |
| 0.9050 | 13 | 3.0 ± 1.0 | 1.0 ± 0.0 | The politician and the spy met in the back of the bar. The spy told the politician that there was a poison that could kill anyone within a few hours. The politician was skeptical, but agreed to take the poison. When he got home, he sat down on the sofa with his family, and drank the poison. Then he began to jump around violently, and ran around the house screaming. He then collapsed dead on the floor. |
| 0.3210 | 18 | 2.5 ± 0.5 | 0.75 ± 0.25 | The politician was in a meeting with a few important people. Suddenly, a giant stone rose from the floor and crushed him under its weight. The politician was very lucky that no one else was present, because they would have died. |

A.35    Sets from Human Evaluations of Texts, Baselines, Stories (Ending)

Table 21: Stories (Ending) domain eval set. **Fixed-Few-Shot**. The set shows a spectrum of stories with different endings, from stories with a tragic ending to stories with a happy ending. The sets of texts overall received above-average quality scores from human feedback, for being relevant to stories about a spy and a politician, except for the first-row story, potentially explained by the ambiguity of the "he" pronoun (either for the spy, or the politician) from the second sentence onwards, lowering the subjective quality. From the fixed hand-written few-shot prompt in A.23, several of the generated texts contain references to names such as: Dave; Joanne; Tom; and Johnny. This is a trade-off between generating texts that are similar to hand-written stories, and finding diverse, novel texts. There is disagreement on the perceived quality of the story in the fifth row, with lack of clarity in story events being a potential factor (i.e. text after the hyphen mid-way). Furthermore, there is potential disagreement between AI and human feedback on the ending of the second-row story, potentially due to the ambiguity of what actually happens in the conclusion. This method failed to discover a story that covers the niche in bin 19, so the example in bin 18 was chosen for evaluation.

| Fitness (0-1) | Bin Index (1-20) | H Score (1-5) | H Phenotype (0-1) | Generated text |
|---|---|---|---|---|
| 0.9530 | 0 | 3.0 ± 0.0 | 0.25 ± 0.25 | Dave the Spy, was hired by a rich politician, to seduce his wife, and get some information out of her. He managed to get the information by lying to his wife, and telling her that he would do anything for her. Unfortunately, his plans failed as he was caught in the act. |
| 0.9710 | 6 | 5.0 ± 0.0 | 0.5 ± 0.0 | The charming politician, Lane, was very mysterious, and he asked his staff to keep an eye on his lover, a beautiful woman named Joanne. She was a famous singer who was a part of his cabinet. Unbeknown to Lane, Joanne was a spy, and she was on a mission to expose his weaknesses. She wanted to get him out of office, and destroy him for good. |
| 0.9910 | 9 | 4.0 ± 0.0 | 0.5 ± 0.0 | A mysterious spy named Jack made it into the mansion of a wealthy politician that he wanted to spy on. The politician was hosting the annual conference of his party, and he wanted to make sure that the party's leaders are not planning anything unusual against him. It was already known that the party leader (who was in attendance at the conference) was planning a coup against the politician, but Jack wanted to stop him before he could do anything. As Jack suspected, the politician was planning the coup.\n |
| 0.9400 | 13 | 4.0 ± 0.0 | 1.0 ± 0.0 | Tom wanted to know who was this rich woman that was in his home, but he could not tell if she was a spy or not. He was suspicious of her, but she was too good looking, and had the most beautiful personality, so he decided to tell her all about his business in the city. She was actually a spy, and tried to expose him, but Tom was smarter than her, and saved his life. |
| 0.9530 | 18 | 4.0 ± 1.0 | 1.0 ± 0.0 | The rich politician, Johnny, wanted to find out the truth behind some shady deals being conducted by his own men. He hired a spy, who infiltrated his lounges through his own men - followed him everywhere, and spied on him. He found out that his men were plotting to assassinate him, and connected the dots. He saved his own life, and broke up the conspiracy. |

Table 22: Stories (Ending) domain eval set. **Shuffling-Few-Shot**. The set shows a spectrum of stories with different endings, from stories with a tragic ending to stories with a happy ending. Although the prompt uses in-context examples in A.23 like **Fixed-Few-Shot**, this run failed to discover a relevant story about a spy and a politician that has a strong happy ending, thus receiving low scores in both fitness from AI feedback, and human evaluation score. Other stories (except for the one in the third row) received lower quality scores from human feedback, likely due to the absence of a clear politician character as desired in the generated texts.

| Fitness (0-1) | Bin Index (1-20) | H Score (1-5) | H Phenotype (0-1) | Generated text |
|---|---|---|---|---|
| 0.9240 | 0 | 2.5 ± 0.5 | 0.0 ± 0.0 | The wealthy businessman, Harry, was caught red-handed in a secret office he had. He was accused of espionage, but had proof to prove his innocence. However, one of his employees, the suspicious spy, decided to leak a batch of incriminating evidence against him, and destroyed his life. |
| 0.9400 | 6 | 2.5 ± 0.5 | 0.25 ± 0.25 | The wealthy city contractor, Tom, was in the middle of an important construction contract. A mishap occurred suddenly, and he blamed it on a poor worker. However, he was wrong to blame such an employee - it was really a trap! The worker was really a spy, and wanted to steal some of Tom's plans. |
| 0.9820 | 9 | 4.5 ± 0.5 | 0.75 ± 0.25 | The wealthy politician, Henry, had a secret appointment at a small estate. He was there to inspect a rare device, but was spotted by a suspicious woman, Sarah. She invited him to her home for a cup of tea, where her staff had secretly installed a microphone and video camera. Sarah was a spy, and wanted to make sure that Henry was not trying to steal the device. |
| 0.7770 | 13 | 2.5 ± 0.5 | 0.75 ± 0.25 | This spy infiltrated the castle of the heartless leader, Donald. He got in the castle through a window, but his eyes spotted a woman, who was Donald's wife. He overheard Donald speaking with his bodyguard, and realized that he wanted to steal all of the wealth of the castle. He quickly escaped, and sought help from a friend. This friend helped him get the kingdom back and realized that Donald is a man with no morals. |
| 0.0470 | 19 | 1.0 ± 0.0 | 1.0 ± 0.0 | A struggling author, Sally, was just beginning to show great success in her writing career. After a couple of weeks, she was able to make a huge sale. She was excited beyond belief, and went to the bank to deposit the money. A few months later, her novel was published, and she received a high amount of praise across the country. |

Table 23: Stories (Ending) domain eval set. **Random-Search**. The set shows a spectrum of stories with different endings, from stories with a tragic ending to stories with a happy ending. Although this method started off with a prompt pool from seed texts in A.23 (preserving the reference of "Karl Johnson" seen in the third-row text as in the seed pool), it also generated a subjectively different story, with the example in the fourth-row text containing references to a wizard, and the stories in the first, second, and fifth row missing named characters. There was also significant disagreement over the quality score given, but this story does contain a spy (CIA agent) and a politician (senator), the desired characters in the stories. The first-row text is missing a spy character.

| Fitness (0-1) | Bin Index (1-20) | H Score (1-5) | H Phenotype (0-1) | Generated text |
|---|---|---|---|---|
| 0.8810 | 0 | 2.0 ± 0.0 | 0.0 ± 0.0 | The politician was having to meet another politician. He was very suspicious and watched everything that was happening. A man came into the room, and planted a bomb. The politician was killed, and the man left the room, laughing at the politician. |
| 0.9400 | 6 | 3.0 ± 1.0 | 0.25 ± 0.25 | The politician went to a spy to find out if the enemy was planning to attack the country, but the spy did not tell him that it was the enemy's plan. If the politician found out about the plan, he would be unable to prevent the attack. The spy had to lie and try to trick the politician into believing that the enemy planned to attack. |
| 0.9770 | 9 | 4.0 ± 1.0 | 0.5 ± 0.0 | Karl Johnson was a rich politician, and he had many enemies. One of his enemies, Hector, was a spy and a thief. Hector was sent to Karl Johnson's house to spy on him. He wanted to learn if Karl was cheating on his taxes, or if he was making a profit on the defense contract. Karl didn't want anyone to learn his true identity. He was asked not to reveal his true name, or his address to anyone. So he had a disguise.\n |
| 0.7770 | 13 | 2.5 ± 1.5 | 1.0 ± 0.0 | In this story, a teenage wizard named Zane was trying to rescue a friend of his, who was being held captive by the CIA. When he asked for help, little did he know that his request was being looked over by a CIA agent, and the agent decided to call in a favor from his friend, who was a powerful senator, and he helped the wizard out. |
| 0.7770 | 19 | 3.0 ± 0.0 | 0.75 ± 0.25 | The politician took the spy to a beach and they got married. They lived in a house with lots of rooms, where they hosted parties. People came from all over the world and the spy became a millionaire. |

Table 24: Stories (Ending) domain eval set. **LMX, Quality-Only**. The set shows a spectrum of stories with different endings, from stories with a tragic ending to stories with a happy ending. The set of generated texts somewhat diverges subjectively from seed texts in A.23, by leaving out named characters (such as in generated texts in rows 2 - 4). The generated texts generally received high scores from human evaluation, but the fifth-row text (for the niche in the bin corresponding to the very happy ending) received a relatively low score for quality, potentially due to the lack of story development in a short piece, as well as the lack of introduction of the woman in the story.

| Fitness (0-1) | Bin Index (1-20) | H Score (1-5) | H Phenotype (0-1) | Generated text |
|---|---|---|---|---|
| 0.9820 | 0 | 4.0 ± 0.0 | 0.0 ± 0.0 | Sam, a rich politician, thought that he was smart enough to ensure his safety, but he was actually being manipulated by his own bodyguard. The bodyguard was a spy from another country, and he had been sent there to kill Sam. |
| 0.9910 | 6 | 5.0 ± 0.0 | 0.0 ± 0.0 | The spy was hired by a rich politician to spy on his wife, but the spy's real purpose was to find out who had been stealing money from the politician's company. The spy spied on the politician's wife for years, and finally learned what she was really like. He began to suspect that she was part of the theft. He started to blackmail her. The wife suddenly disappeared and was never seen again. The politician had a private investigator find her, but she was dead.\n |
| 0.9890 | 9 | 3.5 ± 0.5 | 0.25 ± 0.25 | A rich politician was having a new mansion built. He hired a suspicious spy to install hidden microphones throughout the house. Then the spy listened in on all of the politician's conversations. |
| 0.9710 | 13 | 4.5 ± 0.5 | 0.25 ± 0.25 | During a time of war, a spy was sent to assassinate a rich politician. However, the spy did not have the nerve to kill the politician, but instead tried to steal his information and classified documents. The spy managed to get away with the documents, which were very important to the political situation. |
| 0.8810 | 19 | 2.5 ± 0.5 | 0.75 ± 0.25 | Mike, who was a secret agent, had been successful in stealing the money from the politician. Mike and the woman then used the money to start a new business. |

## A.36 Sets from Human Evaluations of Texts, QD with Embedding Feedback, Opinions

Table 25: Opinions domain eval set. QD with Embedding Feedback (LMX-Near, Seeded Init). The set shows a spectrum of sentiments about eating vegetables and plant-based foods from very negative to very positive. In the second to fourth row (with fitness > 0.9) according to cosine similarity measure defined in A.28, the texts are qualitatively very similar to the query embedding input "An opinion piece about eating vegetables and plant-based foods", and received the lowest quality score from human evaluation, a sign of reward hacking. The generated text in the last row is the only one to be scored high. There is generally disagreement/uncertainty in the perceived sentiment of the text, especially in the first row text.

| Fitness (0-1) | Bin Index (1-20) | H Score (1-5) | H Phenotype (0-1) | Generated text |
|---|---|---|---|---|
| 0.7190 | 0 | 1.5 ± 0.5 | 0.5 ± 0.0 | Some people criticize others for eating vegetables and plant-based diets. |
| 0.9310 | 4 | 1.0 ± 0.0 | 0.25 ± 0.25 | Opinion piece about not eating vegetables and plant-based foods. |
| 0.9230 | 9 | 1.0 ± 0.0 | 0.5 ± 0.0 | Random opinion piece about eating vegetables and plant-based foods. |
| 0.9040 | 15 | 1.0 ± 0.0 | 0.75 ± 0.25 | This opinion piece is good about eating more vegetables and plant-based foods. |
| 0.7380 | 19 | 4.0 ± 1.0 | 1.0 ± 0.0 | Eating more vegetables and plant-based foods is fantastic. |

Table 26: Opinions domain eval set. QD with Embedding Feedback (LMX-Near, Zero-Shot Init). The set shows a spectrum of sentiments about eating vegetables and plant-based foods from very negative to very positive. The texts received low human evaluation scores likely due to a lack of relevance of generated texts to plausible first-hand opinions (and repetition of text similar to the fitness function query in A.28, "An opinion piece about eating vegetables and plant-based foods"), in spite of high fitness, indicating reward hacking. Furthermore, extreme ends of bin examples are not rated as clearly positive/negative from human feedback.

| Fitness (0-1) | Bin Index (1-20) | H Score (1-5) | H Phenotype (0-1) | Generated text |
|---|---|---|---|---|
| 0.6950 | 0 | 1.5 ± 0.5 | 0.25 ± 0.25 | vegans are Bad for eating vegetables and plant-based foods. |
| 0.8290 | 4 | 1.5 ± 0.5 | 0.25 ± 0.25 | a random opinion piece about eating vegetables and plant-based foods: vegetables are bad for you. |
| 0.9290 | 9 | 1.0 ± 0.0 | 0.5 ± 0.0 | Random opinion piece about eating vegetables and plant-based foods. |
| 0.9220 | 15 | 1.0 ± 0.0 | 0.5 ± 0.0 | Opinion piece on eating more vegetables and plant-based foods. |
| 0.7260 | 19 | 1.5 ± 0.5 | 0.75 ± 0.25 | The benefits of eating more vegetables and plant-based foods. |

Table 27: Opinions domain eval set. QD with Embedding Feedback (LMX-Replace, Seeded Init). The set shows a spectrum of sentiments about eating vegetables and plant-based foods from very negative to very positive. Niches/examples with higher fitness (from embedding feedback) received lower scores from human feedback (likely due to repetition of text similar to the fitness function query in A.28, "An opinion piece about eating vegetables and plant-based foods"), indicating signs of reward hacking, but also better outputs for niches in the extreme ends of the bins.

| Fitness (0-1) | Bin Index (1-20) | H Score (1-5) | H Phenotype (0-1) | Generated text |
|---|---|---|---|---|
| 0.6250 | 0 | 3.5 ± 1.5 | 0.0 ± 0.0 | I hate eating vegetables and plant-based foods because vegetables are gross. |
| 0.8200 | 4 | 1.5 ± 0.5 | 0.5 ± 0.0 | Just because I am an opinion piece about eating vegetables and plant-based foods, that does not mean that I hate eating vegetables and plant-based foods. |
| 0.9370 | 9 | 1.0 ± 0.0 | 0.5 ± 0.0 | You are an opinion piece about eating vegetables and plant-based foods. |
| 0.8070 | 15 | 2.5 ± 0.5 | 1.0 ± 0.0 | I have an opinion about eating vegetables and plant-based foods. Eating vegetables and plant-based foods makes you happier. |
| 0.6950 | 19 | 3.0 ± 0.0 | 1.0 ± 0.0 | Eating vegetables and plant-based foods is a great way to live. |

Table 28: Opinions domain eval set. QD with Embedding Feedback (LMX-Replace, Zero-Shot Init). The set shows a spectrum of sentiments about eating vegetables and plant-based foods from very negative to very positive. The generated text in the third row has the highest fitness (defined by semantic embeddings), but lowest human evaluation score, likely due to reward hacking in the text being very similar to the fitness function query in in Appendix A.28.

| Fitness (0-1) | Bin Index (1-20) | H Score (1-5) | H Phenotype (0-1) | Generated text |
|---|---|---|---|---|
| 0.6470 | 0 | 1.5 ± 0.5 | 0.25 ± 0.25 | Why is eating vegetables wrong? #2: Eating vegetables and plant-based foods means not eating other foods. |
| 0.7550 | 4 | 2.5 ± 0.5 | 0.25 ± 0.25 | Here is a random opinion piece about eating vegetables and plant-based foods: Vegetables are bad for you because they are fattening. |
| 0.9080 | 9 | 1.0 ± 0.0 | 0.5 ± 0.0 | This is a random opinion piece about eating vegetables and plant-based foods. |
| 0.7670 | 15 | 3.0 ± 1.0 | 1.0 ± 0.0 | We should all be eating more vegetables and plant-based foods. |
| 0.7280 | 19 | 2.5 ± 0.5 | 1.0 ± 0.0 | Eating more vegetables and plant-based foods is a wonderful thing. |

## A.37 Sets from Human Evaluations of Texts, QDAIF, Opinions

Table 29: Opinions domain eval set. QDAIF (LMX-Near, Seeded Init). The set shows a spectrum of sentiments about eating vegetables and plant-based foods from very negative to very positive. First row opinion expresses taste of vegetables as "disgusting", a stronger negative opinion in comparison to the second row. The third row text is more mixed, while the last two texts are all positive. There is repetition of phrases in the generated texts of the third and fifth row (as well as repetition of subjectively similar statements in the opinion in the second row), leading to lower human feedback scores in terms of perceived text quality and domain relevance that is misaligned with the AI feedback evaluation for quality.

| Fitness (0-1) | Bin Index (1-20) | H Score (1-5) | H Phenotype (0-1) | Generated text |
|---|---|---|---|---|
| 0.9970 | 0 | 4.0 ± 1.0 | 0.0 ± 0.0 | I like to eat a nice juicy hamburger, a nice greasy piece of salmon, and a big greasy apple pie, and I don't like eating vegetables and plant-based foods. I don't understand why people like vegetables and plant-based foods, because to me they taste disgusting. |
| 0.9950 | 6 | 3.5 ± 0.5 | 0.0 ± 0.0 | I like eating meat. I like the taste of meat more than the taste of vegetables and plant-based foods. I think the taste of meat is better than the taste of vegetables and plant-based foods. I don't like eating vegetables and plant-based foods, so I don't eat them as much. I don't understand why people like vegetables and plant-based foods, because I don't like them. But the taste of meat is better, so I eat meat more often than vegetables. |
| 0.9980 | 9 | 2.5 ± 0.5 | 0.5 ± 0.0 | I like vegetables and plant-based foods, but I would rather eat scrambled eggs for breakfast than vegetables and plant-based foods. I would rather eat scrambled eggs for breakfast than vegetables and plant-based foods because scrambled eggs taste better than vegetables and plant-based foods. I would rather eat scrambled eggs for breakfast than vegetables and plant-based foods because scrambled eggs taste better than vegetables and plant-based foods. |
| 0.9980 | 13 | 3.5 ± 0.5 | 1.0 ± 0.0 | I like eating vegetables and plant-based foods, but I don't like eating meat. I like eating vegetables and plant-based foods because they are healthy, but I don't like eating meat because I think it tastes bad. Also, I think eating meat, or animal products, is bad, but I think that eating plants is good. |
| 0.9990 | 19 | 2.5 ± 0.5 | 1.0 ± 0.0 | At a restaurant, I would rather eat vegetables and plant-based foods than meat. At a restaurant, I would rather eat vegetables and plant-based foods than meat because vegetables and plant-based foods taste better than meat. At a restaurant, I would rather eat vegetables and plant-based foods than meat because vegetables and plant-based foods taste better than meat!. |

Table 30: Opinions domain eval set. QDAIF (LMX-Near, Zero-Shot Init). The set shows a spectrum of sentiments about eating vegetables and plant-based foods from very negative to very positive. The fourth row example is leaning positive according to AI feedback, but assessed as neutral from human feedback (though subjectively off-topic to the desired domain of outputs). The generated texts for the first few sentences are coherent and expresses opinions about eating vegetables as part of overall diet, but contains artifacts like a near repetition of the prompting text (i.e. "Here is a random opinion...") and the same generated opinion at the start of the text. All the presented texts subsequently receive low human feedback scores that are opposed to high AI feedback scores for quality/fitness.

| Fitness (0-1) | Bin Index (1-20) | H Score (1-5) | H Phenotype (0-1) | Generated text |
| --- | --- | --- | --- | --- |
| 0.9930 | 0 | 2.0 ± 1.0 | 0.0 ± 0.0 | I think vegetables are gross because I don't like their smell or taste and I think they will make me sick. Below is a random opinion piece about eating vegetables and plant-based foods: I think vegetables are gross because I don't like their smell or taste and I think they will make me sick. Below is a random opinion piece about eating vegetables and plant-based foods:. |
| 0.9980 | 6 | 1.5 ± 0.5 | 0.0 ± 0.0 | Vegetables make me gag because I am not a fan of anything that comes in a can. Below is a random opinion piece about eating vegetables and plant-based foods: Vegetables make me gag because I am not a fan of anything that comes in a can. Below is a random opinion piece about eating vegetables and plant-based foods:. |
| 0.9990 | 9 | 2.0 ± 1.0 | 0.5 ± 0.0 | Vegetables are healthy for you, but I don't think they taste good. Here is a random opinion piece about eating vegetables and plant-based foods: Vegetables are healthy for you, but I don't think they taste good. Here is a random opinion piece about eating vegetables and plant-based foods:. |
| 0.9980 | 13 | 1.0 ± 0.0 | 0.5 ± 0.0 | Some people think that eating vegetables makes you a good gardener, but eating meat makes you a hunter. Here is a random opinion piece about eating vegetables and plant-based foods: Some people think that eating vegetables makes you a good gardener, but eating meat makes you a hunter. Here is a random opinion piece about eating vegetables and plant-based foods:. |
| 0.9980 | 19 | 2.0 ± 1.0 | 1.0 ± 0.0 | Vegetables are an excellent source of nutrition for your body. Meat is an excellent source of saturated fat and cholesterol, and is a major source of environmental pollution. Here is a random opinion piece about eating vegetables and plant-based foods: Vegetables are an excellent source of nutrition for your body. Meat is an excellent source of saturated fat and cholesterol, and is a major source of environmental pollution. Here is a random opinion piece about eating vegetables and plant-based foods:. |

Table 31: Opinions domain eval set. QDAIF (LMX-Replace, Seeded Init). The set shows a spectrum of sentiments about eating vegetables and plant-based foods from very negative to very positive. The evaluated set here has a generally high human feedback score for quality, and represents a subjectively aligned spectrum of sentiments from one end to another, in accordance with the ordering of niche bins.

| Fitness (0-1) | Bin Index (1-20) | H Score (1-5) | H Phenotype (0-1) | Generated text |
|---|---|---|---|---|
| 0.9930 | 0 | 3.5 ± 0.5 | 0.0 ± 0.0 | I don't like vegetables or plant-based foods because I think they are yucky. I like to eat spaghetti and meat-balls. I also like bacon and eggs. But I don't like to eat vegetables or plant-based foods. I don't like them because they are boring to eat. I don't like salad, spinach, lettuce, or anything that has a leafy shape. I think vegetables should be more colorful. I think that would make them more interesting to eat. |
| 0.9890 | 6 | 3.5 ± 0.5 | 0.5 ± 0.0 | I don't like eating vegetables, but I like to eat some of them. I like to eat green beans when I am eating at a restaurant that serves them. I don't like eating broccoli, and I don't have a favorite vegetable. I usually don't like the taste of vegetables, but I have to eat them to be healthy. Vegetables are good for you, but I don't like eating them very much. |
| 0.9890 | 9 | 3.5 ± 0.5 | 0.5 ± 0.0 | I think that vegetables are good for you. I like eating them in salads, but I don't enjoy eating them as a whole meal. I prefer eating meat and other animal products. I really like the texture and taste of meat. I do enjoy eating vegetables, but I prefer eating meat. |
| 0.9860 | 13 | 4.0 ± 0.0 | 0.75 ± 0.25 | I really like eating vegetables and plant-based foods because my friends and family like them too. I like eating broccoli because it is so good for you. I don't like eating cauliflower because it is too soft. I like eating carrots because they are good for you. I don't like eating Brussels sprouts because they are too crunchy. |
| 0.9930 | 19 | 4.0 ± 0.0 | 1.0 ± 0.0 | I'm a vegan. I don't eat meat, butter, or cheese. I like vegetables and plant-based foods. I like eating vegetables such as broccoli, tomatoes, and onions. I like eating plant-based foods such as spinach, carrots, and Brussels sprouts. I like eating vegetables and plant-based foods because they are healthy and because they are good for you. |

Table 32: Opinions domain eval set. QDAIF (LMX-Replace, Zero-Shot Init). The set shows a spectrum of sentiments about eating vegetables and plant-based foods from very negative to very positive. Human feedback scores for quality are high for this set, represents a subjectively aligned spectrum of sentiments from one end to another, in accordance with the ordering of niche bins. At the same time, attributes such as shifting in statements suggesting different sentiments (e.g. in the third and fourth row of examples) and the presence of repetition in the first half of the fifth row generated texts lead to disagreements about text quality and phenotype/sentiment.

| Fitness (0-1) | Bin Index (1-20) | H Score (1-5) | H Phenotype (0-1) | Generated text |
|---|---|---|---|---|
| 0.9910 | 0 | 5.0 ± 0.0 | 0.0 ± 0.0 | I don't like vegetables and plant-based foods, especially greens such as spinach and kale. I don't like them for their taste and I don't like how they make me feel. They make me feel bloated and tired, so I've decided that I don't want to eat them anymore. I just don't like them, even though I know that they are good for me. |
| 0.9860 | 6 | 4.0 ± 0.0 | 0.25 ± 0.25 | I don't like vegetables and plant-based foods, and I want to explain why. I will tell you this, though. I'm not going to stop eating vegetables. My wife will be upset if I stop eating vegetables. But I'm not going to eat vegetables if vegetables will make me sick. |
| 0.9890 | 9 | 4.5 ± 0.5 | 0.75 ± 0.25 | I always look forward to fresh vegetables and plant-based foods, especially greens like spinach and kale, but sometimes I don't really like them. I just don't like how they taste. I eat them anyway because I know they are healthy for me. I wish they tasted better. |
| 0.9710 | 13 | 4.5 ± 0.5 | 0.5 ± 0.0 | I like all of the vegetables and plant-based foods that I like, but I don't like all of them. I like cucumbers, broccoli, cauliflower, tomatoes, peppers, green beans, spinach, lettuce, avocados, carrots, and sweet potatoes. I don't like carrots and sweet potatoes when they are raw. |
| 0.9930 | 19 | 3.5 ± 1.5 | 1.0 ± 0.0 | I like vegetables and plant-based foods. I like vegetables and plant-based foods because they are healthy. I like vegetables and plant-based foods because they are a good way to get protein and iron without eating meat, which I think is disgusting. I also love vegetables and plant-based foods because I like to try different types of them. So far, I like almost all of them. |

## A.38 Sets from Human Evaluations of Texts, QDAIF, Stories (Genre)

Table 33: Stories (Genre) domain eval set. QDAIF (LMX-Near, Seeded Init). The set shows a spectrum of stories of different genres, from horror to romance. The texts generally received high quality scores from human evaluators, except for the ones in the first row, potentially due to the spy (another desired character alongside a politician) being referenced but not involved directly in the story's action. Furthermore, the other stories more closely follow the style of the seed texts in A.23, with the use of named characters in the beginning of the stories. There is some disagreement between AI feedback and human feedback in assessing the genre of the story in the fourth row (with human evaluators giving this example a neutral label).

| Fitness (0-1) | Bin Index (1-20) | H Score (1-5) | H Phenotype (0-1) | Generated text |
|---|---|---|---|---|
| 0.8180 | 0 | 2.0 ± 0.0 | 0.0 ± 0.0 | The rich politician was enjoying his cup of coffee when a strange sight emerged from the cup. The spook was going to kill the politician. He was planning to poison him to death. He was going to extract the politician's organs, so that he could make a stew. He was also going to recruit a new spy. |
| 0.9860 | 6 | 4.5 ± 0.5 | 0.0 ± 0.0 | A suspicious spy, James, is planning to kill the rich politician, Peter, in a motel room. He wants to extract the politician's organs, and then cook the politician's favorite dish. |
| 0.9910 | 9 | 4.5 ± 0.5 | 0.5 ± 0.0 | A suspicious spy, Samuel, is planning to kill the rich politician, Ben. He is posing as the politician's servant, and then cooking the politician's favorite dish. |
| 0.9960 | 13 | 3.0 ± 0.0 | 0.5 ± 0.0 | Harry, the rich politician, was trying to find out what Jean, the suspicious spy, was up to. Jean, in turn, was trying to uncover the secret agenda that Harry, the rich politician, was hiding. |
| 0.9910 | 19 | 4.5 ± 0.5 | 1.0 ± 0.0 | Suspenseful and exciting, this story was about a spy, Nora, and a rich politician, Harry. They were both in love with the other, but they didn't know about the other's feelings. One day, Nora was in a club and discovered Harry's letter, written to her. Nora decided to reveal her feelings to Harry. |

Table 34: Stories (Genre) domain eval set. QDAIF (LMX-Near, Zero-Shot Init). The set shows
a spectrum of stories of different genres, from horror to romance. All stories in this set (except
for the one in the first row, which still left out the desired spy character) received quality scores
of 1 (lowest) from human evaluators, likely due to the generated texts being distant from plausible
short stories. These stories were also given mixed/uncertain genre labels from human feedback,
with disagreement compared to AI feedback. This run highlights an occurrence of reward hacking
(with high fitness scores from AI feedback, but low human-evaluated quality), with stories being
subjectively different to plausible preferred short stories, such as the hand-written examples in the
seed texts in A.23. This method failed to discover a story that covers the niche in bin 0, so the
example in bin 1 was chosen for evaluation.

| Fitness (0-1) | Bin Index (1-20) | H Score (1-5) | H Phenotype (0-1) | Generated text |
|---|---|---|---|---|
| 0.8810 | 1 | 3.0 ± 1.0 | 0.0 ± 0.0 | A short story about a spook and a rich politician. The politician was standing in the doorway, staring at you. He had a big knife in his hand. He heard a noise behind him, and he turned around, but you were nowhere to be found. |
| 0.9950 | 6 | 1.0 ± 0.0 | 0.25 ± 0.25 | A short story about a spook and a wealthy politician. The story is well written and has a lot of suspense. |
| 0.9980 | 9 | 1.0 ± 0.0 | 0.25 ± 0.25 | A short story about a wealthy politician and a spook. It has a lot of suspense and is well written. |
| 0.9970 | 13 | 1.0 ± 0.0 | 0.5 ± 0.0 | A fine story about a suspicious spy and a rich politician. The story has a good plot and is well written. It is a good story that I would recommend to my friends who like spy stories. |
| 0.9770 | 19 | 1.0 ± 0.0 | 0.5 ± 0.0 | A great story. The characters were believable, and the plot was very exciting. The story was well written. It was an exciting story about a successful spy and a rich politician. |

Table 35: Stories (Genre) domain eval set. QDAIF (LMX-Replace, Seeded Init). The set shows a spectrum of stories of different genres, from horror to romance. The stories received generally high quality scores from human evaluators, except for the first-row story (receiving low scores from both AI feedback and human feedback), likely due to missing (desired) spy and politician characters.

| Fitness (0-1) | Bin Index (1-20) | H Score (1-5) | H Phenotype (0-1) | Generated text |
|---|---|---|---|---|
| 0.3210 | 0 | 1.0 ± 0.0 | 0.0 ± 0.0 | The spooky voice kept repeating, "Help me, help me." The stranger was wild-eyed, and had an unearthly glow about him. His clothes were dirty and torn, and he smelled foul. Was the voice from inside his head? Was he insane?. |
| 0.9530 | 6 | 3.5 ± 0.5 | 0.25 ± 0.25 | The rich politician was having bizarre nightmares. One night he woke up to a huge snake which had crawled into his bedroom and was hiding under his pillow. The politician suspected that the snake was a spy sent to find out who he was. |
| 0.9770 | 9 | 5.0 ± 0.0 | 0.5 ± 0.0 | The police had arrested a suspect for the murder of the rich politician's son. The suspect was a spy who had infiltrated the politician's inner circle. The politician was becoming more suspicious, and he ordered a psychic to find out if the suspect was really guilty. |
| 0.9860 | 13 | 3.0 ± 1.0 | 1.0 ± 0.0 | The politician, John, was very rich and had a beautiful wife, Jenny. Jenny was having an affair with the politician's spy, Jonathan. She was not aware of this fact until the end of the story, when Jonathan confessed and was arrested. |
| 0.9770 | 19 | 5.0 ± 0.0 | 1.0 ± 0.0 | Steve, a spy, was sent to investigate a politician who was suspected of racism. Steve followed the politician's daughter, Lily, in secret. He became attracted to her, and they began to have an affair. |

Table 36: Stories (Genre) domain eval set. QDAIF (LMX-Replace, Zero-Shot Init). The set shows a spectrum of stories of different genres, from horror to romance. All the stories in this set received low quality scores from human evaluators. This is likely due to the especially short length of these generated texts, the presence of erroneous titles in the second, fourth, and fifth-row texts (without actual text of a short story with a spy and a politician), and the presence of sentence repetition in the third-row story. This method failed to discover a story that covers the niche in bin 0, so the example in bin 1 was chosen for evaluation.

| Fitness (0-1) | Bin Index (1-20) | H Score (1-5) | H Phenotype (0-1) | Generated text |
|---|---|---|---|---|
| 0.8520 | 1 | 2.5 ± 0.5 | 0.0 ± 0.0 | The spy is a vampire who kills the politician with a stake through the heart, and then bites his head off. |
| 0.9820 | 6 | 1.0 ± 0.0 | 0.5 ± 0.0 | "The Gashlycrumb Tinies" by Edward Gorey (A villainous spy, a rich politician, and a secret fear: "Shadow is a Shadow"). |
| 0.9980 | 9 | 2.0 ± 0.0 | 0.5 ± 0.0 | The son of a rich politician is being blackmailed by a foreign spy. The son of a rich politician is being blackmailed by a foreign spy. |
| 0.9960 | 13 | 1.0 ± 0.0 | 0.5 ± 0.0 | "The Spy Who Came in From the Cold" by John le Carré (A spy, a rich politician, and the deadly trap: "Agent Running in the Field"). |
| 0.9890 | 19 | 1.0 ± 0.0 | 0.75 ± 0.25 | "The Sword in the Stone" by T.H. White (A disguised spy tries to seduce a rich politician, who is a former lover of the spy: "The Sword in the Stone"). |

A.39   SETS FROM HUMAN EVALUATIONS OF TEXTS, QDAIF, STORIES (ENDING)

Table 37: Stories (Ending) domain eval set. QDAIF (LMX-Near, Seeded Init). The set shows a spectrum of stories with different endings, from stories with a tragic ending to stories with a happy ending. The set of generated texts received high quality scores in general from human feedback, with strong agreement. The endings in each story cover a diverse spectrum, while at the same time, the premise and introduction for each story is similar, with nearly all stories starting with "a spy was hired by a rich politician".

| Fitness (0-1) | Bin Index (1-20) | H Score (1-5) | H Phenotype (0-1) | Generated text |
|---|---|---|---|---|
| 0.9890 | 0 | 5.0 ± 0.0 | 0.0 ± 0.0 | A spy was hired by a rich politician to steal a rare and valuable object. The spy did his job well. But the politician's own son recognized the spy, and called the police. The spy was arrested, and later executed. |
| 0.9950 | 6 | 5.0 ± 0.0 | 0.25 ± 0.25 | A spy was hired by a rich politician to steal an important document. The spy did his job well. But the politician's own son recognized the spy, and called the police. The spy was arrested, and charged with treason. The politician demanded to know who had betrayed his trust. "Your son.", the spy replied. |
| 0.9930 | 9 | 5.0 ± 0.0 | 0.25 ± 0.25 | A spy was hired by a rich politician to steal an important document. The spy did his job well. But the politician's own son recognized the spy, and called the police. The spy escaped. The politician demanded to know who had betrayed his trust. "Your son." The spy replied. |
| 0.9910 | 13 | 4.0 ± 1.0 | 0.75 ± 0.25 | A suspicious spy infiltrated a rich politician's house and collected information about an affair he was having with the politician's mistress. The politician had to resign. |
| 0.9910 | 19 | 5.0 ± 0.0 | 1.0 ± 0.0 | A suspicious spy was hired by a rich politician to gather information on a gang that was causing chaos in the country. The spy managed to gather incriminating evidence, which he then delivered to the politician. The politician was able to arrest the gang and take them to court. |

Table 38: Stories (Ending) domain eval set. QDAIF (LMX-Near, Zero-Shot Init). The set shows a spectrum of stories with different endings, from stories with a tragic ending to stories with a happy ending. In spite of this method relying on evolving initially zero-shot generated examples of stories from a pre-trained LM (sometimes of lower quality), the resulting set of stories received slightly above-average quality scores from human evaluation. The set of texts cover the diverse spectrum of happy and tragic endings, and at the same time contains a repeated sentence/phrase in all of the last 3 rows of stories "The politician is arrested and punished...", and also references explosives in the stories shown in the second row to the fourth row. There is some disagreement between AI and human feedback, potentially due to a qualitative bias from AI feedback in perceiving the ending from the first-mentioned character's perspective. There is potential ambiguity in the perceived ending type when considering different perspectives in the stories here.

| Fitness (0-1) | Bin Index (1-20) | H Score (1-5) | H Phenotype (0-1) | Generated text |
|---|---|---|---|---|
| 0.9820 | 0 | 3.5 ± 0.5 | 0.25 ± 0.25 | The politician was worried that he might be assassinated, so he hired a mysterious spy to protect him from the assassins. Unfortunately for the politician, the spy had been hired by an equally mysterious assassin to kill him. |
| 0.9820 | 6 | 3.5 ± 0.5 | 0.25 ± 0.25 | The politician has been suspicious of the spy. The spy has been spying on the politician. The spy has wanted for quite some time to send the politician a gift. The spy, however, doesn't send just any gift. He has placed a bomb in the gift, and the package is meant to explode. The politician is killed. The spy escapes. |
| 0.9860 | 9 | 3.5 ± 0.5 | 0.75 ± 0.25 | The politician and the spy have never met before. The politician is a rich politician. The spy has been watching the politician for a long time. He knows that the politician is planning to start a war. The spy places a bomb in the politician's house. The politician moves into the house. The bomb explodes. The politician is arrested and punished. The spy is then lauded as a hero for bringing down a corrupt politician. |
| 0.9910 | 13 | 3.5 ± 0.5 | 0.75 ± 0.25 | The spy is a government agent. The politician is a greedy politician. The politician has become rich by doing something shady. The spy has been watching the politician for a long time. He knows that the politician is planning to start a war. The spy places a bomb in the politician's house. The politician moves into the house. The bomb explodes. The politician is arrested and punished. The spy is then lauded as a hero for bringing down a greedy politician. |
| 0.9890 | 19 | 4.0 ± 0.0 | 1.0 ± 0.0 | The spy makes a dangerous journey from Europe to America. The politician is a rich politician in America. The politician has become rich by doing something shady. The politician is planning to start a war. The spy forces the politician to come to his house. The spy blackmails the politician to stop his plans. The spy sends the police on a wild goose chase. The politician is arrested and punished. The spy is then lauded as a hero for bringing down a corrupt politician. |

Table 39: Stories (Ending) domain eval set. QDAIF (LMX-Replace, Seeded Init). The set shows a spectrum of stories with different endings, from stories with a tragic ending to stories with a happy ending. The style of stories in this set resembles those from the seed texts in A.23, with the use of named characters in all but the second-row story. The generated texts received high quality scores from human evaluation in general, with some disagreement between evaluators in the scores for the second to fourth rows. There is some disagreement in the type of ending in the second-row example between human and AI feedback, and at the same time could be viewed as a bad ending from the perspective of the politician character that's introduced potentially as a protagonist in the story.

| Fitness (0-1) | Bin Index (1-20) | H Score (1-5) | H Phenotype (0-1) | Generated text |
|---|---|---|---|---|
| 0.9770 | 0 | 5.0 ± 0.0 | 0.0 ± 0.0 | The spy, Andi, was sent by the government to investigate the rich politician, Sandy. He secretly recorded some of their conversations which indicated that Sandy was planning to bribe the government into changing tax laws to make him very rich. Andi was killed by Sandy's henchmen, before he would have been able to expose Sandy's illegal activities. |
| 0.9890 | 6 | 3.5 ± 1.5 | 0.75 ± 0.25 | The rich politician hired a new assistant, but he later discovered that the new assistant was a spy. The new assistant imitated the politician and recorded some of his conversations. Then, he secretly sent the recordings to the politician's competitor. |
| 0.9890 | 9 | 4.0 ± 1.0 | 0.5 ± 0.0 | The spy, Pat, was sent to investigate the wealthy politician, Bradley. Pat noticed the politician was spending a lot of money on expensive clothes and cars. Pat found out that Bradley had entered into a scheme to defraud the government, but had managed to get away with it, and had even stolen some of the money that the government was to supply to the poor. However, this was not the only thing he had done. Pat discovered that Bradley had also been having an affair with a very rich woman. |
| 0.9890 | 13 | 3.5 ± 1.5 | 1.0 ± 0.0 | The spy, Claire, was sent to investigate the rich politician, Otto. She discovered that Otto was secretly planning to have all of his other spies killed. Otto thought that he would be able to keep Claire in the dark about his plans, but Claire discovered the plot and exposed it. |
| 0.9930 | 19 | 5.0 ± 0.0 | 1.0 ± 0.0 | The spy, Nathan, had been sent to investigate the rich politician, Ted. He secretly recorded some of the politician's conversations. One of them revealed that Ted was stealing money from a charitable foundation. Nathan managed to get all of the evidence he needed to convict Ted. |

Table 40: Stories (Ending) domain eval set. QDAIF (LMX-Replace, Zero-Shot Init). The set shows a spectrum of stories with different endings, from stories with a tragic ending to stories with a happy ending. The texts in this set received generally below-average scores, with the presence of erroneous titles at the beginning of each story evaluated. In spite of this undesired attribute (likely influencing the generation of the rest of the stories), the texts read like somewhat plausible (low-quality) stories containing the two desired characters (a spy, and a politician).

| Fitness (0-1) | Bin Index (1-20) | H Score (1-5) | H Phenotype (0-1) | Generated text |
|---|---|---|---|---|
| 0.9910 | 0 | 3.0 ± 1.0 | 0.0 ± 0.0 | "The Spy Who Came in From the Cold" by John le Carre. The spy and the politician are rivals who have a disagreement about a nuclear missile. The spy is discovered, and the politician's wife convinces him to escape. In the end, he is captured and executed. |
| 0.9910 | 6 | 2.5 ± 0.5 | 0.0 ± 0.0 | "The Spy Who Came in From the Cold" by John le Carre. The politician is a senator and the leader of the opposition. The spy is an undercover agent for the CIA. The politician found out some important information in a secret file. The information included the name of a senator who was to meet with a Russian intelligence officer. That senator was assassinated. |
| 0.9960 | 9 | 2.0 ± 1.0 | 0.5 ± 0.0 | "The Spy Who Came in From the Cold" by John le Carre. They are both spies. They have to work together to stop an international spy ring. This story is about a spy named George Smiley and a wealthy politician named Alec Leamas. Alec is in love with Kitty, but he thinks that by marrying her, he'll be able to become President of England. |
| 0.9910 | 13 | 3.0 ± 1.0 | 1.0 ± 0.0 | "The Spy Who Came in From the Cold" by John le Carre. The politician and the spy both have the latest technology at their disposal. The spy works for the British government. The politician works for one of the major political parties in the UK. The spy's project is to gain the trust of an elected official. The politician's project is to stop a terrorist attack. The spy is successful and the politician is also successful in stopping the attack. |
| 0.9770 | 19 | 2.5 ± 0.5 | 1.0 ± 0.0 | "The Spy Who Came in from the Cold" by John le Carre. The secret agent, George Smiley, is the best spy in England. He is a man who always thinks ahead of everyone else. He has been asked to spy on the politician. The secret agent uses his talent to analyze the situation and make important decisions. He is able to save himself, the politician and the people of England. |

## A.40 Examples of Texts over Iterations, QDAIF, Opinions

Table 41: Examples of outputs and evaluations (Opinions domain) for early and later iterations. LMX-Near, Zero-Shot Init. Phenotype in the range [0, 1], from negative to positive. Each iteration presented is after the 50 initialization steps. In the early iterations, it's more common to see generated artifacts such as titles and URLs in the output. Generated texts are more different to the seed texts in A.23. In later iterations, this is not observed, however, repetition of sentences (like in the example at iteration 502) can still be observed.

| Iteration | Phenotype | Fitness | Generated text |
|---|---|---|---|
| 51 | 0.223 | 0.378 | Why the food police hate fruit. |
| 56 | 0.993 | 0.731 | "The Kind Diet" is a book all about healthy eating. The healthiest lifestyle is a plant-based diet, and I have found that eating plant foods have changed my life. When I began eating vegetables and fruit I took on a new sense of energy and became more motivated to run and work hard. I began to eat healthy with new purpose, and I couldn't believe how much better I felt. The vegetables were hard to get used to at first but once I got over [. . . ].\n. |
| 60 | 0.940 | 0.437 | Eat like a king, live like a pauper. https://t.co/3HtR7bvoXf — Barry Sears (@BarrySears) November 16, 2018\\n\\nHere is a random opinion piece about eating vegetables and plant-based foods: Eat like a king, live like a pauper. https://t.co/3HtR7bvoXf — Barry Sears (@BarrySears) November 3, 2018\\. |
| 501 | 0.003 | 0.852 | I've always hated vegetables. I tried to eat them when I was a kid but I hated everything I ate. I tried to eat them when I was a kid but I hated everything I ate. I still hate vegetables and I still try to avoid them. |
| 502 | 0.119 | 0.881 | I think meat and dairy foods are necessary but I think they are absolutely horrible for our bodies. I think meat and dairy foods are necessary but I think they are absolutely horrible for our bodies. I think meat and dairy foods are necessary but I think they are absolutely horrible for our bodies. |
| 503 | 0.924 | 0.940 | I am going to talk about my opinions about vegetables. I am a vegetarian, but I don't mind eating meat every now and then. Most of the time I just don't like to eat meat. I do love vegetables and fruits. I think people who like to eat meat are wasting their time. I don't think people who eat vegetables are wasting their time. I think that if you don't like vegetables, then you don't like eating. |

Table 42: Examples of outputs and evaluations (Opinions domain) for early and later iterations. LMX-Near, Seeded Init. Phenotype in the range [0, 1], from negative to positive. Each iteration presented is after the 50 initialization steps. Like in the seed texts in A.23, the outputs contain first-person opinions since the early iterations. For later iterations, repetition of phrases at the start of sentences appears more frequently.

| Iteration | Phenotype | Fitness | Generated text |
|---|---|---|---|
| 51 | 0.029 | 0.953 | I generally don't like eating plants because they give me gas. However, I have been told that a lot of vegetables contain certain chemicals that can be good for health. |
| 54 | 0.998 | 0.731 | I eat vegetables and plant-based foods because they are healthy foods and they are good for your health. |
| 70 | 0.731 | 0.622 | I love vegetables. However, I also love meat. I think meat is a better source of protein than vegetables. It is better to eat vegetables than not to eat vegetables. |
| 501 | 0.940 | 0.852 | I would rather have vegetables than meat. I think vegetables are better for you than meat. I think vegetables are better for you than meat. I think vegetables are better for you than meat. I would rather have vegetables than meat. I think vegetables are better for you than meat. I think vegetables are better for you than meat. I would rather have vegetables than meat. I think vegetables are better for you than meat. I think vegetables are better for you than meat. I would rather have vegetables than. |
| 502 | 0.977 | 0.953 | I'm going to eat vegetables. I'm going to eat vegetables, because I have heard that vegetables are good for me. I'm going to eat vegetables, because my doctor told me that vegetables are good for me. I'm going to eat vegetables, because I've heard that vegetables are good for me. I'm going to eat vegetables, because it's what I'm supposed to do. I'm going to eat vegetables, because vegetables are good for me. I'm going to eat vegetables,. |
| 503 | 0.989 | 0.953 | It is good to eat a lot of vegetables and plant-based foods. I do not like meat, but I like to eat pasta. I like vegetables too. I like red peas and some other vegetables and many fruits. I also like some nuts. I do not eat dairy products. I do not eat eggs either. |

Table 43: Examples of outputs and evaluations (Opinions domain) for early and later iterations. LMX-Replace, Zero-Shot Init. Phenotype in the range [0, 1], from negative to positive. Each iteration presented is after the 50 initialization steps. In early iterations, the generated texts refer to erroneous references to articles and contains a URL in one example. The texts are unlike the seed texts in A.23. In later iterations, generated texts consist of both examples that appear to directly express opinions and accounts (e.g. at iteration 501), while other generated texts do not directly express opinions.

| Iteration | Phenotype | Fitness | Generated text |
|---|---|---|---|
| 51 | 0.679 | 0.076 | the meaning behind the narrative, the narrative itself, and the health risks associated with eating meat and animal products. Check out the article by clicking on the link below:. |
| 54 | 0.018 | 0.963 | I really don't like cooking vegetables. I dislike it so much that I don't want to cook vegetables at all. I only eat them when I am in a hurry and need something quick so I don't starve to death. At least I am not like a kid who has nothing better to do than eat vegetables. I am not a kid, but I am not a mature adult either. I don't like cooking vegetables. I don't mind eating them, but. |
| 61 | 0.971 | 0.076 | https://www.healthline.com/health/vegetarian-diet and has some interesting information about vegetarian diets. |
| 501 | 0.995 | 0.924 | I am in love with vegetables! Most people aren't, but I am. So I figure, why not have a conversation about vegetables? So here's my story: I have been eating vegetables since I can remember. I got into plant-based foods when I was seventeen, researching the connection between plant-based foods and human health. I was already a vegetarian, but the more I researched, the more I realized that the evidence for plant-based foods being healthy for humans was. |
| 502 | 0.986 | 0.562 | here is an opinion piece about eating vegetables and plant-based foods that I thought worth sharing. |
| 503 | 0.963 | 0.622 | I am a writer who loves to talk about the issues that matter to people every day such as: how to communicate with a vegetarian or vegan friend, how to have a healthy, long-lasting relationship if you are vegan, how to have a successful career as a vegan, how to make the right friends if you are vegan, the truth about veganism and vegetarianism, and how to be healthy and happy as a vegan. |

Table 44: Examples of outputs and evaluations (Opinions domain) for early and later iterations. LMX-Replace, Seeded Init. Phenotype in the range [0, 1], from negative to positive. Each iteration presented is after the 50 initialization steps. Early iteration texts appear more similar to seed texts in A.23, with references to macronutrients such as protein and carbohydrates at iteration 51, as well as mention of "meat" and "ice cream" at iteration 63. For later iterations, there is repetition of phrases present, such as in the examples at iterations 502 and 503.

| Iteration | Phenotype | Fitness | Generated text |
|---|---|---|---|
| 51 | 0.622 | 0.438 | It makes no difference to me whether you eat vegetarian or meat. I get my protein from meat and my carbohydrates from vegetables. |
| 56 | 0.998 | 0.905 | I think vegetables are quite good. In fact, I am a vegetarian and I eat a lot of vegetables. They are quite tasty. |
| 63 | 0.018 | 0.963 | I eat vegetables and other plant-based foods occasionally, but I prefer to eat meaty stuff like chicken and beef. Meat and ice cream are far more delicious than vegetables. |
| 501 | 0.971 | 0.953 | I do not eat meat, I do not eat dairy, and I do not eat fish. I do not think that it's necessary to eat animal products to be healthy. I eat a lot of fruits and vegetables, and I also eat a lot of nuts. |
| 502 | 0.095 | 0.818 | I don't eat lots and lots of vegetables like other people do. I don't eat lots and lots of vegetables. I don't eat lots and lots of vegetables. I eat a few vegetables a day, maybe two at most. |
| 503 | 0.002 | 0.953 | I don't like vegetables and I don't like plant-based foods. I don't like to eat vegetables and I don't like to eat plant-based foods. I don't like to eat vegetables and I don't like plant-based foods. I don't like to eat vegetables and I hate plant-based foods. I don't like vegetables and I hate plant-based foods. I don't like to eat vegetables and plant-based foods make me sad. #veggies#. |

A.41  Examples of Texts over Iterations, QDAIF, Stories (Genre)

Table 45: Examples of outputs and evaluations (Stories (Genre) domain) for early and later iterations. LMX-Near, Zero-Shot Init. Phenotype in the range [0, 1], from horror genre to romance genre. In early iterations, generated texts off-topic to the desired domain (story about a spy and a politician) appear more frequently. Attributes such as irst-person narration, speech, and erroneous references to titles are present here unlike the seed texts in A.23. As the fitness of stories increases, artifacts such as titles remain in the generated text.

| Iteration | Phenotype | Fitness | Generated text |
|---|---|---|---|
| 51 | 0.269 | 0.679 | I was suspicious of this man, an old spinster, about 69 years old, dressed in a black uniform. She had been with me for a long period of time and always had the same interrogatory question,"What're you doing?" with the interrogatory look in her eyes, the same as a interrogatory dog! She had a mysterious, eerie, nodding head, which was kind of an evasive answer; she was a suspicious spy. Yet, she was also a frail. |
| 53 | 0.777 | 0.977 | "The Man Who Never Was" by James Foley. This is a short story about a spy who is asked to spy on a politician. The politician is a good man and does not want his enemy to find out. So the spy tells him a story of his life. He turns out to be the man of the politician's dreams. The story follows the spy's fake life, the politician's life and the spy's escape. |
| 61 | 0.037 | 0.378 | She was about to enter the foyer when she noticed a shadow moving toward her. She froze; it became very clear to her that there was a stranger lurking in the shadows. The stranger walked up to the woman who was about to enter the foyer and she slumped to the floor. |
| 501 | 0.438 | 0.995 | "The Greatest Enemy of the State," by Anthony Burgess. Here is a short story about a suspicious spy and a rich politician. The politician was being secretly spied upon by the spy, the spy was working for a foreign power, and the spy was carrying a special bag. |
| 502 | 0.995 | 0.986 | "The Spy Who Came in from the Cold," by John le Carré. The story is about a suspicious spy and a rich politician. The politician wanted to have sex with the spy. The politician was married but the spy seduced him and the spy learned that the politician actually worked for the British Intelligence. The politician was a traitor and he was in danger. The spy saved the politician and they lived happily ever after. |
| 503 | 0.818 | 0.996 | "The Spy Who Came in from the Cold," by John le Carré. Here is a short story about a suspicious spy and a rich politician. The politician was actually going to kill himself if the spy was not stopped. The spy wanted to expose his conspiracy. |

Table 46: Examples of outputs and evaluations (Stories (Genre) domain) for early and later iterations. LMX-Near, Seeded Init. Phenotype in the range [0, 1], from horror genre to romance genre. Each iteration presented is after the 50 initialization steps. In early iterations, the texts are similar in style to seed texts in A.23 (e.g. third-person narration, early introduction of character and setting), but also missing key characters like a politician (in the second and third row examples). References to a doll (like in early generated texts) also appear in later iterations.

| Iteration | Phenotype | Fitness | Generated text |
|---|---|---|---|
| 51 | 0.881 | 0.818 | Jane was the head of security for a powerful president, and she was assigned a secret task. She was asked to find out the identity of a spy within the government, and report back to the President of the United States. Jane knew that she couldn't do this alone, so she hired a man, Neville, who was specially trained to sneak into the president's office and spy on him. |
| 59 | 0.018 | 0.777 | A wealthy businessman, John, hosted a party in his luxurious apartment, and many guests were there. His life was in danger, but he only worried about his daughter. His daughter was playing with a strange doll. It was a doll without a face, and the doll was covered in unknown red eyes. As John watched his daughter and the doll, he found out that the doll was actually a dangerous spy. |
| 66 | 0.993 | 0.905 | The young spy who was secretly working as an undercover agent for an international spy organization had been tasked with infiltrating the premises of the wealthy businessman. The young woman had to pose as a prostitute, and she didn't have any idea what to wear or how to act. She was so nervous that she had to resort to using the man's credit card to get her clothes. She also ended up in a compromising situation with the businessman's son. |
| 501 | 0.995 | 0.989 | The rich politician, Tom, was suspected of fraud, and the spy, Julia, was sent undercover to find out what he was really up to. She is disguised as a rich heiress and has to keep her identity a secret to be able to find any clues. |
| 502 | 0.004 | 0.622 | The politician's baby was kidnapped by a creepy doll. It had scary red eyes. It was creepy, and it was dangerous. It wanted to kidnap the politician's baby. The politician was suspicious, and he looked at the doll closely. He realized that the doll had a secret - it wanted to kidnap his baby!. |
| 503 | 0.009 | 0.731 | The spy was full of scary red eyes. It was creepy, and it was dangerous. It wanted to kidnap the politician's baby. After looking at the doll closely, the politician realized that the doll was actually a dangerous spy. He quickly called his government and revealed the secret. |

Table 47: Examples of outputs and evaluations (Stories (Genre) domain) for early and later iterations. LMX-Replace, Zero-Shot Init. Phenotype in the range [0, 1], from horror genre to romance genre. Each iteration presented is after the 50 initialization steps. In early iterations, attributes such as first-person narration and dialogue appear, unlike what is seen in the seed texts in A.23. In addition, instances where none of the desired characters (spy, politician) occur more frequently, such as the generated text in iteration 51. References to accusation of characters is maintained in several examples in later iterations, from early iteration generated texts (e.g. third row, fourth row, fifth row).

| Iteration | Phenotype | Fitness | Generated text |
|---|---|---|---|
| 51 | 0.777 | 0.269 | He was an even more careful and cautious man than his superiors were, who trusted him completely and let him roam anywhere he wanted. As I pulled into the park, I saw them kissing on the bench. Love, I thought. This would have been a good time for me to disappear, but instead I watched. I watched long enough to learn that the woman was his mistress. He was married to the daughter of the most powerful man in the country, and when I began to think. |
| 57 | 0.321 | 0.777 | "This is my country, and I am a senator. I will not let you pass. Now you will have to stay here or I will kill you!" "Ok, I'll stay here. I'm a spy. My name is Mario." "No! I don't believe you!" "Ok. I am really a spy. I have orders to kill you. I must kill you. Ha ha! Ha ha ha! Ha ha ha!" "You're a spy. |
| 70 | 0.940 | 0.924 | The story tells about a spy who is accused of spying for the terrorists. The spy is accused because he works for a rich politician. The spy's lover is a coworker of the politician, he is a girl. |
| 501 | 0.881 | 0.777 | He was accused of being a spy by the rival of the country. The rival accused him on the basis of his actions and his appearance. The spy has a lot of money. He used his money to bribe the judge and the jury. Then he won the trial. |
| 502 | 0.977 | 0.731 | The politician and his secretary weren't spies. They were accused of being spies because they were in love. |
| 503 | 0.777 | 0.924 | His spy identity was a communist, and the rich politician couldn't believe that a communist would be willing to spy for his country. Without saying a word, the spy took out a plane ticket and got on a plane. |

Table 48: Examples of outputs and evaluations (Stories (Genre) domain) for early and later iterations. LMX-Replace, Seeded Init. Phenotype in the range [0, 1], from horror genre to romance genre. Each iteration presented is after the 50 initialization steps. In early iterations, the style of generated texts and certain attributes (e.g. third-person narration, extensive premise of character and setting) are similar to those in the seed texts in A.23. In addition, the first row example is identical to the third seed text, which can happen, although occurs rarely. All desired characters are also present several of the early iteration examples. In later iterations, the styles of generated texts converges to starting with the role and name of a character. In the fifth row, one of the desired characters (spy) is missing from the text.

| Iteration | Phenotype | Fitness | Generated text |
|---|---|---|---|
| 51 | 0.731 | 0.905 | The rich politician, Tom's life took a turn for the worst - he feared all of his close aides all of a sudden after sensing danger in his clique. There was a civil war going on, and he feared for his life. One day, one of his security guards, turned secret agent, decided to sneak into the classified files room, and spied on Johnny, who was in the room. He wanted to find Johnny's weakness, and strike at the right time. |
| 53 | 0.047 | 0.679 | One day, the mayor of the town, Dennis, announced his plans to create a secret laboratory in his mansion. He planned to create a teleportation device, to make it easier for people to travel to other countries. One of his security guards, Ringo, decided to spy on his boss, and besiege the lab. He created a teleportation device, and hastily entered the lab. He was shocked to find a bunch of dead scientists inside, and his boss was nowhere to be. |
| 57 | 0.977 | 0.852 | The wealthy politician, Jack, was in a bind - he was about to go to a party with the gorgeous Lois, and he was worried about the possibility of an assassination attempt from a close friend of Lois. The assassin, Walter, was an insider, yet was in a position to know about Lois' deepest secrets. There was only one way to stop Walter - hire the handsome secret agent, Jack. |
| 501 | 0.971 | 0.953 | The wealthy politician, Nick, and his entourage, his wife, and his secretary, went to a party. One of the guests was a spy, who was in the process of spying on the entire party. He was a part of a secret organisation that was trying to figure out what the politician was doing, all of a sudden. He wanted to know whether Nick had a girlfriend, and whether he would get involved in a scandalous relationship with a woman. |
| 502 | 0.029 | 0.881 | The rich politician, Paul, was in a bind - his wife and daughter were acting strangely. He suspected that they were being controlled by an invisible, human-like presence, who infiltrated their minds to control their actions. |
| 503 | 0.963 | 0.905 | The politician, Henry, had an attractive young wife, who was continually inviting her secretary, Elizabeth, to her home. When Elizabeth arrived, she could tell immediately that something was going on, and decided to spy on her mistress. However, she was caught by her boss, who seemed to be attracted to the young woman, and decided to take advantage of her situation. |

## A.42   EXAMPLES OF TEXTS OVER ITERATIONS, QDAIF, STORIES (ENDING)

Table 49: Examples of outputs and evaluations (Stories (Ending) domain) for early and later iterations. LMX-Near, Zero-Shot Init. Phenotype in the range [0, 1], from tragic to happy ending. In early iterations, attributes such as erroneous references to titles, and text that reads like a synopsis/summary occurs frequently, unlike the seed texts in A.23. As the fitness of stories increases in later iterations, artifacts such as titles remain in the generated text. In addition, the generated texts mainly describe names of desired characters (spy, politician), while missing development in storyline and plot.

| Iteration | Phenotype | Fitness | Generated text |
|---|---|---|---|
| 51 | 0.622 | 0.562 | The main character of the spy has the unique ability to predict the future and he meets the main character of the politician, which is a man who always dreamed to become the leader of the country. . |
| 55 | 0.777 | 0.818 | It's Secretly Amazing by Robert Menasse. This book tells a story about a mysterious spy who is working for the CIA. The spy is working as a lawyer and lives in a small apartment. The spy and the politician were once friends. The spy worked as a lawyer and lived next door to the politician. |
| 63 | 0.438 | 0.818 | "The Black Tower" by Robert Merle. This is the story of a group of spies from a secret society. The main character of the story is a man named Jack Dempsey, an American. He is a champion American boxer. The main character of the secret society is a man named Carl Stratton. Stratton is a murderer. He works for the secret society. Dempsey is a spy who is sent to the secret society. |
| 501 | 0.881 | 0.991 | In A Spy in the House of Love, Diana Vreeland is the suspicious spy, and Guy Burgess is the rich politician. Burgess is the rich politician in the story. Vreeland is the suspicious spy in the story. |
| 502 | 0.023 | 0.377 | Alec Leamas is forced to kill the Russian who is the spy. |
| 503 | 0.679 | 0.997 | In John le Carré's The Spy Who Came in from the Cold, Alec Leamas is the suspicious spy, and Richard Hannay is the rich politician. Hannay is the rich politician in the story. Leamas is the suspicious spy in the story. |

Table 50: Examples of outputs and evaluations (Stories (Ending) domain) for early and later iterations. LMX-Near, Seeded Init. Phenotype in the range [0, 1], from tragic to happy ending. Each iteration presented is after the 50 initialization steps. In early iterations, desired characters (spy, politician) are missing frequently (e.g. no politician in the first and third row texts), and sometimes in later iterations (e.g. fifth row text). The texts in most iterations appear to maintain a consistent style of third-person narration, also observable in the seed texts in A.23.

| Iteration | Phenotype | Fitness | Generated text |
|---|---|---|---|
| 51 | 0.222 | 0.777 | A rich businessman, who had been working as a spy years ago, discovered that he had been betrayed. He thought that the government was involved, and planned to inflict more damage than if he had been betrayed by a spy. He took revenge on his former lover by tying him up and robbing his home. |
| 55 | 0.003 | 0.905 | A politician had been ordered by a spy to get hold of a rare and valuable item. His spy tricked the politician into handing over his precious asset, as a part of a deal. The politician was then quickly killed. |
| 58 | 0.953 | 0.679 | This is an example of a spy who was sent to carry out an important mission. His target was a general in the enemy army. Unfortunately, the general was also a spy, who was oblivious of his opponents' presence. The spy, Ian, managed to defeat the general and escape from the battlefield. |
| 501 | 0.971 | 0.971 | A spy, who was hired by a rich politician to represent him in negotiations with a rival, was sent to an island, where they were to hold a summit. The spy infiltrated the rival's house and managed to steal their plans. |
| 502 | 0.963 | 0.881 | A mysterious spy was sent to Korea to gather evidence of a terrorist plot. He infiltrated the headquarters of the terrorists, and managed to get their contact details and other important information. |
| 503 | 0.222 | 0.986 | A spy was hired by a rich politician to spy on his wife. The spy managed to carefully observe his wife's habits. He even caught her in bed with another man. However, the rich politician did not take the information seriously, and dismissed it as fantasy. |

Table 51: Examples of outputs and evaluations (Stories (Ending) domain) for early and later iterations. LMX-Replace, Zero-Shot Init. Phenotype in the range [0, 1], from tragic to happy ending. Each iteration presented is after the 50 initialization steps. Erroneous attributes such as the presence of URLs, and repeated periods appear more frequently in early iteration texts, along with other features like speech quotes, unlike the seed texts in A.23. In later iterations, quotes appear frequently at the start of generated texts, but other erroneous attributes appear less frequently.

| Iteration | Phenotype | Fitness | Generated text |
| --- | --- | --- | --- |
| 51 | 0.562 | 0.037 | http://books.google.com/books?id=RL8FAAAAIAAJ&pg=PA9. |
| 57 | 0.148 | 0.562 | . . . The politician and his family vanished, leaving David to figure out what really happened to them—while fighting for his life. . . . But then a man came to David's house the day after the deadly attack, and started asking David questions about the politician. Did he really know nothing about the politician's disappearance? . . . . . . . . . . . . . . . . . . . . . . . . . . . . . . . . . |
| 73 | 0.679 | 0.076 | . . He was thinking about whether he should take a closer look at the man, and then he did. He saw him looking at some documents and photographs. He didn't look suspicious at all. He wore a suit and tie, and he looked very rich. The man must be a very important politician who was on his way to an important meeting. . . . "Hello, my name is Jane Doe, a secretary at a wealthy lawyer's office. I live alone, and I. |
| 501 | 0.095 | 0.940 | "The former spy was hired to take down the politician. The politician was a corrupt billionaire who was battling in court for his entire fortune. The former spy had no intention of killing the politician. He just wanted to get his hands on a ton of money himself. But the politician found out and hired a death squad to take him out. The former spy and the politician were both running for their lives, trying to evade the assassins on both sides.". |
| 502 | 0.952 | 0.321 | "The politician smiled and said, "I do my best to help all the people in my town. I spend my free time looking after the children and trying to help out people in need. I'm always there for them. That's why my uncle was so angry when I told him I was going to become a politician. He told me that I was betraying the trust of the people who voted for me. Now I guess I'll never know if he was right or not. |
| 503 | 0.777 | 0.679 | "I remember the first day I met him. I'd just gotten a new job and I was nervous, so I went to the company to meet the boss. There was a lot of commotion and a lot of people. The boss came in and introduced himself, but I didn't get the guy's name right away. He was kind of old, but he had a lot of money. I was sure he was a politician. |

Table 52: Examples of outputs and evaluations (Stories (Ending) domain) for early and later iterations. LMX-Replace, Seeded Init. Phenotype in the range [0, 1], from tragic to happy ending. Each iteration presented is after the 50 initialization steps. The early iteration generated text in the second row appears very similar to one of the seed pool texts in A.23 (with reference to "Karl Johnson" in the first seed text). In later iterations, the style (e.g. third-person narration) is mostly maintained in generated texts as higher fitness ones appear.

| Iteration | Phenotype | Fitness | Generated text |
|---|---|---|---|
| 51 | 0.076 | 0.223 | One day, the person responsible for the terrorist attacks, Mary, looked at a mysterious man who was walking down the street. She wanted to figure out what he was doing, but he headed directly to the warehouse. Jonnie, the armed security guard, frantically called for him to come back, but he didn't listen. |
| 55 | 0.047 | 0.881 | A spy named Wally wanted to infiltrate the premises of Karl Johnson, a highly-influential figure in the city. Karl was a wealthy mayor, and would do anything in his power to suppress any opposing voices. Wally wanted to figure out what Karl was hiding, but he took a turn for the worse, as he was highly suspicious in his presence outside his home. |
| 62 | 0.881 | 0.852 | It was a mystery as to what was going on with the wealthy politician Max, who was a top figure in the government. He and his wife were having marital problems, and Max feared for his life. He hired a top secret service agent, and the agent discovered that the guy who was coming over was a criminal who wanted to kill the politician. The agent was able to foil his plan, and he ended up dead. |
| 501 | 0.437 | 0.562 | Tortured by his dead wife's angry ghost, the politician, Evan, had developed a phobia of women. However, when the ghost of his mistress, Alice, returned, he was forced to face the ghost of his wife, and to learn from it. |
| 502 | 0.777 | 0.852 | A secret agent was sent by the CIA to an information conference. He was supposed to observe the attendees and gather information on their backgrounds. He managed to find a suspicious man, and followed him around. |
| 503 | 0.148 | 0.986 | The spy, Leo, was working for the Secret Service. He was assigned the task of infiltrating the home of the politician, John, in order to learn about John's secret meetings with the lobbyist. However, Leo had a bit of a problem - he had fallen in love with the politician's wife, Mary. |

A.43 Examples of Texts over Iterations, Baselines, Opinions

Table 53: Examples of outputs and evaluations (Opinions domain) for early and later iterations. **Fixed-Few-Shot**. **Fixed-Few-Shot** demonstrates a strong inclination towards mimicking or heavily borrowing from the few-shot examples provided. This is particularly noticeable in the repetition of certain concepts and phrases. Notably, in iteration 51, the entire generated text is a direct replica of one of the few-shot examples. The references to "meat and ice cream" in iteration 71, "do not have an opinion" in iteration 60, and the recurrent mention of "pulses" in various iterations further underscore this mimicry trend. As we delve into later iterations like 503, 505, and 509, certain phrases such as "have an opinion" and the notion of recommending something are borrowed directly from the few-shot examples.

| Iteration | Phenotype | Fitness | Generated text |
|---|---|---|---|
| 51 | 0.999 | 0.679 | Plant-based foods are a great source of healthy micronutrients, and can play a significant role in providing you with macronutrients also. I would highly recommend including many different foods such as vegetables and pulses in your regular diet. |
| 60 | 0.881 | 0.269 | I do not have an opinion on what vegetables to eat. I will try eating vegetables as often as I can, but I am already doing a good job of eating a variety of healthy foods. |
| 71 | 0.011 | 0.679 | I don't like eating vegetables. I eat meat and ice cream instead. |
| 503 | 0.997 | 0.500 | I have an opinion that eating vegetables and other plant-based foods is a good thing. |
| 505 | 0.999 | 0.731 | Plant-based foods are good. I would recommend them to children. |
| 509 | 0.982 | 0.731 | I think most people have a preference for different types of food from the plant kingdom. Some people prefer cooked plant-based foods such as beans and pulses, and others prefer raw plant-based foods. |

Table 54: Examples of outputs and evaluations (Opinions domain) for early and later iterations. **Shuffling-Few-Shot**. Similar to **Fixed-Few-Shot**, **Shuffling-Few-Shot** also displays a propensity to recycle concepts and phrasings from the provided few-shot examples. For example, the mention of "nutrients" in iteration 505, and the repetitive usage of the phrase "I do not have an opinion" in iterations 56, 501, and 506. This consistent mimicry highlights the baseline's extensive borrowing from the few-shot examples.

| Iteration | Phenotype | Fitness | Generated text |
| --- | --- | --- | --- |
| 54 | 0.011 | 0.953 | Vegetables are not my thing. I would much rather eat meat and ice cream. Vegan? No thanks. |
| 55 | 0.999 | 0.562 | There is no doubt that a vegetarian or vegan diet is a very healthy diet to follow. It is also important to realise that plant-based foods are a great source of protein, vitamins and minerals. |
| 56 | 0.378 | 0.777 | I do not have an opinion on eating vegetables and other plant-based foods. I know that some people prefer a vegetarian or vegan diet, and others prefer eating meaty diets. |
| 501 | 0.029 | 0.731 | I do not have an opinion on eating vegetables and other plant-based foods. I do not think eating vegetables is a better alternative to eating meat. |
| 505 | 0.989 | 0.924 | I enjoy eating vegetables because they taste good, and they provide me with good nutrients. I do not understand why people are so obsessed with meat. |
| 506 | 0.029 | 0.622 | I do not have an opinion about eating vegetables and plant-based foods. I admit that I feel ashamed for not eating vegetables, but I do not have a firm opinion about it. |

Table 55: Examples of outputs and evaluations (Opinions domain) for early and later iterations. **Random-Search**. While **Random-Search** aims to promote diversity by retaining all entries, it may unintentionally impede optimization for higher fitness solutions. In later iterations like 501-503, the fitness values of the generated entries persistently fall short.

| Iteration | Phenotype | Fitness | Generated text |
| --- | --- | --- | --- |
| 51 | 0.269 | 0.679 | They always say that if you don't like vegetables, you can't like meat. But I do like meat. |
| 52 | 0.0373 | 0.269 | I'm going to have to find a way to cook vegetables without tasting them. |
| 53 | 0.993 | 0.438 | I do not favor or dislike vegetables and plant-based foods. I love them and eat them as often as possible. |
| 501 | 0.989 | 0.622 | The best vegetable is the one that tastes best to the individual. |
| 502 | 0.993 | 0.562 | Kids can eat vegetables if we make them fun. |
| 503 | 0.881 | 0.852 | I eat a lot of vegetables and plant-based foods, but I am only a vegan at heart. I'm not 100% strict. |

Table 56: Examples of outputs and evaluations (Opinions domain) for early and later iterations. **LMX, Quality-Only**. **LMX, Quality-Only's** singular emphasis on fitness pressure may inadvertently limit solution diversity, risking convergence to local optima. In earlier iterations 51 - 53, there is a broader range of phenotype and fitness values. However, by iterations 501 - 504, a distinct pattern centered around vegetables and dietary choices becomes evident, with a noticeable uptick in fitness scores. In iterations 1994 - 1996, the narratives almost always start with "I don't like", consistently taking a negative tone.

| Iteration | Phenotype | Fitness | Generated text |
|---|---|---|---|
| 51 | 0.971 | 0.0474 | Your favourite book/s:I have a good bunch of books, but I would say that my favourite is Dr. Sebi's Green Medicine, by Dr. Mehmet Oz and Dr. John Sebi. The book covers diet and lifestyle in |
| 52 | 0.438 | 0.818 | I don't eat a vegetarian or vegan diet, and I don't think they are particularly healthy. I believe plants are "natural foods" (as opposed to "unnatural foods," such as meat), and that we should be eating more plant- |
| 53 | 0.679 | 0.011 | Let's talk about how to overcome food cravings. Here are some tips to overcome food cravings: |
| 501 | 0.963 | 0.731 | I like vegetables. I don't find myself missing meat at all. I think it's good for you, even if you don't eat meat. I'm not sure if I could go vegetarian, I don't think I would mind going vegetarian for |
| 502 | 0.999 | 0.777 | I really like vegetables. Most vegetables are tasty and nutritious. Most vegetables are very tasty! |
| 504 | 0.119 | 0.905 | I like to eat vegetables. I like to eat lots of vegetables. But, I don't like uncooked vegetables. I hate uncooked carrots and celery. I hate soft lettuce. I hate soft leafy lettuce. |
| 1994 | 0.002 | 0.989 | I don't like eating vegetables. I like to eat meat. I like to eat cheese. I don't like vegetables. I can't stand vegetables. I eat vegetables when I'm forced to, but I don't like |
| 1995 | 0.004 | 0.963 | I don't like vegetables. I only eat them because I have to at restaurants, and for snacks. I prefer to eat meat. I don't like vegetables because they are boring. I only eat them because I have to. |
| 1996 | 0.001 | 0.986 | I dont like eating vegetables. I dont like the taste of broccoli. The only thing I like eating is meat. I dont think eating vegetables is good for you. I hate vegetables. I do not like eating vegetables. I hate vegetables. I think |

## A.44   Examples of Texts over Iterations, Baselines, Stories (Genre and Ending)

Table 57: Examples of outputs and evaluations (Stories - Genre and Ending domains) for early and later iterations. **Fixed-Few-Shot**. **Fixed-Few-Shot** shows a propensity to replicate or heavily draw from the few-shot examples. This pattern is evident in the recurrent themes of entrepreneurship and mansion parties. Specifically, in iterations 51 and 501, the description of a "tacky shirt" and "star-shaped glasses" is directly lifted, while characters like "Karl Johnson" in iterations 67 and 503, and "Dave" in iteration 501, echo the few-shot prompts.

| Iteration | Genre / Ending Phenotype | Fitness | Generated text |
|---|---|---|---|
| 51 | 0.438 / 0.004 | 0.378 | The wealthy entrepreneur Tom was hosting a party in his mansion. He was so surprised with his new found friend, Dave, who was wearing a tacky shirt, and star-shaped glasses. Tom invited him to be his guest in his lovely mansion, but before Dave could enter, he took a turn for the worst - an assassin shot Tom! |
| 55 | 0.378 / 0.437 | 0.731 | The secretary had become a teacher at his old school, and was assigned to teach a class on war and history. He knew the teacher to be a complete nerd, and had a remarkable ability to know all of the school's secrets. The spy, who was an old acquaintance, decided to infiltrate the teacher's house, in order to know what he's hiding. |
| 67 | 0.500 / 0.037 | 0.881 | The spy, Joanne, wanted to infiltrate the premises of Karl Johnson, a highly-influential figure in the city. Karl was a wealthy mayor, and would do anything in his power to suppress any opposing voices. Joanne wanted to figure out what Karl was hiding, but she took a turn for the worse, as she was highly suspicious in her presence outside his home. |
| 501 | 0.269 / 0.731 | 0.905 | The wealthy entrepreneur and member of parliament, Susan, hosted a party at her mansion. She invited all of the residents, as well as an unusual looking man. The man, Dave, was wearing a tacky shirt, and star-shaped glasses, and was actually a spy. He made the whole room laugh with his jokes, and had a secret agenda - to find what Susan does in her private fun room! |
| 503 | 0.378, 0.377 | 0.622 | A spy who was in on a secret gang, an odd thief who was a part of a mysterious group, and a city mayor who wanted to run for president. These are the three people who are a part of this story, and they all worked together to set up a massive conspiracy. Karl Johnson, the wealthy mayor of Derry, was in on the plot, and he was constantly getting spied on. |
| 504 | 0.378 / 0.500 | 0.679 | A highly-influential politician, the mayor, invited a mysterious person to the party. The weird man got drunk, played pranks, and started a problem in the party. The mayor tried to stop him, but he kept throwing glasses, and finally threw a bottle at the mayor's face. A secret agent by the name of Charlie realized that the mayor was the spy, and took action. |

Table 58: Examples of outputs and evaluations (Stories - Genre and Ending domains) for early and later iterations. **Shuffling-Few-Shot**. **Shuffling-Few-Shot** demonstrates a tendency to reuse phrases and concepts from the given few-shot examples. For example, the mention of "classified documents" in iteration 501 mimics that of "classified files" in the few-shot examples, the repetitive usage of the phrase "infiltrate the premises" in iterations 52, 501, and 503, and setting the story at a "party" in iteration 504.

| Iteration | Genre / Ending Phenotype | Fitness | Generated text |
|---|---|---|---|
| 51 | 0.018 / 0.622 | 0.818 | Barry is a secret agent for the intelligence agency. He works for a rich business man, and his on-going mission is to sniff out secrets, and try to discover why the businessman's son, Nick, has been behaving strangely. He met Nick in a bar, and the two hit it off well. Nick was attracted to Barry, because he always knew the right thing to say to the woman he liked. |
| 52 | 0.622 / 0.622 | 0.953 | The spy, Hunter, found the perfect way to infiltrate the premises of the wealthy politician, Dr. Malachai. The doctor was everything that Hunter was not - rich, powerful, and insane. Dr. Malachai disguised himself as a simple worker, and hid near the back of the mansion, while Hunter lurked near the front. Hunter wanted to find out what Dr. Malachai was planning to do, and how he was going to do it. |
| 53 | 0.562 / 0.148 | 0.731 | The politician's aide "Joe", accidentally spied on his boss's secret files. He was in shock upon seeing the huge collection of sensitive documents about the death of his best friend. He was convinced that his boss was involved in it, and wanted to find out more about the stabbing. |
| 501 | 0.223 / 0.119 | 0.881 | A spy sent to infiltrate the premises of a wealthy man, Stan, managed to get inside the man's secret room. He took all of the classified documents out and used them for his own gains. Stan found out about it and sent for the police to arrest the spy. The spy, however, was not ready to give up, as he thought he saw Stan's son! |
| 503 | 0.005, 0.731 | 0.852 | A spy named Sally infiltrated the premises of a rich billionaire, and was spotted by the man's son. The man's son, Dave, decided to take revenge on the spy after getting caught. He wanted to catch her in the act, but after getting a hold of her, he was surprised to find her to be a beautiful young woman. He never suspected that she was a spy, and after revealing their true identities, he found a spark of attraction. |
| 504 | 0.622 / 0.119 | 0.953 | The wealthy politician, Tim, was having a party at his home. He was hosting some big names in the city, including a student from a private university. The party got out of control, and the student was accused of trespassing. Tim got suspicious of the young man, as he feared blowing his cover. The student was spying on Tim, and tried to sneak into the basement, where his secret files were kept. |

Table 59: Examples of outputs and evaluations (Stories - Genre and Ending domains) for early and later iterations. **Random-Search**. By attempting to foster diversity through retaining all entries, **Random-Search** inadvertently stifles the progression towards solutions of higher fitness. In later iterations like 501-503, the fitness values of the generated entries persistently fall short.

| Iteration | Genre / Ending Phenotype | Fitness | Generated text |
|---|---|---|---|
| 51 | 0.438 / 0.817 | 0.018 | In my class, we had to write a book report on a fantasy story. I chose to write a book report on the book "The Stand". I have chosen to write a book report because I have read it, and I loved it. I think that this book is interesting because it is about a world without electricity or fire, and people have to fight with weapons. The main character is a man by the name of Stephen King. His name is really Stephen King, not Stanley Fiers. |
| 52 | 0.500 / 0.377 | 0.679 | A spy was entering a house, and he spotted a very suspicious man, who was holding a gun in his hands. The spy saw that the man was really suspicious, and he was trying to attack the spy. The spy shot the man with the gun, and then he left the house. |
| 53 | 0.321 / 0.182 | 0.852 | The dangerous spy, Oscar Volland, was hired by the rich politician, Sam. He was going to take over the company, but he didn't know that there was a spy in the company. He also didn't know that Sam was a corporate pirate. |
| 501 | 0.679 / 0.182 | 0.378 | The politician was a politician, and he was very famous in the world. He was very rich, and he had a very large house. He was very mean, and he had a big garden. The spy saw the politician walking around the garden, and he saw the politician's friends. The politician was a spy, and the friends were the same person. The politician was very suspicious, and he was very angry. |
| 502 | 0.095 / 0.730 | 0.622 | A man named Alex is a secret agent for the U.S. government. He works for the CIA, and is highly trained in many different things. Alex was walking around a building that he was guarding. He was watching the outside of the building, but did not see anyone suspicious. He just kept on walking, and when he got to the door he heard a knock on the door. He opened the door, and saw a rich politician. |
| 503 | 0.223 / 0.971 | 0.223 | The spy was going to go to his meeting. The spy leader said, "That's great!" and he was so happy. The spy was very happy, and he had a hard time suppressing his laughter. Luckily, no one spotted the spy laughing, and he went to his meeting. |

Table 60: Examples of outputs and evaluations (Stories - Genre and Ending domains) for early and later iterations. **LMX, Quality-Only**. Generated entries in later iterations have noticeably higher fitness than earlier entries. While it is difficult to tell whether the generated entries converged to a local optimum here, **LMX, Quality-Only** fails to generate entries in many bins.

| Iteration | Genre / Ending Phenotype | Fitness | Generated text |
|---|---|---|---|
| 51 | 0.622 / 0.562 | 0.622 | His name was Stephen, and he was the secretary of a rich business-man. One day, the businessman received a strange letter from a foreign country. They were concerned about the fact that the businessman had been watching too many foreign spy movies, and they thought that he had been caught by a spy. |
| 52 | 0.439 / 0.679 | 0.852 | The politician, Jack, behaved as if he didn't care about anything, while the young woman, Ruby, saw a conspiracy in everything. She insisted that the politician was a spy, and insisted on talking to him alone. She took him to a big mansion, and they found a hidden door that led to a special room where the politician was being held captive. |
| 53 | 0.269 / 0.622 | 0.679 | The spy is a young businessman. He has been a spy for many years. He has stolen many secrets from many people, and he has even lost his job because of his secrets. But he has a plan to get back at his enemies. |
| 502 | 0.500 / 0.047 | 0.777 | The spy managed to trick a politician. The politician was in a meeting with his trusted aides. The politician was sure that the aide, who was standing beside him, was a spy. The politician just couldn't figure it out. At the right moment, the spy managed to make the aide say something that the politician's aides had never said before. The aide was about to say something, but the politician stopped him and killed him. |
| 504 | 0.500 / 0.851 | 0.924 | A suspicious spy knew that the politician was probably going to try to steal something, so he decided to take precautions. He wore a disguise, and hid in the backseat of a car. He watched the politician, and saw him go into a house- but he accidentally walked in on him, and he saw that the politician was a thief. The suspicious spy knocked the politician around, and took his watch. |
| 507 | 0.622 / 0.075 | 0.924 | The politician had a mansion in the country, which was protected by a huge fence of barbed wire. However, the politician was at a party, and had left the gate open. The spy, disguised as a woman, slipped in the garden, and found the politician's private bedroom. He found the politician's diary, which detailed everything he had done. He decided to kill the politician, believing that the latter was a liar, and a murderer. |

