# OpenReview forum: "Quality-Diversity through AI Feedback"
_ICLR.cc/2024/Conference — ICLR 2024 poster_

### Official Review · Reviewer_K2hJ · 2023-10-31

**Soundness:** 3 good
**Presentation:** 3 good
**Contribution:** 4 excellent
**Rating:** 6
**Confidence:** 3

**Summary:**

This paper introduces an novel approach that integrates AI-generated (LMs) feedback into quality-diversity search algorithms, aiming to enhance the capability of AI systems in independent searching, evaluating, and innovation.

**Strengths:**

1. The authors have proposed a novel and effective quality-diversity algorithm that leverages the latest developments in AI feedback, demonstrating superior performance compared to existing alternatives.
2. The paper features a comprehensive set of experiments focused on creative writing, supported by a thorough analysis of the results, showcasing the practical applicability of the proposed method.

**Weaknesses:**

1. The presentation, particularly in the experimental section, could benefit from clarification and better organization to enhance readability and comprehension.
2. The paper occasionally employs exaggerated language and makes promising claims that seem to lack sufficient empirical backing. For example, the paper states “providing a recipe that seemingly generalizes to many domains and modalities” and “it is often easier for a model to evaluate the quality of a generation than to generate the same high-quality text.” These claims would be more convincing if supported by concrete evidence or reference or discussion.

**Questions:**

A separate conclusion section, summarizing the key findings and contributions, would be advantageous for providing clear takeaways for the readers.

---

> ### Author Response · Authors · 2023-11-22
> **Thank you reviewer K2hJ**
>
> We thank the reviewer for their positive review and useful feedback! We address the weaknesses below:
>
> > The presentation, particularly in the experimental section, could benefit from clarification and better organization to enhance readability and comprehension.
>
> We have uploaded a new draft of the paper which we hope greatly improves the readability and structure. In particular, Section 4.2 shows changes in blue text with improvements to clarity in the experimental setup, and a correction to a section reference in the baseline description of **Random-Search**
>
> > The paper occasionally employs exaggerated language and makes promising claims that seem to lack sufficient empirical backing. For example, the paper states “providing a recipe that seemingly generalizes to many domains and modalities” and “it is often easier for a model to evaluate the quality of a generation than to generate the same high-quality text.” These claims would be more convincing if supported by concrete evidence or reference or discussion.
>
> We take this concern seriously and take a moment to clarify on the two highlighted points to resolve the misunderstanding. On the quote “providing a recipe that seemingly generalizes to many domains and modalities”, we highlight in the conclusion through reference that AI feedback can be done with any model that has been instruction-tuned such as the AI feedback model used in QDAIF. We cited in the conclusion Liu et al., 2023, a work that introduces instruction tuning for multimodal text-image understanding models. Instruction tuning enables this model to solve tasks in image and text domains, including solution evaluation - it is interesting future work for example to apply AI feedback to images with the visual instruction-tuned model (or another similar image-text model such as GPT-4+V) to evolve images and creative art, that can be described by AI feedback on the new domain of images, and generate refinements of images.
>
> On the quote “it is often easier for a model to evaluate the quality of a generation than to generate the same high-quality text”, we cited Saunders et al., 2022, to refer to the generation-discrimination gap described by the quote. We also cited this part of their work in Section 4.4 in the original manuscript (now in A.18), but realized that we forgot to include page number 12 in the first mention of this reference (which we did in Section 4.4 originally), containing the reference to generation-discrimination gap which gives context to our results. The page number is added now to the first mention of the reference. We apologize for the lack of clarity, and hope the added detail will prevent future readers from sharing the same confusion.
>
> > A separate conclusion section, summarizing the key findings and contributions, would be advantageous for providing clear takeaways for the readers.
>
> Thank you for sharing this suggestion and for thinking of ways to improve the presentation of QDAIF. We added to the first paragraph of the conclusion a brief statement on key takeaways from results for our draft revision, to support the communication of the contributions messaging in the final paragraph of the conclusion.
>
> ### Conclusion:
> We would be grateful for your thoughts on the quality of our manuscript and contributions of QDAIF following our response here as well as the general comment, and hope to get your support in realizing QDAIF as a valuable contribution to ICLR, during the remaining time of the discussion period.
>
> If you feel that your comments have been adequately addressed, we would greatly appreciate it if you could update your score to reflect that.
>
> Thank you for your encouraging statements on the strengths of the paper. We look forward to hearing more from you on ways to make this paper stronger.

---

> ### Author Response · Authors · 2023-11-23
> **Follow up**
>
> We understand that the author discussion period is closing soon, but we are still open to coming back to you on any remaining questions you may have about the manuscript. We have gone deep into following up on your concerns, and have resolved them from our understanding of your points raised.
>
> We would be happy to quickly answer any potential remaining questions from you. Thank you again for your work on improving the paper, and encouraging statements on the strengths of our work.

---

### Official Review · Reviewer_3nT4 · 2023-10-31

**Soundness:** 2 fair
**Presentation:** 2 fair
**Contribution:** 2 fair
**Rating:** 6
**Confidence:** 3

**Summary:**

This paper proposed a pipeline combining a quality-diversity search algorithm and LLM as feedback for quality and diversity, aiming to improve the quality-diversity in the creativity domain, such as creative writing. The authors also conduct human evaluations to evaluate the output of the pipeline.

**Strengths:**

Strength:
* The paper addresses an interesting problem in the creativity domain, namely, generating solutions that are both diverse and of high quality. The integration of AI feedback and the existing QD algorithm seems to be novel and interesting

**Weaknesses:**

Weakness
* One key motivation of the paper using AI feedback seems to bypass the necessity to articulate a set of criteria. However, the prompt strategies still resort to specified diversity and quality criteria
* The evaluation result seems to be a bit confusing; for example, in Table 1, one of the methods is LMX, Fitness-only, then in section 4.3 when it explains the method, there is LMX Quality-only. Is that the same method as LMX, fitness-only
* It would be interesting to see ablation analysis to compare with the QD with and without AI feedback (not sure if LMX fitness-only or quality only serves the baseline)
* QD metric is used throughout, which is the “sum of highest quality value found in each bin” - it seems to only focus on quality rather than diversity. For readers not familiar with QD metrics, some explanation/justification of why QD measures Quality and Diversity will be helpful

**Questions:**

Table 1:  there is a lack of explanation of the quality metrics. For example, what is the difference between human QD score and quality rating? Is quality rating from humans?

* On page 5, section quantifying performance and diversity, it is unclear where the probability comes from in the sentence “ the solutions’ quality estimate is derived from the logarithm of the probability of the LM’s answer” …  Please clarify

* Figure 3 illustrates the differential performance of various methods on different generation tasks. Is there any qualitative difference in terms of the QD score difference of less than one point

---

> ### Author Response · Authors · 2023-11-22
> **Thank you reviewer 3nT4 [1/3]**
>
> We thank the reviewer for their thoughtful review of the paper. We come back to the reviewer’s questions and concerns below, starting off with some clarifications to aid in the understanding of our findings and explanations, and then in the second part of the reply clarifying on main concerns.
>
> > The evaluation result seems to be a bit confusing; for example, in Table 1, one of the methods is LMX, Fitness-only, then in section 4.3 when it explains the method, there is LMX Quality-only. Is that the same method as LMX, fitness-only
>
> Thank you for highlighting this for correction! It was a typo on our part, “LMX, fitness-only” should be worded as “LMX, Quality-Only”. We corrected this in Table 1 in the revised draft, shown in blue text. The Quality-Diversity (QD) literature frequently uses the term “fitness” for the notion of quality as a metric, for general context.
>
> >QD metric is used throughout, which is the “sum of highest quality value found in each bin” - it seems to only focus on quality rather than diversity. For readers not familiar with QD metrics, some explanation/justification of why QD measures Quality and Diversity will be helpful
>
> In many domains you want to get a wide range of high-quality diverse artifacts (the motivating case as described in the introduction); the QD score [1] is one popular measure used in the literature. QD score increases are possible in two ways - by filling a previously empty bin with a solution of that kind (i.e. finding a novel solution, with improvement to the overall archive by the amount of quality improvement introduced from the addition of this new solution of a particular quality value), or by improving over an existing solution in a filled, non-empty bin with an evolved solution that is evaluated to be of this kind/bin, but has a higher quality score, as described in the method overview in Section 3 regarding the acceptance of evaluations to the archive. Coverage of the archive with new, diverse solutions, is, therefore, a key contributor to improvements in QD score (which we highlight in new additional stats including coverage in the added A.7 section, referenced in figure 3).
>
> > Table 1: there is a lack of explanation of the quality metrics. For example, what is the difference between human QD score and quality rating? Is quality rating from humans?
>
> Yes, quality rating is from humans. Thank you for sharing your concern on readability here. We provide full details on the human evaluation study referenced in the Table 1 results in Appendix A.1 and reference it in the “Evaluation” paragraph of Section 4.1. We added minor clarification and further self-contained referencing of A.1 in the revision for improved clarity.
>
> > On page 5, section quantifying performance and diversity, it is unclear where the probability comes from in the sentence “ the solutions’ quality estimate is derived from the logarithm of the probability of the LM’s answer” … Please clarify
>
> As part of the standard AI feedback method, the quality and diversity prediction values are obtained from LM next-token-prediction log probabilities, where we assess the probability of the AI feedback LM in predicting for a given text/story one label/token(s) (e.g. “horror”) versus another label/token(s) (e.g. “romance”). We quote in the section referred here in the paper, “The log probability of these responses serves as our measure of solution diversity” for context, applying the same approach to quality AI feedback. We slightly reworded a part of this sentence, highlighted in blue text. Thank for raising possible suggestions to improve the readability of the paper.

---

> > ### Author Response · Authors · 2023-11-22
> > **Thank you reviewer 3nT4 [2/3]**
> >
> > > One key motivation of the paper using AI feedback seems to bypass the necessity to articulate a set of criteria. However, the prompt strategies still resort to specified diversity and quality criteria
> >
> > We would like to push back on the claim by the reviewer mentioned here in the quote’s first sentence, and resolve the potential misunderstanding in the process. We highlight that AI feedback enables us to have qualitative criteria and in theory for models to do more with less human effort. This is described in the introduction on the potential of prompting LMs to do the evaluation steps of QD search via QDAIF, Section 2.3 on the role of AI feedback (with minor clarification improvements in the revised draft), as well as the description of the QDAIF method in Figure 2 and Section 3 paragraph on “Quantifying Performance and Diversity”. Furthermore, Figure 1 showcases the strengths of QDAIF in returning diverse, high-quality solutions for story writing within the desired domain and specified genre and story ending axes. These points support QDAIF as a contribution to making QD search feasible in new, more-qualitative domains, to tackle QD problems in the context of search with foundation models), while not claiming that QDAIF bypasses the need to articulate a set of criteria, which is the standard approach of MAP-Elites, a common standard QD algorithm [2].
> >
> > Still, we noted (as was done in the discussion on limitations in Section 5) this particular requirement specified diversity and quality criteria as the limitation of most QD algorithms in the literature (which be build upon with QDAIF applying MAP-Elites and the usage of defined binning archives spanning a diverse range of diversity attributes, and each cell/bin occupying different kinds of possible solutions). Hence, we added A.10 in the manuscript revision to study a promising approach toward automating the definition and expansion of such diversity criteria for MAP-Elites. We highlight that recent advances in LMs (such as GPT-4) can allow us to prompt LMs to automatically generate such AI feedback prompts for diversity, or other aspects of defining interesting diversity measures (in line with OMNI [3], which we cited, as a relevant approach of utilizing LM prompting to automate search through general knowledge guidance). This does not answer the question of potential performance improvements that can be obtained from expanding the dimensions of MAP-Elites axes, so we obtained results proving the viability of conducting this expansion of archive dimensions, approaching the performance in QD score compared to searching from the start with already-initialized high-dimensional archives (QD score is the sum of fitness scores across all bins, where the archive bins span diverse axes covering different kinds of solutions, e.g. in the Figure 1 visualization of different, diverse kinds of stories expected in the search). Our results here highlight the potential of our proposed approach to tackle this limitation of QD algorithms.
> >
> > > It would be interesting to see ablation analysis to compare with the QD with and without AI feedback (not sure if LMX fitness-only or quality only serves the baseline)
> >
> > We conducted an ablation presented in the original manuscript, where we replaced AI feedback evaluation with a different semantic embedding-based approach. This is referenced in the last part of Section 2.2, and described in Appendix A.2. The human evaluation study showed that conducting this QDEF (QD through Embedding Feedback) leads to optimization with respect to these proxy measures of quality and diversity, but also demonstrates poor results where humans found the sampled output texts (at the end of seach) from this ablation method variant QDEF to possess lower quality and diversity.

---

> > > ### Author Response · Authors · 2023-11-22
> > > **Thank you reviewer 3nT4 [3/3]**
> > >
> > > > Figure 3 illustrates the differential performance of various methods on different generation tasks. Is there any qualitative difference in terms of the QD score difference of less than one point
> > >
> > > Yes, differences in QD score performance of less than one point can lead to notable qualitative differences in observed solutions, depending on the context of the results of the search. Taking the example from qualitative samples shown in Figure 1, when looking at the same bin between QDAIF and the baseline (e.g. the central bin, where the baseline contains a low-quality solution with a red cell/bin with blue arrow pointing to the cell from a text box on the right of the baseline archive, and QDAIF a high-quality solution with a bright cell in the same bin location), the differential is based on quality scores. In this particular example, where the difference in quality score between is less than one, we can observe this from the quote (in the revised sentence with minor correction on figure object location): “The baseline produced a story (right-middle position, starting with "Jason") with a lower quality score due to the lack of a desired spy character (denoted by the red-colored bin, for a story with a neutral ending, and leaning to horror). QDAIF discovered a better, more-relevant story (bottom-middle position, starting with "a wealthy politician") for this same neutral bin”. We can see that if both methods find solutions of similar quality across most (of the same) bins except for one bin where one method finds the higher quality solution (e.g. a better horror story), then the QD score differential observed would be less than one, but means a qualitative (subjectively higher quality) solution if the method can improve on the quality of this genre of story/solution. A differential of less than one can also occur if a method finds a solution in one more previously unfilled bin (that would be still empty in the result of a different method in this case) - the QD score improvement introduced by filling this empty bin with a new solution would virtually always be less than one, the maximum possible quality score achievable in this setup of story-writing. Just finding a previously unknown solution in a region of the archive that's further away in the extreme corner (cf. Figure 1) can lead to noticeable qualitative differences in solutions (e.g. the horror story from QDAIF in the top-left text box with a tragic ending has more horror elements, with reference to a deadly monster, compared to the closest bin story found by a baseline shown in the top-right text box, which only shows a character's tragic ending via gun violence, without traditional horror elements in the text).
> > >
> > > While deeper qualitative review of solutions on a case-by-case basis is needed to understand the situational differences in qualitative results between runs, We hope that this intuition of what influences QD score and how it could affect qualitative analysis brings more clarity to this question.
> > >
> > > ### Conclusion:
> > > We would be grateful for your thoughts on the quality of our manuscript and contributions of QDAIF following our response here as well as the general comment, and hope to get your support in realizing QDAIF as a valuable contribution to ICLR, during the remaining time of the discussion period.
> > >
> > > If you feel that your comments have been adequately addressed, we would greatly appreciate it if you could update your score to reflect that.
> > >
> > > Thank you for your interest in our work, we hope that this response brought a deeper understanding of the intuitions of QDAIF.
> > >
> > > [1] QD Score - https://www.frontiersin.org/articles/10.3389/frobt.2016.00040/full
> > >
> > > [2] MAP-Elites - https://arxiv.org/abs/1504.04909
> > >
> > > [3] OMNI - https://arxiv.org/abs/2306.01711

---

> ### Author Response · Authors · 2023-11-23
> **Follow up**
>
> We have added additional material in the v2 revision to further expand upon the topic of Automating QDAIF Search (Diversity Axes).
>
> We hope that this gives you further insights into the strengths of QDAIF. We have followed up on all of your concerns, and see that we have resolved them from our understanding of your points.
>
> We would be happy to quickly answer any potential remaining questions from you. Thank you again for your work on improving the paper.

---

### Official Review · Reviewer_8ua9 · 2023-10-31

**Soundness:** 3 good
**Presentation:** 4 excellent
**Contribution:** 3 good
**Rating:** 6
**Confidence:** 3

**Summary:**

This paper addresses the problem of generate a diverse range of high-quality outputs by using AI feedbacks, instead of traditional Quality-Diversity (QD) search algorithms. The authors propose a Quality-Diversity through AI feedback (QDAF) where an evolutionary algorithm applies LMs to both generate variation and evaluate the quality and diversity of candidate text. Experiments show that produced outputs have a reasonable agreement between AI and human evaluation.

The proposed approach QDAIF builds upon MAP-Elites [Mouret and Clune, 2015], which follows those steps: randomly select a solution, mutate it, evaluate the new solution in terms of quality and diversity. Finally, if the new solution is better, all previous cells are replaced. The improvement of QDAIF is significant and leverage the characteristics of LLMs. Instead of using a uniformly-separated grid, the authors split the grid by density. This makes a lot of sense because we output distributions generated by LLMs are skewed. The initialization and mutation are based on few-shot prompting. Finally, the quality and diversity evaluation is done via prompting the LLMs and observing whether the answer is "yes" or "no" and their log-probabilities.
Overall, the method is a composition of simply ideas that make it easy to follow.

The experiments consists of creative text generation in the domains of opinions and stories. The domain in the former is about eating plant-based diets, and for the latter a short story containing a spy and a politician. Diversity is based on the sentiment towards a topic in case o opinions and genre and ending for the stories. The authors evaluate using QD score [Pugh et al. 2016] and human evaluation. In terms of baseline, they seem to be simple variations of the framework. It would be needed to have baselines cited in prior work. The results are significantly better for QDAIF, but it is unclear whether this is due because the baselines are bad or whether the model is really better. I appreciate the other experiments that the authors have conducted on scaling up the models and trying other mutation methods.

Overall, the paper is well structured and written. The idea is simple but sounds effective. My only concern is the lack of more sophisticated baselines. I would ask the authors to evaluate their proposed approach on another task, such as control text generation (e.g., writing about specific topics and using a classifier to assess the topic being discussed)

POST-REBUTTAL: Thank you for your answers. I will keep my current rating.

**Strengths:**

- A combination of simple ideas that allow to generate diverse high-quality outputs
- Strong performance in the experiment section
- The paper is well written

**Weaknesses:**

- The lack of strong baselines
- More datasets for the experiments would be appreciated

**Questions:**

Could you add more baselines and tasks?

---

> ### Author Response · Authors · 2023-11-22
> **Thank you reviewer 8ua9**
>
> We thank the reviewer for their insightful review, and are encouraged by their comments on the strengths of the paper. We come back to the reviewer’s concerns on experimental soundness below.
>
> > In terms of baseline, they seem to be simple variations of the framework. It would be needed to have baselines cited in prior work. The results are significantly better for QDAIF, but it is unclear whether this is due because the baselines are bad or whether the model is really better… My only concern is the lack of more sophisticated baselines.
>
> Thank you for sharing your feedback on the importance of detailed and informative baseline comparisons. We agree with this statement, and revised the draft with additional baseline comparisons, introducing new baselines from prior works (on top of the LMX, Quality-Only baseline introduced in the original manuscript, and studied in the LMX, or Language Model Crossover paper [1]) and detailing performance comparison results with QDAIF in Appendix A.8, with detailed implementation details of added diversity-seeking baselines in A.9. In addition to further demonstrations of performance improvements by QDAIF compared to baselines, the new comparisons highlight the importance of each component of QDAIF in improving the QD score performance during search, with new insights on the value of diversity-seeking elements of relevant search algorithms, often outperforming the baseline which optimizes for quality in solutions only (LMX, Quality-Only).
>
> Furthermore, as highlighted in the general response, we introduced a new variant of QDAIF for Poetry, LMX-rewrite (in edited Section 4.4), which becomes a fair comparison to the Random-Poems baseline, where both methods do not request for solutions of specific genres and tones in poetry, but LMX-rewrite adds the rewriting mutation and evolution of archive bin solutions to improve over Random-Poems. This experiment improvement also enabled us to conduct an ablation with the method Fixed Seed Rewrite, which reveals that the element of evolving an improving population with QDAIF (in addition to rewriting mutation) is important for significant performance improvements with QDAIF.
>
> Given the improvements to the experimental soundness with sophisticated baselines, and a showcasing of the strengths and weaknesses of baselines, we have shown performance gains from QDAIF to be meaningful (with improved insights on the contributions of components of QDAIF), in line with seminal works in the literature which also compare to baselines (such as Random-Search, as well as LMX, Quality-Only for single objective quality/fitness optimization), in the experiments of the Novelty Search [2] and MAP-Elites [3] papers.
>
> > I would ask the authors to evaluate their proposed approach on another task, such as control text generation (e.g., writing about specific topics and using a classifier to assess the topic being discussed)
>
> Thank you for your interest in the applicability of QDAIF to other domains in addition to creative writing such as the existing **Opinions**, **Stories**, and **Poetry** domains.
>
> Firstly, the **Opinions** domain seems to be relevant to the idea of topic writing, which is highlighted in the review here in the paper explanation. The topic of eating vegetables and plant-based foods would likely be relevant here.
>
> Still, we take the suggestion on the value of studying QDAIF in different kinds of domains into account, and added results of QDAIF applied to a new (non-creative-writing) code-writing domain in Appendix A.20. We saw marked improvement in the diversity and quality of generated code from QDAIF (LMX-rewrite) for the task of implementing sorting algorithms to solve a specified problem, compared to the Random-Code baseline.
>
> We believe that results here also demonstrate the flexibility and generality of QDAIF, in being helpful to creative search in another, more-technical domain.
>
> ### Conclusion
> We would be grateful for your thoughts on the quality of our manuscript and contributions of QDAIF following our response here as well as the general comment, and hope to get your support in realizing QDAIF as a valuable contribution to ICLR, during the remaining time of the discussion period.
>
> If you feel that your comments have been adequately addressed, we would greatly appreciate it if you could update your score to reflect that.
>
> Thank you again for your helpful feedback that further improved the paper’s quality!
>
> [1] Language Model Crossover (LMX) - https://arxiv.org/abs/2302.12170
>
> [2] Novelty Search - https://stars.library.ucf.edu/facultybib2010/1530/
>
> [3] MAP-Elites - https://arxiv.org/abs/1504.04909

---

> ### Author Response · Authors · 2023-11-23
> **Follow up**
>
> We have added additional material in the v2 revision to further expand upon the topic of Additional Domain Beyond Creative Writing and an Additional Baselines.
>
> We hope that this gives you further insights into the strengths of QDAIF. We have followed up on all of your concerns, and see that we have resolved them from our understanding of your points.
>
> We would be happy to quickly answer any potential remaining questions from you. Thank you again for your work on improving the paper.

---

### Official Review · Reviewer_gkKB · 2023-11-01

**Soundness:** 2 fair
**Presentation:** 3 good
**Contribution:** 2 fair
**Rating:** 5
**Confidence:** 3

**Summary:**

This paper introduces Quality-Diversity through AI Feedback (QDAIF), a novel method that leverages advances in foundation models to evaluate the quality and diversity of generated text in qualitative domains. QDAIF employs an evolutionary algorithm that uses language models to generate variation and evaluate the quality and diversity of candidate text. The results demonstrate that QDAIF covers more of a specified search space with high-quality samples compared to non-QD controls and aligns with human perception of quality and diversity.

**Strengths:**

1. QDAIF presents a novel approach to discover diverse and high-quality solutions in qualitative domains by leveraging AI feedback, which contributes to the development of AI systems.
2. The paper thoroughly discusses limitations and potential future work, offering some insights for further research in this area.

**Weaknesses:**

1. QDAIF still requires researchers to define the axes of diversity they are most interested in, which may limit its autonomy in creative search.
2. Could we have a detailed comparison with other AI feedback methods or discuss how QDAIF specifically addresses their limitations?
3. The generalizability of QDAIF to other domains and tasks beyond creative writing is not extensively discussed.

**Questions:**

1. Can you provide more insight into the scalability and computational efficiency of QDAIF in more complex and large-scale tasks?
2. How does the proposed QDAIF approach perform in other domains and tasks beyond creative writing?

---

> ### Author Response · Authors · 2023-11-22
> **Thank you reviewer gkKb [1/2]**
>
> We thank the reviewer for their insightful review and thoughtful comments on the strengths of the paper. We come back to the reviewer’s questions and concerns below.
>
> > QDAIF still requires researchers to define the axes of diversity they are most interested in, which may limit its autonomy in creative search.
>
> Thank you for raising this interesting discussion point following the presented statements on the limitations of QD methods in the final section of the manuscript. We added Appendix A.10 which investigates and tackles the concern raised here through additional experiments and literature review, and referenced A.10 in the conclusion section on the limitations discussion.
>
> Firstly, to give more context from prior works in the literature (as we described in the context of the discussion on limitations since the original manuscript), most existing algorithms since the first introduction of QD search have not yet studied approaches (if at all) to alleviate the need to define the axes of diversity, especially ones that would likely be most interesting to users in a creative search. In A.10, we added references for an existing possible direction of defining higher dimensions of diversity via unsupervised diversity representation learning. Here we noted limitations of this approach in the case where for our desired use cases, we *do* want to leverage the general knowledge of assessing all types of creative writing from foundation models to inspire creative QDAIF search; we can then exploit LM domain knowledge for expanding on possible output text solutions where the distinctions in diverse, high-quality texts can be subjectively appreciated and informatively cataloged.
>
> We describe in A.10 an approach where such domain knowledge for automating the initialization or expansion of diversity axes can be explored for more diverse, high-quality texts. With recent advances in LM capabilities (e.g. GPT-4), we can specify the search problem or domain of interest, and ask it, for example, to directly generate ideas for plausible new diversity axes to explore, or even directly generate the AI feedback prompts which may be used in the existing QDAIF setups, maybe following the structure and format of say the feedback prompts we tested.
>
> The question remains - how effectively would QD score performance (the measure of quality and diversity in archive solutions) improve if one tested such a pipeline, and continued search with the introduction of additional diversity axes over time.
>
> We demonstrated in A.10 results that applying dimension expansion on the **Stories** domain as an example is effective in enabling the improvement of QD score in higher dimensions of diversity, surprisingly even providing gains in “Best Solution Quality” (mentioned in general response).
>
> Overall, new results that we found and added bring us closer to the potential of automatically defining diversity axes, proving practical performance gains to be had.
>
> > The generalizability of QDAIF to other domains and tasks beyond creative writing is not extensively discussed… How does the proposed QDAIF approach perform in other domains and tasks beyond creative writing?
>
> We added Appendix A.20, referenced in Section 4.4, to show the results of QDAIF applied to coding problems, which is not focused on creative writing, and compared to the Random-Code baseline.
>
> We are encouraged that you highlighted the importance of applying QDAIF to highly-subjective, qualitative domains, where no viable measures except for AI feedback (the alternative to expensive human feedback) exist, unlike the domains studied by QD works in prior work. Interestingly, we show in new results that AI feedback’s ability to bring evaluation to qualitative domains is also effective for code-writing tasks, where QDAIF generated qualitatively more diverse sorting algorithms (from first principles) with higher quality scores obtained, and overall higher QD score than that of the baseline (that returned nearly always one particular type of sorting algorithm, 95% of the time).
>
> We appreciate the interest of the reviewer in QDAIF when applied to domains beyond creative writing. We further highlight the generalizability of QDAIF with this finding.

---

> ### Author Response · Authors · 2023-11-22
> **Thank you reviewer gkKb [2/2]**
>
> > Could we have a detailed comparison with other AI feedback methods or discuss how QDAIF specifically addresses their limitations?
> Yes, we highlight the relevance of AI feedback (AIF) methods in the context of QDAIF in the original manuscript in Section 2.3, and added minor clarifications in this section in the revised draft, providing more clarity on how QDAIF leverages AIF in a new setting to introduce a new kind of QD algorithm, enabling other recent advancements in QD research studied in different domains to be applied on top of the high-level QDAIF approach.
>
> We show with this clarification that QDAIF is a complementary direction of research that takes inspiration from works applying AIF such as Constitutional AI [1] and Self-Refine [2], and their proven results in utilizing AI feedback for qualitative text quality assessment (which led to LM refinements being able to improve text solutions), to apply AIF to a new context towards tackling QD problems (adding the diversity and evolution aspects for a new application in QD).
>
> We hope that this brings a clearer picture of the differences between QDAIF and other AI feedback methods, where QDAIF enables QD approaches in new domains through qualitative AI feedback.
>
> > Can you provide more insight into the scalability and computational efficiency of QDAIF in more complex and large-scale tasks?
>
> The scalability and computational efficiency of QDAIF both hinge on the capabilities and sizes of available existing LMs (which carry out both the mutation of texts, and the evaluation of text solutions defined by AI feedback prompts).
>
> We refer to an additional analysis carried out and described since the original manuscript version to study the effect of the generator/mutator LM’s model size in Section 4.3, on “LMX Model Size”. Results summarize likely gains in scalability, where discovered samples of creative writing solutions found by QDAIF (with larger model sizes as generator LMs) obtained higher human feedback scores for quality compared to their ratings of samples from smaller LMs.
>
> Furthermore, our additional analysis on the qualitative sample poems in the **Poetry** domain in the revised manuscript, Appendix A.18 (in particular, Figures 21 and 22) suggests potential improvements in meaningful mutations of texts towards more diverse, higher-quality texts when running QDAIF with larger, more capable models such as GPT-4 compared to GPT-3.5 Turbo. While QDAIF benefits from more search iterations (meaning more LM generation and evaluation calls via label/attribute prediction), it still shows QD score performance sample efficiency improvements over existing baselines, hinting at the potential scalability of QDAIF that can improve with advancements to available LMs.
>
>  ### Conclusion:
> We would be grateful for your thoughts on the quality of our manuscript and contributions of QDAIF following our response here as well as the general comment, and hope to get your support in realizing QDAIF as a valuable contribution to ICLR, during the remaining time of the discussion period.
>
> If you feel that your comments have been adequately addressed, we would greatly appreciate it if you could update your score to reflect that.
>
> Thank you again for your work on the review.
>
> [1] Constitutional AI - https://arxiv.org/abs/2212.08073
>
> [2] Self-Refine - https://arxiv.org/abs/2303.17651

---

> ### Author Response · Authors · 2023-11-23
> **Follow up**
>
> We have added additional material in the v2 revision to further expand upon the topic of Automating QDAIF Search (Diversity Axes) and an Additional Domain Beyond Creative Writing.
>
> We hope that this gives you further insights into the strengths of QDAIF. We have followed up on all of your concerns, and see that we have resolved them from our understanding of your points.
>
> We would be happy to quickly answer any potential remaining questions from you. Thank you again for your work on improving the paper.

---

### Author Response · Authors · 2023-11-22
**General comment and thanks to all reviewers**

We thank the reviewers for their thoughtful and informative reviews of QDAIF.

We appreciate that reviewers found the QDAIF solution to be “novel” (gkKB, 3nT4, K2hJ), “interesting” (3nT4), “effective” (K2hJ, 8ua9), “practical” (K2hJ), and “simple” (8ua9); that reviewers felt QDAIF’s strong “performance” over “existing alternatives'' to be noteworthy (8ua9, K2hJ); that the problem of discovering “diverse and high-quality solutions” in “qualitative” and “creative” domains (gkKB, 3nT4) is “interesting” (3nT4); that the paper is “well-written” (8ua9), contains a “comprehensive set of experiments” (K2hJ, 8ua9) with “thorough analysis of the results” (K2hJ), and “thoroughly” discusses “limitations and potential future work” (gkKB). We are encouraged that reviewers found the integration of a human evaluation study (as well as its findings) to be noteworthy (gkKB, 8ua9, 3nT4).

We are grateful for the feedback on suggestions to make the paper stronger, and have done our best to address all concerns raised by the reviewers. As a result, we feel that the revised manuscript has been significantly improved through implementing reviewer suggestions.

Most importantly, we reinforced the results with **additional deeper analysis of baseline comparisons on domains**, **an analysis of QDAIF on a new non-creative-writing domain**, and **an investigation of an approach to alleviate the need to manually define diversity axes at the start of search**.

We display edited parts in the revised manuscript with blue font text, with some condensation of the text in Section 4.3 to meet the main text 9-page limit for the discussion phase.

# Main Changes

## Baselines
- Added Appendix A.8, A.9, referenced in Section 4.2, paragraph "Performance Comparison" - we found that QDAIF outperforms additional (*diversity*-based search) baselines tested on the **Opinions** and **Stories** domains. These baselines contribute to previously missing comparisons between QDAIF and other established approaches of diversity-seeking, now adapted to our creative writing domains for method comparisons and deeper insights on the most valuable components of QDAIF.
- Revised Section 4.4 - In **Poetry**, we found a simpler, more-general, and more comparable approach to QDAIF with less domain knowledge requirements during generation. This involves applying **LMX-rewrite** mutation, which prompts LMs (such as GPT-4) to rewrite poems to simply be different and high-quality, without the explicit guidance (with **LMX-guided**) on domain-specific targets for diversity (i.e., specifying which genres and tones the rewritten poems should be). This tweak lets us more closely compare QDAIF to the existing baseline, **Random-Poems**, which simply generates award winning poems from scratch, without the rewriting step, or the aspect of diverse solution evolution with QDAIF. We also added an ablation of QDAIF with the **Fixed Seed Rewrite** method, which carries out the rewrite step of **LMX-rewrite**, but is restricted to just the seed poem that initializes the QDAIF search. From this, we found significant QD score performance improvements from QDAIF over both methods, with **Fixed Seed Rewrite** also outperforming **Random-Poems**. This further reveals insights about the importance of the different elements of QDAIF in improving quality and diversity in solutions. Extensive improvements through qualitative analyses were added here (e.g. in added A.17 - A.19, referenced in Section 4.4).
- Added A.7, referenced in Figure 3 - General extensions to the metrics studied in our method comparisons: archive coverage (i.e., proportion of diverse bins filled with a solution), and best solution quality (the global highest-quality solution, across all bins).

## Additional Domain Beyond Creative Writing
- Added Appendix A.20, referenced at the end of Section 4.4 - additional comparison between QDAIF (LMX-rewrite) and a baseline on a new domain/task that is focused on writing code to solve problems with diverse approaches, showing again performance gains and novel insights beyond creative writing.

## Automating QDAIF Search (Diversity Axes)
- Added A.10, referenced in Section 5 (about discussion on limitations) - we introduce the first step (and performance potential) in enabling search, to build upon LMs automatically generating expanding diversity axes.

QDAIF aims to tackle the (quality-)diversity problem in outputs from foundation models, by introducing a new approach to conducting QD search, thereby showing improvements to the quality and diversity of solutions returned by models in creative writing, a domain where the only viable approach with existing tools to effective evaluations of creative writing (which QD methods hinge on in all previously studied domains) is AI feedback, replacing expensive human feedback.

We thank the reviewers for their work, and hope they consider strengthening their support in making this work a valuable addition to the wider ML community at ICLR.

---

### Author Response · Authors · 2023-11-23
**General comment on v2 revision update**

We have just uploaded another revision to our draft with minor proactive additions to add to our initial responses.

## Automating QDAIF Search (Diversity Axes) (gkKB, 3nT4):
- In A.10, the addition of a new approach to searching through a growing number of automatically added diversity axes through transitioning from one 1D diversity axis to a different 1D diversity axis, with results from this highlighting a promising direction for automating QDAIF search in a simple manner, with improved potential for scaling up the approach computationally.
- Figure 15 updated performance plots and Figure 17 added line plots.

## Additional Domain Beyond Creative Writing - Coding (gkKB, 8ua9):
- Added Figure 27 to Appendix A.20 to show additional QD Score line plot between QDAIF and baseline, for findings on sample efficiency.

## Baselines (8ua9)
- Minor improvement to clarity in the diversity-seeking baseline implementations in A.9, in particular, making the connection between quality AI feedback filtering with diversity-seeking, and minimal criteria novelty search (Lehman and Stanley, 2010) more clear.

We have expanded upon and clarified all points of concern and general questions raised by reviewers, from our understanding of the discussion raised. We would be grateful to be able to respond to any potential remaining questions from reviewers before the end of the author discussion period soon, and we hope that reviewers reflect on the merits of QDAIF towards their final decision, with in-depth treatment of all interesting discussions raised building upon the foundations of this simple approach to a new way of conducting QD search for improving the potential of interactions with foundation models.

We thank the reviewers again for their work on paper feedback that significantly improved our manuscript.

---

### Meta-Review · Area_Chair_ms5V · 2023-12-14

**Metareview:**

This is a very interesting paper that conceptually combines LLMs and QD, two important research directions, in a novel and, I would say, very fruitful way. It is also the first time I have seen QD applied to creative writing. The human evaluations indicate that this really works. It's true that there are no strong baselines to compare with, but that's because there are no strong baselines. This is a general problem for any paper introducing something new and should not be held against it.

**Justification For Why Not Higher Score:**

No reviewers are sufficiently enthusiastic to bump the paper up.

**Justification For Why Not Lower Score:**

It's a novel approach with good results.

---

### Decision · Program_Chairs · 2024-01-16

Accept (poster)